# Provably Global Convergence of Actor-Critic: A Case for Linear Quadratic Regulator with Ergodic Cost

**Zhuoran Yang**
Princeton University
zy6@princeton.edu

**Yongxin Chen**
Georgia Institute of Technology
yongchen@gatech.edu

**Mingyi Hong**
University of Minnesota
mhong@umn.edu

**Zhaoran Wang**
Northwestern University
zhaoran.wang@northwestern.edu

## Abstract

Despite the empirical success of the actor-critic algorithm, its theoretical understanding lags behind. In a broader context, actor-critic can be viewed as an online alternating update algorithm for bilevel optimization, whose convergence is known to be fragile. To understand the instability of actor-critic, we focus on its application to linear quadratic regulators, a simple yet fundamental setting of reinforcement learning. We establish a nonasymptotic convergence analysis of actor-critic in this setting. In particular, we prove that actor-critic finds a globally optimal pair of actor (policy) and critic (action-value function) at a linear rate of convergence. Our analysis may serve as a preliminary step towards a complete theoretical understanding of bilevel optimization with nonconvex subproblems, which is NP-hard in the worst case and is often solved using heuristics.

## 1   Introduction

The actor-critic algorithm [36] is one of the most used algorithms in reinforcement learning [46]. Compared with the classical policy gradient algorithm [73], actor-critic tracks the action-value function (critic) in policy gradient in an online manner, and alternatively updates the policy (actor) and the critic. On the one hand, the online update of critic significantly reduces the variance of policy gradient and hence leads to faster convergence. On the other hand, it also introduces algorithmic instability, which is often observed in practice [33] and parallels the notoriously unstable training of generative adversarial networks [50]. Such instability of actor-critic originates from several intertwining challenges, including (i) function approximation of actor and critic, (ii) improper choice of stepsizes, (iii) the noise arising from stochastic approximation, (iv) the asynchrony between actor and critic, and (v) possibly off-policy data used in the update of critic. As a result, the convergence of actor-critic remains much less well understood than that of policy gradient, which itself is open. Consequently, the practical use of actor-critic often lacks theoretical guidance.

In this paper, we aim to theoretically understand the algorithmic instability of actor-critic. In particular, under a bilevel optimization framework, we establish the global rate of convergence and sample complexity of actor-critic for linear quadratic regulators (LQR) with ergodic cost, a simple yet fundamental setting of reinforcement learning [54], which captures all the above challenges. Compared with the classical two-timescale analysis of actor-critic [11], which is asymptotic in nature and requires finite action space, our analysis is fully nonasymptotic and allows for continuous action space. Moreover, beyond the convergence to a stable equilibrium obtained by the classical two-timescale stochastic approximation via ordinary differential equations, we for the first time establish the linear rate of convergence to a globally optimal pair of actor and critic. In addition, we

characterize the required sample complexity. As a technical ingredient and byproduct, we for the first time establish the sublinear rate of convergence for the gradient temporal difference algorithm [66, 67] for ergodic cost and dependent data, which is of independent interest. Furthermore, although we only focus on the setting of LQR, our theoretical analysis framework can be readily extended to general RL problems with other policy optimization methods for the actor (e.g. trust-region policy optimization (TRPO) [57] and proximal policy optimization (PPO) [58]) and other policy evaluation methods for the critic such as TD(0) [65].

Our work adds to two lines of works in machine learning, stochastic analysis, and optimization:

(i) Actor-critic falls into the more general paradigm of bilevel optimization [42, 24, 4]. Bilevel optimization is defined by two nested optimization problems, where the upper-level optimization problem relies on the output of the lower-level one. As a special case of bilevel optimization, minimax optimization is prevalent in machine learning. Recent instances include training generative adversarial neural networks [27], (distributionally) robust learning [63], and imitation learning [31, 15]. Such instances of minimax optimization remain challenging as they lack convexity-concavity in general [25, 56, 16, 53, 38, 17, 18, 19, 41]. The more general paradigm of bilevel optimization remains even more challenging, as there does not exist a unified objective function for simultaneous minimization and maximization. In particular, actor-critic couples the nonconvex optimization of actor (policy gradient) as its upper level and the convex-concave minimax optimization of critic (gradient temporal difference) as its lower level, each of which is challenging to analyze by itself. Most existing convergence analysis of bilevel optimization is based on two-timescale analysis [10]. However, as two-timescale analysis abstracts away most technicalities via the lens of ordinary differential equations, which is asymptotic in nature, it often lacks the resolution to capture the nonasymptotic rate of convergence and sample complexity, which are obtained via our analysis.

(ii) As a proxy for analyzing more general reinforcement learning settings, LQR is studied in a recent line of works [12, 54, 26, 69, 70, 22, 23, 61, 21, 30]. In particular, a part of our analysis is based on the breakthrough of [26], which gives the global convergence of the population-version policy gradient algorithm for LQR and its finite-sample version based on the zeroth-order estimation of policy gradient based on the cumulative reward or cost. However, such zeroth-order estimation of policy gradient often suffers from large variance, as it involves the randomness of an entire trajectory. In contrast, actor-critic updates critic in an online manner, which reduces such variance but also introduces instability and complicates the convergence analysis. In particular, as the update of critic interleaves with the update of actor, the policy gradient for the update of actor is biased due to the inexactness of critic. Meanwhile, the update of critic has a "moving target", as it attempts to evaluate an actor that evolves along the iterations. A key to our analysis is to handle such asynchrony between actor and critic, which is a ubiquitous challenge in bilevel optimization. We hope our analysis may serve as the first step towards analyzing actor-critic in more general reinforcement learning settings.

**Notation.** For any integer $n > 0$, we denote $\{1, \ldots, n\}$ to be $[n]$. For any symmetric matrix $X$, let $\mathrm{svec}(X)$ denote the vectorization of the upper triangular submatrix of $X$ with off-diagonal entries weighted by $\sqrt{2}$. Hence, for any symmetric matrices $X$ and $Y$, we have $\mathrm{tr}(XY) = \langle X, Y \rangle = \mathrm{svec}(X)^\top \mathrm{svec}(Y)$. Meanwhile, let $\mathrm{smat}(\cdot)$ be the inverse operation of $\mathrm{svec}(\cdot)$, which maps a vector to a symmetric matrix. Besides, we denote by $A \otimes_s B$ the symmetric Kronecker product of $A$ and $B$. We use $\|v\|_2$ to denote the $\ell_2$-norm of a vector $v$. Finally, for a matrix $A$, we use $\|A\|$, $\|A\|_{\mathrm{fro}}$, and $\rho(A)$ to denote its the operator norm, Frobenius norm, and spectral radius, respectively.

## 2 Background

In the following, we introduce the background of actor-critic and LQR. In particular, we show that actor-critic can be cast as a first-order online alternating update algorithm for a bilevel optimization problem [42, 24, 4].

### 2.1 Actor-Critic Algorithm

We consider a Markov decision process, which is defined by $(\mathcal{X}, \mathcal{U}, P, c, D_0)$. Here $\mathcal{X}$ and $\mathcal{U}$ are the state and action spaces, respectively, $P\colon \mathcal{X} \times \mathcal{U} \to \mathcal{P}(\mathcal{X})$ is the Markov transition kernel, $c\colon \mathcal{X} \times \mathcal{U} \to \mathbb{R}$ is the cost function, and $D_0 \in \mathcal{P}(\mathcal{X})$ is the distribution of the initial state $x_0$. For

any $t \geq 0$, at the $t$-th time step, the agent takes action $u_t \in \mathcal{U}$ at state $x_t \in \mathcal{X}$, which incurs a cost $c(x_t, u_t)$ and moves the environment into a new state $x_{t+1} \sim P(\cdot \,|\, x_t, u_t)$. A policy specifies how the action $u_t$ is taken at a given state $x_t$. Specifically, in order to handle infinite state and action spaces, we focus on a parametrized policy class $\{\pi_\omega \colon \mathcal{X} \to \mathcal{P}(\mathcal{U}), \omega \in \Omega\}$, where $\omega$ is the parameter of policy $\pi_\omega$, and the agent takes action $u \sim \pi_\omega(\cdot \,|\, x)$ at a given state $x \in \mathcal{X}$. The agent aims to find a policy that minimizes the infinite-horizon time-average cost, that is,

$$\operatorname*{minimize}_{\omega \in \Omega} \ J(\omega) = \lim_{T \to \infty} \mathbb{E}\left[ \frac{1}{T} \sum_{t=0}^{T} c(x_t, u_t) \,\bigg|\, x_0 \sim D_0, u_t \sim \pi_\omega(\cdot \,|\, x_t), \forall t \geq 0 \right]. \tag{2.1}$$

Moreover, for policy $\pi_\omega$, we define the (advantage) action-value and state-value functions respectively as

$$\begin{aligned} Q_\omega(x, u) &= \sum_{t \geq 0} \mathbb{E}_\omega\big[ c(x_t, u_t) \,|\, x_0 = x, u_0 = u \big] - J(\omega), \\ V_\omega(x) &= \mathbb{E}_{u \sim \pi_\omega(\cdot \,|\, x)}\big[ Q_\omega(x, u) \big], \end{aligned} \tag{2.2}$$

where we use $\mathbb{E}_\omega$ to indicate that the state-action pairs $\{(x_t, u_t)\}_{t \geq 1}$ are obtained from policy $\pi_\omega$.

Actor-critic is based on the idea of solving the minimization problem in (2.1) via first-order optimization, which uses an estimator of $\nabla_\omega J(\omega)$. In detail, by the policy gradient theorem [68, 6, 36], we have

$$\nabla_\omega J(\omega) = \mathbb{E}_{x \sim \rho_\omega, u \sim \pi_\omega(\cdot \,|\, x)}\big[ \nabla_\omega \log \pi_\omega(u \,|\, x) \cdot Q_\omega(x, u) \big], \tag{2.3}$$

where $\rho_\omega \in \mathcal{P}(\mathcal{X})$ is the stationary distribution induced by $\pi_\omega$. Based on (2.3), actor-critic [36] consists two steps: (i) a policy evaluation step that estimates the action-value function $Q_\omega$ (critic) via temporal difference learning [20], where $Q_\omega$ is estimated using a parametrized function class $\{Q^\theta \colon \theta \in \Theta\}$, and (ii) a policy improvement step that updates the parameter $\omega$ of policy $\pi_\omega$ (actor) using a stochastic version of the policy gradient in (2.3), where $Q_\omega$ is replaced by the corresponding estimator $Q^\theta$.

As shown in [74], actor-critic can be cast as solving a bilevel optimization problem, which takes the form

$$\operatorname*{minimize}_{\omega \in \Omega} \quad \mathbb{E}_{x \sim \rho_\omega, u \sim \pi_\omega(\cdot \,|\, x)}\big[ Q^\theta(x, u) \big], \tag{2.4}$$

$$\text{subject to} \quad (\theta, J) = \operatorname*{argmin}_{\theta \in \Theta, J \in \mathbb{R}} \mathbb{E}_{x \sim \rho_\omega, u \sim \pi_\omega(\cdot \,|\, s)}\Big\{ \big[ Q^\theta(x, u) + J - c(x, u) - (\mathcal{B}^\omega Q^\theta)(x, u) \big]^2 \Big\}, \tag{2.5}$$

where $\mathcal{B}^\omega$ is an operator that depends on $\pi_\omega$. In this problem, the actor and critic correspond to the upper-level and lower-level variables, respectively. Under this framework, the policy update can be viewed as a stochastic gradient step for the upper-level problem in (2.4). The objective in (2.5) is usually the mean-squared Bellman error or mean-squared projected Bellman error [20]. Moreover, when $\mathcal{B}^\omega$ is the Bellman evaluation operator associated with $\pi_\omega$ and we solve the lower-level problem in (2.5) via stochastic semi-gradient descent, we obtain the TD(0) update for policy evaluation [65]. Similarly, when $\mathcal{B}^\omega$ is the projected Bellman evaluation operator associated with $\pi_\omega$, solving the lower level problem naturally recovers the GTD2 and TDC algorithms for policy evaluation [8]. Therefore, the actor-critic algorithm is a first-order online algorithm for the bilevel optimization problem in (2.4) and (2.5). We remark that bilevel optimization contains a family of extremely challenging problems. Even when the objective functions are linear, bilevel programming is NP-hard [29]. In practice, various heuristic algorithms are applied to solve them approximately [62].

## 2.2 Linear Quadratic Regulator

As the simplest optimal control problem, linear quadratic regulator serves as a perfect baseline to examine the performance of reinforcement learning methods. Viewing LQR from the lens of an MDP, the state and action spaces are $\mathcal{X} = \mathbb{R}^d$ and $\mathcal{U} = \mathbb{R}^k$, respectively. The state transition dynamics and cost function are specified by

$$x_{t+1} = A x_t + B u_t + \epsilon_t, \qquad c(x, u) = x^\top Q x + u^\top R u, \tag{2.6}$$

where $\epsilon_t \sim N(0, \Psi)$ is the random noise that is i.i.d. for each $t \geq 0$, and $A$, $B$, $Q$, $R$, $\Psi$ are matrices of proper dimensions with $Q, R, \Psi \succ 0$. Moreover, we assume that the dimensions $d$ and $k$ are fixed throughout this paper. For the problem of minimizing the infinite-horizon time-average cost $\limsup_{T \to \infty} T^{-1} \sum_{t=0}^{T} \mathbb{E}[c(x_t, u_t)]$ with $x_0 \sim D_0$, it is known that the optimal actions are linear in the corresponding state [76, 3, 7]. Specifically, the optimal actions $\{u_t^*\}_{t \geq 0}$ satisfy $u_t^* = -K^* x_t$ for all $t \geq 0$, where $K^* \in \mathbb{R}^{k \times d}$ can be written as $K^* = (R + B^\top P^* B)^{-1} B^\top P^* A$, with $P^*$ being the solution to the discrete algebraic Riccati equation

$$P^* = Q + A^\top P^* A + A^\top P^* B (R + B^\top P^* B)^{-1} B^\top P^* A. \tag{2.7}$$

In the optimal control literature, it is common to solve LQR by first estimating matrices $A$, $B$, $Q$, $R$ and then solving the Riccati equation in (2.7) with these matrices replaced by their estimates. Such an approach is known as model-based as it requires estimating the model parameters and the performance of the planning step in (2.7) hinges on how well the true model is estimated. See, e.g, [21, 70] for theoretical guarantees of model-based methods.

In contrast, from a purely data-driven perspective, the framework of model-free reinforcement learning offers a general treatment for optimal control problems without the prior knowledge of the model. Thanks to its simple structure, LQR enables us to assess the performances of reinforcement learning algorithms from a theoretical perspective. Specifically, it is shown that policy iteration [12, 14, 45], adaptive dynamic programming [52], and policy gradient methods [26, 44, 70] are all able to obtain the optimal policy of LQR. Also see [54] for a thorough review of reinforcement learning methods in the setting of LQR.

# 3 Actor-Critic Algorithm for LQR

In this section, we establish the actor-critic algorithm for the LQR problem introduced in §2.2. Recall that the optimal policy of LQR is a linear function of the state. Throughout the rest of this paper, we focus on the family of linear-Gaussian policies

$$\left\{ \pi_K(\cdot \,|\, x) = N(-Kx, \sigma^2 I_k), K \in \mathbb{R}^{k \times d} \right\}, \tag{3.1}$$

where $\sigma > 0$ is a fixed constant. That is, for any $t \geq 0$, at state $x_t$, we could write the action $u_t$ by $u_t = -Kx_t + \sigma \cdot \eta_t$, where $\eta_t \sim N(0, I_k)$. We note that if $\sigma = 0$, then the optimal policy $u = -K^* x$ belongs to our policy class. Here, instead of focusing on deterministic policies, we adopt Gaussian policies to encourage exploration. For policy $\pi_K$, the corresponding time-average cost $J(K)$, state-value function $V_K$, and action-value function $Q_K$ are specified as in (2.1) and (2.2), respectively.

In the following, we first establish the policy gradient and value functions for the ergodic LQR in §3.1. Then, in §3.2, we present the on-policy natural actor-critic algorithm, which is further extended to the off-policy setting in §B.

## 3.1 Policy Gradient Theorem for Ergodic LQR

For any policy $\pi_K$, by (2.6), the state dynamics are given by a linear dynamical system

$$x_{t+1} = (A - BK)x_t + \varepsilon_t, \qquad \text{where} \quad \varepsilon_t = \epsilon_t + \sigma \cdot B\eta_t \sim N(0, \Psi_\sigma). \tag{3.2}$$

Here we define $\Psi_\sigma := \Psi + \sigma^2 \cdot BB^\top$ in (3.2) to simplify the notation. It is known that, when $\rho(A - BK) < 1$, the Markov chain in (3.2) has stationary distribution $N(0, \Sigma_K)$, denoted by $\nu_K$ hereafter, where $\Sigma_K$ is the unique positive definite solution to the Lyapunov equation

$$\Sigma_K = \Psi_\sigma + (A - BK)\Sigma_K(A - BK)^\top. \tag{3.3}$$

In the following proposition, we establish $J(K)$, the value functions, and the gradient $\nabla_K J(K)$.

**Proposition 3.1.** For any $K \in \mathbb{R}^{k \times d}$ such that $\rho(A - BK) < 1$, let $P_K$ be the unique positive definite solution to the Bellman equation

$$P_K = (Q + K^\top RK) + (A - BK)^\top P_K(A - BK). \tag{3.4}$$

In the setting of LQR, for policy $\pi_K$, both the state- and action-value functions are quadratic. Specifically, we have

$$V_K(x) = x^\top P_K x - \mathrm{tr}(P_K \Sigma_K), \tag{3.5}$$

$$Q_K(x,u) = x^\top \Theta_K^{11} x + x^\top \Theta_K^{12} u + u^\top \Theta_K^{21} x + u^\top \Theta_K^{22} u - \sigma^2 \cdot \mathrm{tr}(R + P_K B B^\top) - \mathrm{tr}(P_K \Sigma_K), \tag{3.6}$$

where $\Sigma_K$ is specified in (3.3), and we define matrix $\Theta_K$ by

$$\Theta_K = \begin{pmatrix} \Theta_K^{11} & \Theta_K^{12} \\ \Theta_K^{21} & \Theta_K^{22} \end{pmatrix} = \begin{pmatrix} Q + A^\top P_K A & A^\top P_K B \\ B^\top P_K A & R + B^\top P_K B \end{pmatrix}. \tag{3.7}$$

Moreover, the time-average cost $J(K)$ and its gradient are given by

$$J(K) = \mathrm{tr}\big[(Q + K^\top R K)\Sigma_K\big] + \sigma^2 \cdot \mathrm{tr}(R) = \mathrm{tr}(P_K \Psi_\sigma) + \sigma^2 \cdot \mathrm{tr}(R). \tag{3.8}$$

$$\nabla_K J(K) = 2\big[(R + B^\top P_K B)K - B^\top P_K A\big]\Sigma_K = 2 E_K \Sigma_K, \tag{3.9}$$

where we define $E_K := (R + B^\top P_K B)K - B^\top P_K A$.

*Proof.* See §D.1 for a detailed proof. □

To see the connection between (3.9) and the policy gradient theorem in (2.3), note that by direct computation we have

$$\nabla_K \log \pi_K(u \,|\, x) = \nabla_K \big[-(2\sigma^2)^{-1} \cdot (u + Kx)^2\big] = -\sigma^{-2} \cdot (u + Kx)x^\top. \tag{3.10}$$

Thus, combining, (3.6), (3.10), and the fact that $u = -Kx + \sigma \cdot \eta$ under $\pi_K$, the right-hand side of (2.3) can be written as

$$\mathbb{E}_{x \sim \nu_K, u \sim \pi_K}\big[\nabla_K \log \pi_K(u \,|\, x) \cdot Q_K(x,u)\big] = -\sigma^{-2} \cdot \mathbb{E}_{x \sim \nu_K, \eta \sim N(0,I_k)}[\sigma \cdot \eta x^\top \cdot Q_K(x, -Kx + \sigma \cdot \eta)].$$

Recall that for $\eta \in N(0, I_k)$, Stein's identity [64], $\mathbb{E}[\eta \cdot f(\eta)] = \mathbb{E}[\nabla f(\eta)]$, holds for all differentiable function $f \colon \mathbb{R}^k \to \mathbb{R}$, which implies that

$$\begin{aligned}
\nabla_K J(K) &= -\mathbb{E}_{x \sim \nu_K, \eta \sim N(0,I_d)}[(\nabla_u Q_K)(x, -Kx + \sigma \cdot \eta) \cdot x^\top] \\
&= -2\mathbb{E}_{x \sim \nu_K, \eta \sim N(0,I_d)}\big\{[(R + B^\top P_K B)(-Kx + \sigma \cdot \eta) + B^\top P_K Ax]x^\top\big\} \\
&= 2\big[(R + B^\top P_K B)K - B^\top P_K A\big]\Sigma_K = 2 E_K \Sigma_K. \tag{3.11}
\end{aligned}$$

Thus, (3.9) is exactly the policy gradient theorem (2.3) in the setting of LQR. Moreover, it is worth noting that Proposition 3.1 also holds for $\sigma = 0$. Thus, setting $\sigma = 0$ in (3.11), we obtain

$$\nabla_K J(K) = -\mathbb{E}_{x \sim \nu_K}[(\nabla_u Q_K)(x, -Kx) \cdot x^\top] = \mathbb{E}_{x \sim \nu_K}\big[(\nabla_u Q_K)(x,u)\big|_{u=\pi_K(x)} \nabla_K \pi_K(x)\big],$$

where $\pi_K(x) = -Kx$. Thus we obtain the deterministic policy gradient theorem [60] for LQR. Although the optimal policy for LQR is deterministic, due to the lack of exploration, behaving according to a deterministic policy may lead to suboptimal solutions. Thus, we focus on the family of stochastic policies where a Gaussian noise $\sigma \cdot \eta$ is added to the action so as to promote exploration.

### 3.2 Natural Actor-Critic Algorithm

Natural policy gradient updates the variable along the steepest direction with respect to Fisher metric. For the Gaussian policies defined in (3.1), by (3.10), the Fisher's information of policy $\pi_K$, denoted by $\mathcal{I}(K)$, is given by

$$\begin{aligned}
[\mathcal{I}(K)]_{(i,j),(i',j')} &= \mathbb{E}_{x \sim \nu_K, a \sim \pi_K}\big[\nabla_{K_{ij}} \log \pi_K(u \,|\, x) \cdot \nabla_{K_{i'j'}} \log \pi_K(u \,|\, x)\big] \\
&= \sigma^{-2} \cdot \mathbb{E}_{x \sim \nu_K, \eta \sim N(0,I_k)}[\eta_i x_j \cdot \eta_{i'} x_{j'}] = \sigma^{-2} \cdot \mathbb{1}\{i = i'\} \cdot [\Sigma_K]_{jj'}, \tag{3.12}
\end{aligned}$$

where $i, i' \in [k]$, $j, j' \in [d]$, $K_{ij}$ and $K_{i'j'}$ are the $(i,j)$- and $(i',j')$-th entries of $K$, respectively, and $[\Sigma_K]_{jj'}$ is the $(j,j')$-th entry of $\Sigma_K$. Thus, in view of (3.9) in Proposition 3.1 and (3.12), natural policy gradient algorithm updates the policy parameter in the direction of

$$[\mathcal{I}(K)]^{-1}\nabla_K J(K) = \nabla_K J(K)\Sigma_K^{-1} = 2 E_K.$$

By (3.7), we can write $E_K$ as $\Theta_K^{22} K - \Theta_K^{21}$, where $\Theta_K$ is the coefficient matrix of the quadratic component of $Q_K$. Such a connection lays the foundation of the natural actor-critic algorithm. Specifically, in each iteration of the algorithm, the actor updates the policy via $K - \gamma \cdot (\widehat{\Theta}^{22} K - \widehat{\Theta}^{21})$, where $\gamma$ is the stepsize and $\widehat{\Theta}$ is an estimator of $\Theta_K$ returned by any policy evaluation algorithm. We present such a general natural actor-critic method in Algorithm 1 §A, under the assumption that we are given a stable policy $K_0$ for initialization. Such an assumption is standard in literatures on model-free methods for LQR [22, 26, 44].

To obtain an online actor-critic algorithm, in the sequel, we propose an online policy evaluation algorithm for ergodic LQR based on temporal difference learning. Let $\pi_K$ be the policy of interest. For notational simplicity, for any state-action pair $(x, u) \in \mathbb{R}^{d+k}$, we define the feature function

$$\phi(x, u) = \mathrm{svec}\left[ \begin{pmatrix} x \\ u \end{pmatrix} \begin{pmatrix} x \\ u \end{pmatrix}^{\top} \right], \tag{3.13}$$

and denote $\mathrm{svec}(\Theta_K)$ by $\theta_K^*$. Using this notation, the quadratic component in $Q_K$ can be written as $\phi(x, u)^{\top} \theta_K^*$, and the Bellman equation for $Q_K$ becomes

$$\langle \phi(x, u), \theta_K^* \rangle = c(x, u) - J(K) + \langle \mathbb{E}[\phi(x', u') \,|\, x, u], \theta_K^* \rangle, \ \forall (x, u) \in \mathbb{R}^{d+k}. \tag{3.14}$$

In order to further simplify the notation, hereafter, we define $\vartheta_K^* = (J(K), \theta_K^{*\top})^{\top}$, denote by $\mathbb{E}_{(x,u)}$ the expectation with respect to $x \sim \nu_K$ and $u \sim \pi_K(\cdot \,|\, x)$, and let $(x', u')$ be the state-action pair subsequent to $(x, u)$.

Furthermore, to estimate $J(K)$ and $\theta_K^*$ in (3.14) simultaneously, we define

$$\Xi_K = \mathbb{E}_{(x,u)}\big\{ \phi(x, u)\big[\phi(x, u) - \phi(x', u')\big]^{\top} \big\}, \qquad b_K = \mathbb{E}_{(x,u)}\big[ c(x, u)\phi(x, u) \big], \tag{3.15}$$

Notice that $J(K) = \mathbb{E}_{(x,u)}[c(x, u)]$. By direct computation, it can be shown that $\vartheta_K^*$ satisfies the following linear equation

$$\begin{pmatrix} 1 & 0 \\ \mathbb{E}_{(x,u)}[\phi(x, u)] & \Xi_K \end{pmatrix} \begin{pmatrix} \vartheta^1 \\ \vartheta^2 \end{pmatrix} = \begin{pmatrix} J(K) \\ b_K \end{pmatrix}, \tag{3.16}$$

whose solution is unique if and only if $\Xi_K$ in (3.15) is invertible. The following lemma shows that, when $\pi_K$ is a stable policy, $\Xi_K$ is indeed invertible.

**Lemma 3.2.** When $\pi_K$ is stable in the sense that $\rho(A - BK) < 1$, $\Xi_K$ defined in (3.15) is invertible and thus $\vartheta_K^*$ is the unique solution to the linear equation (3.16). Furthermore, the minimum singular value of the matrix in the left-hand side of (3.16) is lower bounded by a constant $\kappa_K^* > 0$, where $\kappa_K^*$ only depends on $\rho(A - BK)$, $\sigma$, and $\sigma_{\min}(\Psi)$.

*Proof.* See §D.2 for a detailed proof. $\qquad\qquad\qquad\qquad\qquad\qquad\qquad\qquad\qquad\qquad\square$

By this lemma, when $\rho(A - BK) < 1$, policy evaluation for $\pi_K$ can be reduced to finding the unique solution to a linear equation. Instead of solving the equation directly, it is equivalent to minimize the least-squares loss:

$$\underset{\vartheta}{\mathrm{minimize}}\Big\{ [\vartheta^1 - J(K)]^2 + \big\| \vartheta^1 \cdot \mathbb{E}_{(x,u)}[\phi(x, u)] + \Xi_K \vartheta^2 - b_K \big\|_2^2 \Big\}, \tag{3.17}$$

where $\vartheta^1 \in \mathbb{R}$ and $\vartheta^2$ has the same shape as $\mathrm{svec}(\Theta_K)$, which are the two components of $\vartheta$. It is clear that the global minimizer of (3.17) is $\vartheta_K^*$. Note that we have Fenchel's duality $x^2 = \sup_y\{2x \cdot y - y^2\}$. By this relation, we further write (3.17) as a minimax optimization problem

$$\min_{\vartheta \in \mathcal{X}_{\Theta}} \max_{\omega \in \mathcal{X}_{\Omega}} F(\vartheta, \omega) = [\vartheta^1 - J(K)] \cdot \omega^1$$
$$+ \big\langle \vartheta^1 \cdot \mathbb{E}_{(x,u)}[\phi(x, u)] + \Xi_K \vartheta^2 - b_K, \omega^2 \big\rangle - 1/2 \cdot \|\omega\|_2^2, \tag{3.18}$$

where the dual variable $\omega = (\omega^1, \omega^2)$ has the same shape as $\vartheta$. Here we restrict the primal and dual variables to compact sets $\mathcal{X}_{\Theta}$ and $\mathcal{X}_{\Omega}$ for algorithmic stability, which will be specified in the next section. Note that the objective in (3.18) can be estimated unbiasedly using two consecutive state-action pairs $(x, u)$ and $(x', u')$. Solving the minimax optimization in (3.18) using stochastic

gradient method, we obtain the gradient-based temporal difference (GTD) algorithm for policy evaluation [66, 67]. See Algorithm 2 for details. More specifically, by direct computation, we have

$$\nabla_{\vartheta^1} F(\theta, \omega) = \omega^1 + \langle \mathbb{E}_{(x,u)}(\phi), \omega^2 \rangle, \qquad \nabla_{\vartheta^2} F(\theta, \omega) = \mathbb{E}_{(x,u)}\big[(\phi - \phi') \cdot \phi^\top \omega^2\big], \qquad (3.19)$$

$$\nabla_{\omega^1} F(\theta, \omega) = \vartheta^1 - J(K) - \omega^1, \qquad \nabla_{\omega^2} F(\theta, \omega) = \theta^1 \cdot \mathbb{E}_{(x,u)}(\phi) + \Xi_K \vartheta^2 - b_K - \omega^2, \qquad (3.20)$$

where we denote $\phi(x, u)$ and $\phi(x', u')$ by $\phi$ and $\phi'$, respectively. In the GTD algorithm, we update $\vartheta$ and $\omega$ in gradient directions where the gradients in (3.19) and (3.20) are replaced by their sample estimates. After $T$ iterations of the algorithm, we output the averaged update $\widehat{\vartheta^2} = (\sum_{t=1}^T \alpha_t \cdot \vartheta_t^2)/(\sum_{t=1}^T \alpha_t)$ and use $\widehat{\Theta} = \text{smat}(\widehat{\vartheta^2})$ to estimate $\Theta_K$ in (3.7), which is further used in Algorithm 1 to update the current policy $\pi_K$. Therefore, we obtain the online natural actor-critic algorithm [9] for ergodic LQR.

Meanwhile, using the perspective of bilevel optimization, similar to (2.4) and (2.5), our actor-critic algorithm can be viewed as a first-order online algorithm for

$$\begin{aligned}
\underset{K \in \mathbb{R}^{k \times d}}{\text{minimize}} \quad & \mathbb{E}_{x \sim \nu_K, u \sim \pi_K}\big[\langle \phi(x, u), \vartheta^2 \rangle\big], \\
\text{subject to} \quad & (\vartheta, \omega) = \underset{\theta \in \mathcal{X}_\Theta}{\text{argmin}} \, \underset{\omega \in \mathcal{X}_\Omega}{\text{argmax}} \, F(\vartheta, \omega),
\end{aligned} \qquad (3.21)$$

where $F(\vartheta, \omega)$ is defined in (3.18) and depends on $\pi_K$. In our algorithm, we solve the upper-level problem via natural gradient descent and solve the lower-level saddle point optimization problem using stochastic gradient updates.

Furthermore, we emphasize that our method defined by Algorithms 1 and 2 is online in the sense that each update only requires a single transition. More specifically, let $\{(x_n, u_n, c_n)\}_{n \geq 0}$ be the sequence of transitions experienced by the agent. Combining Algorithms 1 and 2 and neglecting the projections, we can write the updating rule as

$$\begin{aligned}
K_{n+1} &= K_n - \overline{\gamma}_n \cdot \big\{ [\text{smat}(\vartheta_n^2)]^{22} K_n - [\text{smat}(\vartheta_n^2)]^{21} \big\}, \\
\vartheta_{n+1} &= \vartheta_n - \overline{\alpha}_n \cdot g_\vartheta(x_n, u_n, c_n, x_{n+1}, u_{n+1}), \\
\omega_{n+1} &= \omega_n + \overline{\alpha}_n \cdot g_\omega(x_n, u_n, c_n, x_{n+1}, u_{n+1}),
\end{aligned} \qquad (3.22)$$

where $g_\vartheta$ and $g_\omega$ are the update directions of $\vartheta$ and $\omega$ whose definitions are clear from Algorithm 2, $\{\overline{\gamma}_n\}$ and $\{\overline{\alpha}_n\}$ are the stepsizes. Moreover, there exists a monotone increasing sequence $\{N_t\}_{t \geq 0}$ such that $\overline{\gamma}_n = \gamma$ if $n = N_t$ for some $t$ and $\overline{\gamma}_n = 0$ otherwise. Such a choice of the stepsizes reflects the intuition that, although both the actor and the critic are updated simultaneously, the critic should be updated at a faster pace. From the same viewpoint, classical actor-critic algorithms [36, 9, 28] establish convergence results under the assumption that

$$\sum_{n \geq 0} \overline{\gamma}_n = \sum_{n \geq 0} \overline{\alpha}_n = \infty, \qquad \sum_{n \geq 0} (\overline{\gamma}_n^2 + \overline{\alpha}_n^2) < \infty, \qquad \lim_{n \to \infty} \overline{\gamma}_n / \overline{\alpha}_n = 0.$$

The condition that $\overline{\gamma}_n / \overline{\alpha}_n = 0$ ensures that the critic updates at a faster timescale, which enables the asymptotic analysis utilizing two-timescale stochastic approximation [10, 37]. However, such an approach uses two ordinary differential equations (ODE) to approximate the updates in (3.22) and thus only offers asymptotic convergence results. In contrast, as shown in §4, our choice of the stepsizes yields nonasymptotic convergence results which shows that natural actor gradient converges in linear rate to the global optimum.

In addition, we note that in Algorithm 2 we assume that the initial state $x_0$ is sampled from the stationary distribution $\nu_K$. Such an assumption is made only to simplify theoretical analysis. In practice, we could start the algorithm after sampling a sufficient number of transitions so that the Markov chain induced by $\pi_K$ approximately mixes. Moreover, as shown in [69], when $\pi_K$ is a stable policy such that $\rho(A - BK) < 1$, the Markov chain induced by $\pi_K$ is geometrically $\beta$-mixing and thus mixes rapidly.

Finally, we remark that the minimax formulation of the policy evaluation problem is first proposed in [40], which studies the sample complexity of the GTD algorithm for discounted MDPs with i.i.d. data. Using the same formulation, [72] establishes finite sample bounds with data generated from a Markov process. Our optimization problem in (3.18) can be viewed as the extension of their minimax formulation to the ergodic setting. Besides, our GTD algorithm can be applied to ergodic MDPs in general with dependent data, which might be of independent interest.

# 4 Theoretical Results

In this section, we establish the global convergence of the natural actor-critic algorithm. To this end, we first focus on the problem of policy evaluation by assessing the finite sample performance of the on-policy GTD algorithm.

Note that only $\widehat{\Theta}$ returned by the GTD algorithm is utilized in the natural actor-critic algorithm for policy update. Thus, in the policy evaluation problem for a linear policy $\pi_K$, we only need to study the estimation error $\|\widehat{\Theta} - \Theta_K\|_{\text{fro}}^2$, which characterizes the closeness between the direction of policy update in Algorithm 1 and the true natural policy gradient.

Furthermore, recall that we restrict the primal and dual variables respectively to compact sets $\mathcal{X}_\Theta$ and $\mathcal{X}_\Omega$ for algorithmic stability. We make the following assumption on $\mathcal{X}_\Theta$ and $\mathcal{X}_\Omega$.

**Assumption 4.1.** Let $\pi_{K_0}$ be the initial policy in Algorithm 1. We assume that $\pi_{K_0}$ is a stable policy such that $\rho(A - BK_0) < 1$. Consider the policy evaluation problem for $\pi_K$. We assume that $J(K) \leq J(K_0)$. Moreover, let $\mathcal{X}_\Theta$ and $\mathcal{X}_\Omega$ in (3.18) be defined as

$$\mathcal{X}_\Theta = \{\vartheta\colon 0 \leq \vartheta^1 \leq J(K_0), \|\vartheta^2\|_2 \leq \widetilde{R}_\Theta\}, \tag{4.1}$$

$$\mathcal{X}_\Omega = \{\omega\colon |\omega^1| \leq J(K_0), \|\omega^2\|_2 \leq (1 + \|K\|_{\text{fro}}^2)^2 \cdot \widetilde{R}_\Omega\}. \tag{4.2}$$

Here, $\widetilde{R}_\Theta$ and $\widetilde{R}_\Omega$ are two parameters that do not depend on $K$. Specifically, we have

$$\widetilde{R}_\Theta = \|Q\|_{\text{fro}} + \|R\|_{\text{fro}} + \sqrt{d}/\sigma_{\min}(\Psi) \cdot (\|A\|_{\text{fro}}^2 + \|B\|_{\text{fro}}^2) \cdot J(K_0), \tag{4.3}$$

$$\widetilde{R}_\Omega = C \cdot \widetilde{R}_\Theta \cdot \sigma_{\min}^{-2}(Q) \cdot [J(K_0)]^2, \tag{4.4}$$

where $C > 0$ is a constant.

The assumption that we have access to a stable policy $K_0$ for initialization is commonly made in literatures on model-free methods for LQR [22, 26, 44]. Besides, $\rho(A - BK_0) < 1$ implies that $J(K_0)$ is finite. Here we assume $J(K) \leq J(K_0)$ for simplicity. Even if $J(K) > J(K_0)$, we can replace $J(K_0)$ in (4.1)–(4.4) by $J(K)$ and the theory of policy evaluation still holds. Moreover, as we will show in Theorem 4.3, the actor-critic algorithm creates a sequence policies whose objective values decreases monotonically. Thus, here we assume $J(K) \leq J(K_0)$ without loss of generality.

Furthermore, as shown in the proof, the construction of $\widetilde{R}_\Theta$ and $\widetilde{R}_\Omega$ ensures that $(\vartheta_K^*, 0)$ is the saddle-point of the minimax optimization in (3.18). In other words, the solution to (3.18) is the same as the unconstrained problem $\min_\vartheta \max_\omega F(\vartheta, \omega)$. When replacing the population problem by a sample-based optimization problem, restrictions on the primal and dual variables ensures that the iterates of the GTD algorithm remains bounded. Thus, setting $\widetilde{R}_\Theta$ and $\widetilde{R}_\Omega$ essentially guarantees that restricting $(\vartheta, \omega)$ to $\mathcal{X}_\Theta \times \mathcal{X}_\Omega$ incurs no "bias" in the optimization problem.

We present the theoretical result for the online GTD algorithm as follows.

**Theorem 4.2** (Policy evaluation). Let $\widehat{\vartheta}^1$ and $\widehat{\Theta}$ be the output of Algorithm 2 based on $T$ iterations. We set the stepsize to be $\alpha_t = \alpha/\sqrt{t}$ with $\alpha > 0$ being a constant. Under Assumption 4.1, for any $\rho \in (\rho(A - BK), 1)$, when the number of iterations $T$ is sufficiently large, with probability at least $1 - T^{-4}$, we have

$$\|\widehat{\Theta} - \Theta_K\|_{\text{fro}}^2 \leq \frac{\Upsilon[\widetilde{R}_\Theta, \widetilde{R}_\Omega, J(K_0), \|K\|_{\text{fro}}, \sigma_{\min}^{-1}(Q)]}{\kappa_K^{*2} \cdot (1 - \rho)} \cdot \frac{\log^6 T}{\sqrt{T}}, \tag{4.5}$$

where $\Upsilon[\widetilde{R}_\Theta, \widetilde{R}_\Omega, J(K_0), \|K\|_{\text{fro}}, \sigma_{\min}^{-1}(Q)]$ is a polynomial of $\widetilde{R}_\Omega$, $\widetilde{R}_\Omega$, $J(K_0)$, $\|K\|_{\text{fro}}$, and $1/\sigma_{\min}(Q)$.

*Proof.* See §C.1 for a detailed proof. □

This theorem establishes the statistical rate of convergence for the on-policy GTD algorithm. Specifically, if we regard $\Upsilon[\widetilde{R}_\Theta, \widetilde{R}_\Omega, J(K_0), \|K\|_{\text{fro}}, \sigma_{\min}^{-1}(Q)]$, $\rho$, and $\kappa_K^*$ as constant, (4.5) implies that the estimation error is of order $\log^6 T/\sqrt{T}$. Thus, ignoring the logarithmic term, we conclude that the GTD algorithm converges in the sublinear rate $\mathcal{O}(1/\sqrt{T})$, which is optimal for convex-concave

stochastic optimization [49] and is also identical to the rate of convergence of the GTD algorithm in the discounted setting with bounded data [40, 72]. Note that we focus on the ergodic case and the feature mapping $\phi(x, u)$ defined in (3.13) is unbounded. We believe this theorem might be of independent interest. Furthermore, $1/\kappa_K^*$ is approximately the condition number of the linear equation of (3.16), which reflects the fundamental difficulty of estimating $\Theta_K$. Specifically, when $\kappa_K^*$ is close to zero, the matrix on the left-hand side of (3.16) is close to a singular matrix. In this case, estimating $\Theta_K$ can be viewed as solving an ill-conditioned regression problem and thus huge sample size is required for consistent estimation. Finally, $1/[1 - \rho(A - BK)]$ also reflects the intrinsic hardness of estimating $\Theta_K$. Specifically, for any $\rho \in (\rho(A - BK), 1)$, the Markov chain induced by $\pi_K$ is $\beta$-mixing where the $k$-th mixing coefficients is bounded by $C \cdot \rho^k$ for some constant $C > 0$ [69]. Thus, when $\rho$ is close to one, this Markov chain becomes more dependent, which makes the estimation problem more difficult.

Equipped with the finite sample error of the policy evaluation algorithm, now we are ready to present the global convergence of the actor-critic algorithm. For ease of presentation, we assume that $Q$, $R$, $A$, $B$, $\Psi$ are all constant matrices.

**Theorem 4.3** (Global convergence of actor-critic). Let the initial policy $K_0$ be stable. We set the stepsize $\gamma = [\|R\| + \sigma_{\min}^{-1}(\Psi) \cdot \|B\|^2 \cdot J(K_0)]$ in Algorithm 1 and perform $N$ actor updates in the actor-critic algorithm. Let $\{K_t\}_{0 \le t \le N}$ be the sequence of policy parameters generated by the algorithm. For any sufficiently small $\epsilon > 0$, we set $N > C \cdot \|\Sigma_{K^*}\|/\gamma \cdot \log\{2[J(K_0) - J(K^*)]/\epsilon\}$ for some constant $C > 0$. Moreover, for any $t \in \{0, 1, \ldots, N\}$, in the $t$-th iteration, we set the number $T_t$ of GTD updates in Algorithm 2 to be

$$T_t \ge \Upsilon[\|K_t\|, J(K_0)] \cdot {\kappa_{K_t}^*}^{-5} \cdot [1 - \rho(A - BK_t)]^{-5/2} \cdot \epsilon^{-5},$$

where $\Upsilon[\|K_t\|, J(K_0)]$ is a polynomial of $\|K_t\|$ and $J(K_0)$. Then with probability at least $1 - \epsilon^{10}$, we have $J(K_N) - J(K^*) \le \epsilon$.

*Proof Sketch.* The proof of this Theorem is based on combining the convergence of the natural policy gradient and the finite sample analysis of the GTD algorithm established in Theorem 4.2. Specifically, for each $K_t$, we define $K'_{t+1} = K_t - \eta \cdot E_{K_t}$, which is the one-step natural policy gradient update starting from $K_t$. Similar to [26], for ergodic LQR, it can be shown that

$$J(K'_{t+1}) - J(K^*) \le [1 - C_1 \cdot \gamma \cdot \|\Sigma_{K^*}\|^{-1}] \cdot [J(K_t) - J(K^*)] \qquad (4.6)$$

for some constant $C_1 > 0$. In addition, for policy $\pi_{K_t}$, when the number of GTD iteration $T_t$ is sufficiently large, $K_{t+1}$ is close to $K'_{t+1}$, which further implies that $|J(K'_{t+1}) - J(K_{t+1})|$ is small. Thus, combining this and (4.6), we obtain the linear convergence of the actor-critic algorithm. See §C.2 for a detailed proof. □

This theorem shows that natural actor-critic algorithm combined with GTD converges linearly to the optimal policy of LQR. Furthermore, the number of policy updates in this theorem matches those obtained by natural policy gradient algorithm [26, 44]. To the best of our knowledge, this result seems to be the first nonasymptotic convergence result for actor-critic algorithms with function approximation, whose existing theory are mostly asymptotic and based on ODE approximation. Furthermore, from the viewpoint of bilevel optimization, Theorem 4.3 offers theoretical guarantees for the actor-critic algorithm as a first-order online method for the bilevel program defined in (3.21), which serves a first attempt of understanding bilevel optimization with possibly nonconvex subproblems.

Furthermore, although we only consider the problem of LQR and analyze the natural actor-critic with GTD for policy evaluation, our theoretical framework can be applied to general reinforcement learning problems with other policy optimization methods for the actor (e.g. vanilla policy gradient [68], trust-region policy optimization [57] and proximal policy optimization [58]) and other policy evaluation methods for the critic such as TD(0) [65], least-square temporal-difference (LSTD) [13], and Retrace [47]. In particular, suppose the critic adopts compatible features [68] for policy evaluation using nonconvex optimization techniques, it can be shown that vanilla policy gradient converges to a local minimum of the expected total return with a sublinear rate [75]. Moreover, by leveraging the geometry of the expected total return as a functional of the policy, recently, [1, 39, 71, 59] prove that natural policy gradient [34], TRPO and PPO are all able to find the globally optimal policy. Using similar approaches, we can establish the convergence and global optimality of actor-critic methods.

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
