[Supplementary Material 1]

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

# A   Algorithms

In this section, we present the details of the actor-critic algorithm.

---

**Algorithm 1** Natural Actor-Critic Algorithm for Linear Quadratic Regulator

---

    **Input:** Initial policy $\pi_{K_0}$ such that $\rho(A - BK_0) < 1$, stepsizes $\gamma$ for policy update, and a policy evaluation algorithm.

    **Initialization:** Set the current policy $\pi_K$ by letting $K \leftarrow K_0$.

    **while** updating current policy **do**

        **Critic step.** Estimate $\Theta_K$ in (3.7) via a policy evaluation algorithm, e.g., the on-policy GTD algorithm (Algorithm 2), which returns an estimator $\widehat{\Theta}$ of $\Theta_K$.

        **Actor step.** Update the policy parameter by $K \leftarrow K - \gamma \cdot (\widehat{\Theta}^{22} K - \widehat{\Theta}^{21})$.

    **end while**

    **Output:** The final policy $\pi_K$, matrix $\widehat{\Theta}$ that estimates $\Theta_K$, and $\widehat{J}$ that approximates $J(K)$.

---

---

**Algorithm 2** On-Policy Gradient-Based Temporal-Difference Algorithm for Policy Evaluation

---

    **Input:** Policy $\pi_K$, number of iterations $T$, and stepsizes $\{\alpha_t\}_{t \in [T]}$.

    **Output:** Estimator $\widehat{\Theta}$ of $\Theta_K$ in (3.7).

    Initialize the primal and dual variables by $\vartheta_0 \in \mathcal{X}_\Theta$ and $\omega_0 \in \mathcal{X}_\Omega$, respectively.

    Sample the initial state $x_0 \in \mathbb{R}^d$ from the stationary distribution $\rho_K$. Take action $u_0 \sim \pi_K(\cdot \,|\, x_0)$ and obtain the reward $c_0$ and the next state $x_1$.

    **for** $t = 1, 2, \ldots, T$ **do**

        Take action $u_t$ according to policy $\pi_K$, observe the reward $c_t$ and the next state $x_{t+1}$.

        Compute the TD-error $\delta_t = \vartheta_{t-1}^1 - c_{t-1} + [\phi(x_{t-1}, u_{t-1}) - \phi(x_t, u_t)]^\top \vartheta_{t-1}^2$.

        Update $\vartheta^1$ by $\vartheta_t^1 = \vartheta_{t-1}^1 - \alpha_t \cdot [\omega_{t-1}^1 + \phi(x_{t-1}, u_{t-1})^\top \omega_{t-1}^2]$.

        Update $\vartheta^2$ by $\vartheta_t^2 = \vartheta_{t-1}^2 - \alpha_t \cdot [\phi(x_{t-1}, u_{t-1}) - \phi(x_t, u_t)] \cdot \phi(x_{t-1}, u_{t-1})^\top \omega_{t-1}^2$.

        Update $\omega^1$ by $\omega_t^1 = (1 - \alpha_t) \cdot \omega_t^1 + \alpha_t \cdot (\vartheta_{t-1}^1 - c_{t-1})$.

        Update $\omega^2$ by $\omega_t^2 = (1 - \alpha_t) \cdot \omega_t^2 + \alpha_t \cdot \delta_t \cdot \phi(x_{t-1})$.

        Project $\vartheta_t$ and $\omega_t$ to $\mathcal{X}_\Theta$ and $\mathcal{X}_\Omega$, respectively.

    **end for**

    Define $\widehat{\vartheta} = (\widehat{\vartheta}^1, \widehat{\vartheta}^2) = (\sum_{t=1}^T \alpha_t \cdot \vartheta_t)/(\sum_{t=1}^T \alpha_t)$ and $\widehat{\omega} = (\sum_{t=1}^T \alpha_t \cdot \omega_t)/(\sum_{t=1}^T \alpha_t)$.

    Return $\widehat{\vartheta}^1$ and $\widehat{\Theta} = \mathrm{smat}(\widehat{\vartheta}^2)$ as the estimators of $J(K)$ and $\Theta_K$, respectively.

---

# B   Extension to the Off-Policy Setting

Recall that in our natural actor-critic algorithm, the critic can apply any policy evaluation algorithm to estimate $\widehat{\Theta}_K$. When using an off-policy method, we obtain an off-policy actor-critic algorithm. In this section, we extend Algorithm 2 to the off-policy setting using importance sampling. Specifically, let $\pi_b$ be the behavior policy and suppose it induces a stationary distribution $\nu_b$ over the state space $\mathbb{R}^d$. Moreover, let $\pi_K$ be the policy of interest and let $\tau_K(x, u) = \pi_K(u \,|\, x)/\pi_b(u \,|\, x)$ be the importance sampling ratio. Then, the Bellman equation in (3.14) can be written as

$$\langle \phi(x, u), \theta_K^* \rangle = c(x, u) - J(K) + \big\langle \mathbb{E}[\phi(x', u') \cdot \tau_K(x', u') \,|\, x, u], \theta_K^* \big\rangle, \tag{B.1}$$

where $(x, u) \in \mathbb{R}^{d+k}$, $x'$ is the next state given $(x, u)$, and $u' \sim \pi_b(\cdot \,|\, x)$. In the following, we denote by $\mathbb{E}_{(x,u)}$ the expectation with respect to $x \sim \nu_b$ and $u \sim \pi_b(\cdot \,|\, x)$. Similar to $\Xi_K$ and $b_K$ defined in (3.15), for the off-policy setting, we define

$$\overline{\Xi}_K = \mathbb{E}_{(x,u)}\big\{\phi(x, u)\big[\phi(x, u) - \tau_K(x', u') \cdot \phi(x', u')\big]^\top\big\}, \qquad \overline{b}_K = \mathbb{E}_{(x,u)}\big[c(x, u)\phi(x, u)\big],$$

$$\overline{h}_K = \mathbb{E}_{(x,u)}[\phi(x, u) - \tau_K(x', u') \cdot \phi(x', u')], \qquad \overline{g}_K = \mathbb{E}_{(x,u)}[\phi(x, u)], \qquad \overline{a}_K = \mathbb{E}_{(x,u)}[c(x, u)].$$

Based on (B.1) and direct computation, it can be shown that $\vartheta_K^* = (J(K), \mathrm{svec}(\Theta_K^\top)^\top)^\top$ is the solution to linear equation

$$\Lambda_K \vartheta = \lambda_K, \qquad \Lambda_K = \begin{pmatrix} 1 & \overline{h}_K^\top \\ \overline{g}_K & \overline{\Xi}_K \end{pmatrix}, \qquad \vartheta = \begin{pmatrix} \vartheta^1 \\ \vartheta^2 \end{pmatrix}, \qquad \lambda_K = \begin{pmatrix} \overline{a}_K \\ \overline{b}_K \end{pmatrix}. \tag{B.2}$$

Similar to the derivations in §3.2, we propose to estimate $\vartheta_K^*$ by solving a minimax optimization problem:

$$\min_{\vartheta \in \mathcal{X}_\Theta} \max_{\omega \in \mathcal{X}_\Omega} \overline{F}(\vartheta, \omega) = \omega^\top [\Lambda_K \vartheta - \lambda_K] - 1/2 \cdot \|\omega\|_2^2. \tag{B.3}$$

Here, similar to Assumption (4.1), we let $\mathcal{X}_\Theta$ and $\mathcal{X}_\Omega$ be Euclidean balls given by $\mathcal{X}_\Theta$ and $\mathcal{X}_\Omega$ in (3.18) be defined as $\mathcal{X}_\Theta = \{\vartheta \colon \|\vartheta^1\|_2 \leq \widetilde{R}_\Theta\}$ and $\mathcal{X}_\Omega = \{\omega \colon \|\omega\|_2 \leq \widetilde{R}_\Omega\}$, where $\widetilde{R}_\Theta$ and $\widetilde{R}_\Omega$ are chosen appropriately. Notice that both $\overline{F}(\vartheta, \omega)$ and its gradient can be estimated unbiasedly using transitions sampled from the behavior policy. Solving (B.3) using stochastic gradient method, we obtain the off-policy GTD algorithm for the ergodic setting. See Algorithm 3 for the details. Combining this policy evaluation method with Algorithm 1, we establish the off-policy on-line natural actor-critic algorithm.

---

**Algorithm 3** Off-Policy Gradient-Based Temporal-Difference Algorithm for Policy Evaluation

---

**Input:** Policy $\pi_K$, number of iterations $T$, and stepsizes $\{\alpha_t\}_{t \in [T]}$, the behavior policy $\pi_b$ and its stationary distribution $\nu_b$.

**Output:** Estimators $\widehat{J}$ and $\widehat{\Theta}$ of $J(K)$ in (3.8) and $\Theta_K$ in (3.7), respectively.

Initialize the primal and dual variables by $\vartheta_0 \in \mathcal{X}_\Theta$ and $\omega_0 \in \mathcal{X}_\Omega$, respectively.

Sample the initial state $x_0 \in \mathbb{R}^d$ from the stationary distribution $\nu_b$. Take action $u_0 \sim \pi_b(\cdot \mid x_0)$ and obtain the reward $c_0$ and the next state $x_1$.

**for** $t = 1, 2, \ldots, T$ **do**

    Take action $u_t$ according to policy $\pi_K$, observe the reward $c_t$ and the next state $x_{t+1}$.

    Compute the TD-error $\delta_t = \vartheta_{t-1}^1 - c_{t-1} + [\phi(x_{t-1}, u_{t-1}) - \tau_K(x_t, u_t) \cdot \phi(x_t, u_t)]^\top \vartheta_{t-1}^2$.

    Update the primal variable $\vartheta$ by

$$\vartheta_t^1 = \vartheta_{t-1}^1 - \alpha_t \cdot [\omega_{t-1}^1 + \phi(x_{t-1}, u_{t-1})^\top \omega_{t-1}^2],$$
$$\vartheta_t^2 = \vartheta_{t-1}^2 - \alpha_t \cdot [\phi(x_{t-1}, u_{t-1}) - \tau_K(x_t, u_t) \cdot \phi(x_t, u_t)] \cdot [\phi(x_{t-1}, u_{t-1})^\top \omega_{t-1}^2 + \omega_{t-1}^1].$$

    Update the dual variable $\omega$ by

$$\omega_t^1 = (1 - \alpha_t) \cdot \omega_t^1 + \alpha_t \cdot \{\vartheta_{t-1}^1 + [\phi(x_{t-1}, u_{t-1}) - \tau_K(x_t, u_t) \cdot \phi(x_t, u_t)]^\top \vartheta_{t-1}^2 - c_{t-1}\},$$
$$\omega_t^2 = (1 - \alpha_t) \cdot \omega_t^2 + \alpha_t \cdot \delta_t \cdot \phi(x_{t-1}).$$

    Project $\vartheta_t$ and $\omega_t$ to $\mathcal{X}_\Theta$ and $\mathcal{X}_\Omega$, respectively.

**end for**

Define $\widehat{\vartheta} = (\widehat{\vartheta}^1, \widehat{\vartheta}^2) = (\sum_{t=1}^T \alpha_t \cdot \vartheta_t)/(\sum_{t=1}^T \alpha_t)$ and $\widehat{\omega} = (\sum_{t=1}^T \alpha_t \cdot \omega_t)/(\sum_{t=1}^T \alpha_t)$.

Return $\widehat{\vartheta}^1$ and $\widehat{\Theta} = \mathrm{smat}(\widehat{\vartheta}^2)$ as the estimators of $J(K)$ and $\Theta_K$, respectively.

---

Similar to Theorem 4.2, we have the following theorem that shows that Algorithm 3 converges at a sublinear rate to the desired solution $\vartheta_K^*$.

**Theorem B.1** (off-policy GTD). Let $\widehat{\vartheta}^1$ and $\widehat{\Theta}$ be the output of Algorithm 3 based on $T$ iterations. We set the stepsize to be $\alpha_t = \alpha/\sqrt{t}$ with $\alpha > 0$ being a constant. We assume that $\Lambda_K$ in (B.2) is invertible and that its minimum singular value is lower bounded by a constant $\kappa_k^* > 0$. Moreover, we assume that the Markov chain induced by the behavioral policy $\pi_b$ is geometrically $\beta$-mixing with parameter $\rho \in (0, 1)$. Let $\nu_b$ be the stationary distribution of this induced Markov chain. We assume that, for $(x, u) \sim \nu_b$, both $\phi(x, u)$ and $\tau_K(x, u)$ are sub-exponential random variables. Then, when the number of iterations $T$ is sufficiently large, with probability at least $1 - T^{-4}$, we have

$$\|\widehat{\Theta} - \Theta_K\|_{\mathrm{fro}}^2 \leq \frac{\Upsilon[\widetilde{R}_\Theta, \widetilde{R}_\Omega, J(K_0), \|K\|_{\mathrm{fro}}, \sigma_{\min}^{-1}(Q)]}{\kappa_K^{*2} \cdot (1 - \rho)} \cdot \frac{\log^6 T}{\sqrt{T}},$$

where $\Upsilon[\widetilde{R}_\Theta, \widetilde{R}_\Omega, J(K_0), \|K\|_{\mathrm{fro}}, \sigma_{\min}^{-1}(Q)]$ is a polynomial of $\widetilde{R}_\Omega$, $\widetilde{R}_\Omega$, $J(K_0)$, $\|K\|_{\mathrm{fro}}$, and $1/\sigma_{\min}(Q)$.

*Proof.* The proof of this theorem is parallel to that of Theorem 4.2, thus here we only sketch the proof for brevity.

The proof can be completed in three steps. In the first step, we show that $(\vartheta, \omega) = (\vartheta_K^*, 0)$ is the saddle point of the optimization problem in (B.3). To simplify the notation, we define a vector-valued function $G(x, u, x', u'; \vartheta)$ by

$$G^1(x, u, x', u'; \vartheta) = \vartheta^1 - c(x, u) + \langle \phi(x, u) - \tau_K(x', u') \cdot \phi(x', u'), \vartheta^2 \rangle,$$

$$G^2(x, u, x', u'; \vartheta) = \vartheta^1 \cdot \phi(x, u) + \left\{ \left[ \phi(x, u) - \phi(x', u') \cdot \tau_K(x', u') \right]^\top \vartheta^2 - c(x, u) \right\} \cdot \phi(x, u).$$

(B.4)

By definition, for all $(\vartheta, \omega)$, $\overline{F}(\vartheta, \omega)$ in (B.3) can be equivalently written as

$$\overline{F}(\vartheta, \omega) = \langle \mathbb{E}_{(x, u, x', u')}[G(x, u, x', u'; \vartheta)], \omega \rangle - 1/2 \cdot \|\omega\|_2^2.$$

(B.5)

Thus, for any $\vartheta$, the solution to the unconstrained maximization problem $\max_\omega F(\theta, \omega)$ is

$$w(\vartheta) = \mathbb{E}_{(x, u, x', u')}[G(x, u, x', u'; \vartheta)].$$

Recall that $c(x, u) = \langle \phi(x, u), \text{svec}[\text{diag}(Q, R)] \rangle$. Since both $\phi(x, u)$ and $\tau_K(x, u) \cdot \phi(x, u)$ are sub-exponential random vectors, when $\widetilde{R}_\Theta$ and $\widetilde{R}_\Omega$ are chosen properly, we can show that $\vartheta_K^* \in \mathcal{X}_\Theta$ and that $w(\vartheta) \in \mathcal{X}_\Omega$ for all $\vartheta \in \mathcal{X}_\Theta$. Notice that $w(\vartheta_K^*) = 0$. Thus, $(\vartheta_K^*, 0)$ is the solution to the minimax optimization problem in (B.3).

Then, in the second step, we relate the estimation error $\|\widehat{\Theta} - \Theta_K\|_{\text{fro}}^2$ to the primal-dual gap

$$\texttt{Gap}(\widehat{\vartheta}, \widehat{\omega}) = \max_{\omega \in \mathcal{X}_\Omega} \overline{F}(\widehat{\vartheta}, \omega) - \min_{\vartheta \in \mathcal{X}_\Theta} \overline{F}(\vartheta, \widehat{\omega}).$$

(B.6)

Similar to the derivations in (C.14)–(C.17), since the minimum singular value of $\Lambda_K$ is lower bounded by $\kappa_K^*$, it holds that

$$|\widehat{\vartheta}^1 - J(K)|^2 + \|\widehat{\Theta} - \Theta_K\|_{\text{fro}}^2 \leq {\kappa_K^*}^{-2} \cdot \texttt{Gap}(\widehat{\vartheta}, \widehat{\omega}).$$

(B.7)

Thus, it suffices to bound the primal-dual gap in (B.6), which is achieved in the last step.

Specifically, we would like to utilize Theorem C.4 obtained from [72]. Since this theorem requires bounded iterates and Lipschitz gradient, similar to the third step in §C.1, we truncate the feature vector $\phi(x, u)$. In particular, we define

$$\mathcal{E} = \bigcap_{0 \leq t \leq T} \left\{ \|\phi(x_t, u_t)\|_2^2 \leq C_K \cdot \log T, \|\tau_K(x_t, u_t) \cdot \phi(x_t, u_t)\|_2^2 \leq C_K \cdot \log T \right\},$$

(B.8)

where $C_b$ is a constant specified by the stationary distribution $\nu_b$. Since both is $\phi(x, u)$ and $\phi(x, u) \cdot \tau_K(x, u)$ are sub-exponential random vector when $(x, u) \sim \nu_b$, it can be shown that $\mathcal{E}$ holds with probability at least $1 - 2T^{-5}$. Then we define truncated random vectors

$$\widetilde{\phi}(x, u) = \phi(x, u) \cdot \mathbb{1}_{\mathcal{E}}, \qquad \widetilde{\varphi}_K(x, u) = \phi(x, u) \cdot \tau_K(x, u) \cdot \mathbb{1}_{\mathcal{E}}$$

and the truncated minimax optimization problem,

$$\min_{\vartheta \in \mathcal{X}_\Theta} \max_{\omega \in \mathcal{X}_\Omega} \widetilde{F}(\vartheta, \omega) = \langle \mathbb{E}_{(x, u, x', u')}[\widetilde{G}(x, u, x', u'; \vartheta)], \omega \rangle - 1/2 \cdot \|\omega\|_2^2,$$

(B.9)

where we define $\widetilde{G}(x, u, x', u'; \vartheta)$ by

$$\widetilde{G}^1(x, u, x', u'; \vartheta) = \vartheta^1 - \widetilde{c}(x, u) + \langle \widetilde{\phi}(x, u) - \widetilde{\varphi}_K(x', u'), \vartheta^2 \rangle,$$

$$G^2(x, u, x', u'; \vartheta) = \vartheta^1 \cdot \widetilde{\phi}(x, u) + \left\{ \left[ \widetilde{\phi}(x, u) - \widetilde{\varphi}_K(x', u') \right]^\top \vartheta^2 - \widetilde{c}(x, u) \right\} \cdot \widetilde{\phi}(x, u).$$

(B.10)

Here we let $\widetilde{c}(x, u) = \langle \widetilde{\phi}(x, u), \text{svec}[\text{diag}(Q, R)] \rangle$ in (B.10). Similar to the derivations from (C.25) to (C.31), we can show that $\sup_{\vartheta, \omega} |\overline{F}(\vartheta, \omega) - \widetilde{F}(\vartheta, \omega)| \leq 1/T$, which implies that

$$\left| \texttt{Gap}(\widehat{\vartheta}, \widehat{\omega}) - \left[ \max_{\omega \in \mathcal{X}_\Omega} \widetilde{F}(\widehat{\vartheta}, \omega) - \min_{\vartheta \in \mathcal{X}_\Theta} \widetilde{F}(\vartheta, \widehat{\omega}) \right] \right| \leq 2/T.$$

(B.11)

Since $\widetilde{F}$ defined in (B.9) have Lipschitz gradients, by Theorem C.4, we have

$$\max_{\omega \in \mathcal{X}_\Omega} \widetilde{F}(\widehat{\vartheta}, \omega) - \min_{\vartheta \in \mathcal{X}_\Theta} \widetilde{F}(\vartheta, \widehat{\omega}) \leq \frac{C_K \cdot \log^6 T}{(1 - \rho) \cdot \sqrt{T}}$$

(B.12)

with probability at least $1 - T^{-5}$, where $C_K$ is a constant that depends polynomially on $\widetilde{R}_\Theta$, $\widetilde{R}_\Omega$, $J(K_0)$, $\|K\|_{\text{fro}}$, and $1/\sigma_{\min}(Q)$. Finally, combining (B.7), (B.11), and (B.12), we conclude the proof of this theorem. $\qquad \square$

# C   Proofs of the Main Results

In this section, we provide the proofs of the main results, namely, Theorems 4.2 and 4.3, which are proved in §C.1 and §C.2, respectively. The proofs of the supporting results are deferred to the appendix.

## C.1   Proof of Theorem 4.2

*Proof.* Our proof can be decomposed into three steps. In the first step, we show that, with $\mathcal{X}_\Theta$ and $\mathcal{X}_\Omega$ given in (4.1) and (4.2), $(\vartheta, \omega) = (\vartheta_K^*, 0)$ is the solution to the minimax optimization problem in (3.18). Then, in the second step, we show that the primal-dual gap of this optimization problem yields an upper bound for the estimation error $\|\widehat{\Theta} - \Theta_K\|_{\mathrm{fro}}^2$, where $\widehat{\Theta} = \mathrm{smat}(\widehat{\vartheta}^2)$ is the estimator of $\Theta_K$ returned by the GTD algorithm. Finally, in the last step, we study the performance of such a minimax optimization problem, which enables us to establish the error of policy evaluation.

**Step 1.** In the first step, we show that $(\vartheta, \omega) = (\vartheta_K^*, 0)$ is the saddle point of the optimization problem in (3.18). To simplify the notation, we define a vector-valued function $G(x, u, x', u'; \vartheta)$ by

$$
\begin{aligned}
G^1(x, u, x', u'; \vartheta) &= \vartheta^1 - c(x, u), \\
G^2(x, u, x', u'; \vartheta) &= \vartheta^1 \cdot \phi(x, u) + \left\{ \left[ \phi(x, u) - \phi(x', u') \right]^\top \vartheta^2 - c(x, u) \right\} \cdot \phi(x, u).
\end{aligned}
\tag{C.1}
$$

By definition, $G(x, u, x', u'; \vartheta)$ is of the same shape as $\vartheta$ and $\omega$. Moreover, for all $(\vartheta, \omega)$, $F(\vartheta, \omega)$ in (3.18) can be equivalently written as

$$
F(\vartheta, \omega) = \left\langle \mathbb{E}_{(x,u,x',u')}[G(x, u, x', u'; \vartheta)], \omega \right\rangle - 1/2 \cdot \|\omega\|_2^2.
\tag{C.2}
$$

Thus, for any $\vartheta$, the solution to the unconstrained maximization problem $\max_\omega F(\theta, \omega)$ is

$$
w(\vartheta) = \mathbb{E}_{(x,u,x',u')}[G(x, u, x', u'; \vartheta)].
\tag{C.3}
$$

In the following, we show that $\vartheta_K^* \in \mathcal{X}_\Theta$. Moreover, we also prove that, for any $\vartheta \in \mathcal{X}_\Theta$, $w(\vartheta)$ in (C.3) belongs to $\mathcal{X}_\Omega$, where $\mathcal{X}_\Theta$ and $\mathcal{X}_\Omega$ are defined in (4.1) and (4.2), respectively. Since $w(\vartheta_K^*) = 0$, it holds that $(\vartheta_K^*, 0)$ is the solution to the minimax optimization problem in (3.18).

Recall that we assume $J(K) \le J(K_0)$, where $K_0$ is the initial policy that is stable. Thus, $J(K_0)$ is finite. By the definition of $\vartheta_K^*$, to show $\vartheta_K^* \in \mathcal{X}_\Theta$, it suffices to bound $\|\Theta_K\|_{\mathrm{fro}}$. By the definition of $\Theta_K$ in (3.7), we have

$$
\Theta_K = \begin{pmatrix} Q + A^\top P_K A & A^\top P_K B \\ B^\top P_K A & R + B^\top P_K B \end{pmatrix} = \begin{pmatrix} Q & \\ & R \end{pmatrix} + \begin{pmatrix} A^\top \\ B^\top \end{pmatrix} P_K \begin{pmatrix} A & B \end{pmatrix},
$$

which implies that

$$
\|\Theta_K\|_{\mathrm{fro}} \le (\|Q\|_{\mathrm{fro}} + \|R\|_{\mathrm{fro}}) + (\|A\|_{\mathrm{fro}}^2 + \|B\|_{\mathrm{fro}}^2) \cdot \|P_K\|_{\mathrm{fro}}.
\tag{C.4}
$$

Now we apply the following lemma to obtain an upper bound on $\|P_K\|_{\mathrm{fro}}$.

**Lemma C.1.** When $\pi_K$ is a stable policy, we have

$$
\|\Sigma_K\| \le J(K)/\sigma_{\min}(Q), \qquad \|P_K\| \le J(K)/\sigma_{\min}(\Psi),
$$

where $\sigma_{\min}(\cdot)$ denotes the minimal eigenvalue of a matrix.

*Proof.* By (3.8) in Proposition 3.1, we have

$$
\begin{aligned}
J(K) &\ge \mathrm{tr}[(Q + K^\top R K)\Sigma_K] \ge \sigma_{\min}(Q) \cdot \mathrm{tr}(\Sigma_K) \ge \sigma_{\min}(Q) \cdot \|\Sigma_K\|, \\
J(K) &\ge \mathrm{tr}(P_K \Psi_\sigma) \ge \sigma_{\min}(\Psi_\sigma) \cdot \mathrm{tr}(P_K) \ge \|P_K\| \ge J(K)/\sigma_{\min}(\Psi),
\end{aligned}
$$

where we use the fact that $\Psi_\sigma \succeq \Psi$. Therefore, we conclude the proof.   $\square$

Applying Lemma C.1 to (C.4), we have

$$
\|\Theta_K\|_{\mathrm{fro}} \le (\|Q\|_{\mathrm{fro}} + \|R\|_{\mathrm{fro}}) + (\|A\|_{\mathrm{fro}}^2 + \|B\|_{\mathrm{fro}}^2) \cdot \sqrt{d} \cdot J(K)/\sigma_{\min}(\Psi).
\tag{C.5}
$$

Combining (C.5) and the definition of $\widetilde{R}_\Theta$ in (4.3) we conclude that $\vartheta_K^* \in \mathcal{X}_\Theta$.

Furthermore, it remains to show that the vector in (C.3) belongs to $\mathcal{X}_\Omega$ for all $\vartheta \in \mathcal{X}_\Theta$. We consider the two components of $G(x, u, x', u'; \vartheta)$ separately. By (C.1), we have

$$\left| \mathbb{E}_{(x,u,x',u')}[G^1(x, u, x', u'; \vartheta)] \right| = |\vartheta^1 - J(K)| \leq J(K_0), \tag{C.6}$$

where the second inequality follows from the fact that $0 \leq \vartheta^1 \leq J(K_0)$. Moreover, by (C.1), for the second component of $G(x, u, x', u'; \vartheta)$, we have

$$\mathbb{E}_{(x,u,x',u')}[G^2(x, u, x', u'; \vartheta)] = \vartheta^1 \cdot \mathbb{E}_{(x,u)}[\phi(x, u)] + \Xi_K \vartheta^2 - b_K, \tag{C.7}$$

where $\Xi_K$ and $b_K$ are defined in (3.15). By Lemma D.2, we have

$$\|\Xi_K \vartheta^2\|_2 \leq \|\Xi_K\| \cdot \|\vartheta^2\|_2 \leq 4(1 + \|K\|_{\text{fro}}^2)^2 \cdot \|\Sigma_K\|^2 \cdot \widetilde{R}_\Theta. \tag{C.8}$$

Moreover, for any positive definite matrix $\Gamma$, we have

$$b_K^\top \text{svec}(\Gamma) = \mathbb{E}_{(x,u)}\big\{ \langle \phi(x, u), \text{svec}[\text{diag}(Q, R)] \rangle \cdot \langle \phi(x, u), \text{smat}(\Gamma) \rangle \big\}, \tag{C.9}$$

where $\text{diag}(Q, R)$ is the block diagonal matrix constructed by $Q$ and $R$. Note that the joint distribution of $(x, u)$ is the Gaussian distribution $N(0, \widetilde{\Sigma}_K)$, where $\widetilde{\Sigma}_K$ is defined in (D.16). Thus, $b_K^\top \text{svec}(\Gamma)$ can be written as the product of two quadratic forms of Gaussian random variables. Applying Lemma D.3 to (C.9), we obtain that

$$b_K^\top \text{svec}(\Gamma) = 2\langle \widetilde{\Sigma}_K \text{diag}(Q, R) \widetilde{\Sigma}_K, \Gamma \rangle \cdot + \langle \widetilde{\Sigma}_K, \text{diag}(Q, R) \rangle \cdot \langle \widetilde{\Sigma}_K, \Gamma \rangle,$$

which implies that

$$\|b_K\|_2 \leq 3(\|Q\|_{\text{fro}} + \|R\|_{\text{fro}}) \cdot \|\widetilde{\Sigma}_K\|^2. \tag{C.10}$$

In addition, the first term on the right-hand side of (C.7) is bounded by

$$\left\| \vartheta^1 \cdot \mathbb{E}_{(x,u)}[\phi(x, u)] \right\|_2 \leq J(K_0) \cdot \left\| \widetilde{\Sigma}_K \right\|_{\text{fro}}. \tag{C.11}$$

Finally, combining (C.8), (C.10), (C.11), and the upper bounds in (D.17), we have

$$\left\| \mathbb{E}_{(x,u,x',u')}[G^2(x, u, x', u'; \vartheta)] \right\|_2$$
$$\leq 2(d + \|K\|_{\text{fro}}^2) \cdot \|\Sigma_K\| + 4(1 + \|K\|_{\text{fro}}^2)^2 \cdot \|\Sigma_K\|^2 \cdot \widetilde{R}_\Theta$$
$$\quad + 12(\|Q\|_{\text{fro}} + \|R\|_{\text{fro}}) \cdot (d + \|K\|_{\text{fro}}^2)^2 \cdot \|\Sigma_K\|^2$$
$$\leq C \cdot (1 + \|K\|_{\text{fro}}^2)^2 \cdot \widetilde{R}_\Theta \cdot \sigma_{\min}^{-2}(Q) \cdot [J(K_0)]^2, \tag{C.12}$$

where $C > 0$ is an absolute constant.

Hence, combining (4.4), (C.6) and (C.12), we conclude that $w(\vartheta) \in \mathcal{X}_\Omega$ for all $\vartheta \in \mathcal{X}_\Theta$. Therefore, we have shown that $(\vartheta_K^*, 0)$ is the saddle point of the optimization problem in (3.18), which concludes the first step of the proof.

**Step 2.** In the following, we relate the estimation error $\|\widehat{\Theta} - \Theta_K\|_{\text{fro}}^2$ to the performance of the optimization in (3.18). Specifically, we consider the primal-dual gap

$$\text{Gap}(\widehat{\vartheta}, \widehat{\omega}) = \max_{\omega \in \mathcal{X}_\Omega} F(\widehat{\vartheta}, \omega) - \min_{\vartheta \in \mathcal{X}_\Theta} F(\vartheta, \widehat{\omega}), \tag{C.13}$$

which characterizes the closeness between $(\widehat{\vartheta}, \widehat{\phi})$ and the optimal solution $(\vartheta_K^*, 0)$, quantified by the objective value.

Recall that $w(\vartheta)$ defined in (C.3) is the optimal dual variable for each $\theta \in \mathcal{X}_\Theta$. Hence, for any $\omega \in \mathcal{X}_\Omega$, it holds that

$$\min_{\vartheta \in \mathcal{X}_\Theta} F(\vartheta, \omega) \leq \min_{\theta \in \mathcal{X}_\Theta} \max_{\omega \in \mathcal{X}_\Omega} F(\theta, \omega)$$
$$\leq \min_{\vartheta \in \mathcal{X}_\Omega} \big\{ [\vartheta^1 - J(K)]^2 + \|\vartheta^1 \cdot \mathbb{E}_{(x,u)}[\phi(x, u)] + \Xi_K \vartheta^2 - b_K\|_2^2 \big\} = 0. \tag{C.14}$$

Thus, for $\widehat{\vartheta}$ returned by the GTD algorithm, we have

$$\big\{ [\widehat{\vartheta}^1 - J(K)]^2 + \|\widehat{\vartheta}^1 \cdot \mathbb{E}_{(x,u)}[\phi(x, u)] + \Xi_K \widehat{\vartheta}^2 - b_K\|_2^2 \big\} = \max_{\omega \in \mathcal{X}_\Omega} F(\widehat{\vartheta}, \omega)$$
$$= \max_{\omega \in \mathcal{X}_\Omega} F(\widehat{\vartheta}, \omega) - \min_{\vartheta \in \mathcal{X}_\Theta} F(\vartheta, \widehat{\omega}) + \min_{\vartheta \in \mathcal{X}_\Theta} F(\vartheta, \widehat{\omega}) \leq \text{Gap}(\widehat{\vartheta}, \widehat{\omega}), \tag{C.15}$$

where the last inequality follows from (C.14).

Furthermore, by direct computation, we can bound the left-hand side of (C.15) via

$$\left\| \begin{pmatrix} 1 & 0 \\ \mathbb{E}_{(x,u)}[\phi(x,u)] & \Xi_K \end{pmatrix} (\widehat{\vartheta} - \vartheta_K^*) \right\|_2^2$$
$$\geq {\kappa_K^*}^2 \cdot \|\widehat{\vartheta} - \vartheta_K^*\|_2^2 = \kappa_K^* \cdot \big[\|\widehat{\Theta} - \Theta_K\|_{\mathrm{fro}}^2 + |\widehat{\vartheta}^1 - J(K)|^2\big], \tag{C.16}$$

where we utilize the fact that $\vartheta_K^*$ is the solution to the linear equation in (3.16) and $\kappa_K^*$ is specified in Lemma 3.2. Therefore, combining (C.15) and (C.16), we have

$$|\widehat{\vartheta}^1 - J(K)|^2 + \|\widehat{\Theta} - \Theta_K\|_{\mathrm{fro}}^2 \leq {\kappa_K^*}^{-2} \cdot \mathtt{Gap}(\widehat{\vartheta}, \widehat{\omega}), \tag{C.17}$$

which establishes the connection between $\|\widehat{\Theta} - \Theta_K\|_{\mathrm{fro}}^2$ and the primal-dual gap in (C.13).

**Step 3.** In the last step, we construct an upper bound for the primal-dual gap. By (C.17), this yields an upper bound for the error of parameter estimation.

Note that the distribution of the state-action pair $(x,u)$ have unbounded support. We first construct an event such that $\{\phi(x_t, u_t)\}_{t=0}^T$ are bounded conditioning on this event. To this end, we establish an upper bound for tail probability of the $\|\phi(x,u)\|_2$ using the Hansen-Wright inequality stated as follows.

**Lemma C.2** (Hansen-Wright inequality). For any integer $m > 0$, let $A$ be a matrix in $\mathbb{R}^{m \times m}$ and let $\eta \sim N(0, I_m)$ be the standard Gaussian random variable in $\mathbb{R}^m$. Then, there exists an absolute constant $C > 0$ such that, for any $t \geq 0$, we have

$$\mathbb{P}\big[\big|\eta^\top A \eta - \mathbb{E}(\eta^\top A \eta)\big| > t\big] \leq 2 \cdot \exp\big[-C \cdot \min(t^2 \cdot \|A\|_{\mathrm{fro}}^{-2}, \; t \cdot \|A\|^{-1})\big]$$

*Proof.* See [55] for a detailed proof. □

Applying Lemma C.2 to $(x, u) \sim N(0, \widetilde{\Sigma}_K)$ with $\widetilde{\Sigma}_K$ defined in (D.16), we obtain

$$\mathbb{P}\big[\big|\|x\|_2^2 + \|u\|_2^2 - \mathrm{tr}\big(\widetilde{\Sigma}_K\big)\big| > t\big] \leq 2 \cdot \exp\big[-C \cdot \min\big(t^2 \cdot \big\|\widetilde{\Sigma}_K\big\|_{\mathrm{fro}}^{-2}, \; t \cdot \big\|\widetilde{\Sigma}_K\big\|^{-1}\big)\big]. \tag{C.18}$$

Setting $t = C_1 \cdot \log T \cdot \big\|\widetilde{\Sigma}_K\big\|$ in (C.18) with constant $C_1$ sufficiently large, it holds that

$$t^2 \cdot \big\|\widetilde{\Sigma}_K\big\|_{\mathrm{fro}}^{-2} = \big\|\widetilde{\Sigma}_K\big\|_{\mathrm{fro}}^{-2} \cdot C_1^2 \cdot \log^2 T \cdot \big\|\widetilde{\Sigma}_K\big\|^2 \geq C_1^2 \cdot (d+k)^{-1} \cdot \log^2 T \geq t \cdot \big\|\widetilde{\Sigma}_K\big\|^{-1}, \tag{C.19}$$

where the first inequality follows from the relation between the operator and Frobenius norms, and the second inequality holds when $\log T \geq C_1^{-1} \cdot (d+k)$. For ease of presentation, for any $t \in \{0, 1, \dots, T\}$, we define

$$\mathcal{E}_t = \Big\{ \big|\|x_t\|_2^2 + \|u_t\|_2^2 - \mathrm{tr}\big(\widetilde{\Sigma}_K\big)\big| \leq C_1 \cdot \log T \cdot \big\|\widetilde{\Sigma}_K\big\| \Big\}, \tag{C.20}$$

and write $\mathcal{E} = \bigcap_{0 \leq t \leq T} \mathcal{E}_t$. Combining (C.18) and (C.19), we obtain that $\mathcal{E}_t$ holds with probability at least $1 - T^{-6}$. Thus, by taking a union bound for $\{(x_t, u_t)\}_{t=0}^T$, we have $\mathbb{P}(\mathcal{E}) \geq 1 - 2T^{-5}$. Moreover, combining (C.20) and (D.17) further implies that, on event $\mathcal{E}$, we have

$$\max_{0 \leq t \leq T} \big\{\|x_t\|_2^2 + \|u_t\|_2^2\big\} \leq C_1 \cdot \log T \cdot \big\|\widetilde{\Sigma}_K\big\| + \mathrm{tr}\big(\widetilde{\Sigma}_K\big) \leq \big(C_1 \cdot \log T + d + k\big) \cdot \big\|\widetilde{\Sigma}_K\big\|$$
$$\leq 2C_1 \cdot \log T \cdot \big\|\widetilde{\Sigma}_K\big\| \leq 2C_1 \cdot \log T \cdot \big[\sigma^2 + (1 + \|K\|_{\mathrm{fro}}^2) \cdot \|\Sigma_K\|\big]. \tag{C.21}$$

In the sequel, we study the stochastic optimization problem in (3.18) with the restriction that $\mathcal{E}$ holds. Specifically, for any state-action pair $(x, u)$, we define the truncated feature function as

$$\widetilde{\phi}(x, u) = \phi(x, u) \cdot \mathbb{1}\Big\{\big|\|\phi(x,u)\|_2^2 - \mathrm{tr}\big(\widetilde{\Sigma}_K\big)\big| \leq C_1 \cdot \log T \cdot \big\|\widetilde{\Sigma}_K\big\|\Big\}. \tag{C.22}$$

By this definition, for any $t \in \{0, \dots, t\}$, we have $\widetilde{\phi}(x_t, u_t) = \phi(x_t, u_t) \cdot \mathbb{1}_{\mathcal{E}_t}$. Now we replace $\phi(x, u)$ by $\widetilde{\phi}(x, u)$ in (3.18) and consider the following minimiax optimization problem:

$$\min_{\vartheta \in \mathcal{X}_\Theta} \max_{\omega \in \mathcal{X}_\Omega} \widetilde{F}(\vartheta, \omega) = \big\langle \mathbb{E}_{(x,u,x',u')}\big[\widetilde{G}(x, u, x', u'; \vartheta)\big], \omega \big\rangle - 1/2 \cdot \|\omega\|_2^2, \tag{C.23}$$

where, similar to $G(x, u, x', u'; \vartheta)$ in (C.1), we define $\widetilde{G}(x, u, x', u'; \vartheta)$ by

$$\widetilde{G}^1(x, u, x', u'; \vartheta) = \vartheta^1 - \widetilde{c}(x, u),$$

$$\widetilde{G}^2(x, u, x', u'; \vartheta) = \vartheta^1 \cdot \widetilde{\phi}(x, u) + \left\{ \left[ \widetilde{\phi}(x, u) - \widetilde{\phi}(x', u') \right]^\top \vartheta^2 - \widetilde{c}(x, u) \right\} \cdot \widetilde{\phi}(x, u). \tag{C.24}$$

Here we denote $\widetilde{c}(x, u) = \langle \widetilde{\phi}(x, u), \text{svec}[\text{diag}(Q, R)] \rangle$ in (C.24) to simplify the notation.

We remark that, when $\mathcal{E}$ is true, $(\widehat{\vartheta}, \widehat{\omega})$ is also the solution returned by the gradient-based algorithm for the minimax optimization problem in (C.23). As a result, when $\mathcal{E}$ holds, the primal-dual gap of (C.23) is equal to $\max_{\omega \in \mathcal{X}_\Omega} \widetilde{F}(\widehat{\vartheta}, \omega) - \min_{\vartheta \in \mathcal{X}_\Theta} \widetilde{F}(\vartheta, \widehat{\omega})$.

In the following, we characterize the difference between the objective functions in (3.18) and (C.23). For any $(\vartheta, \omega) \in \mathcal{X}_\Theta \times \mathcal{X}_\Omega$, by (C.2) and (C.23) we have

$$\left| F(\vartheta, \omega) - \widetilde{F}(\vartheta, \omega) \right| = \left| \langle \mathbb{E}_{(x,u,x',u')} \left[ G(x, u, x', u'; \vartheta) - \widetilde{G}(x, u, x', u'; \vartheta) \right], \omega \rangle \right|$$

$$\leq \left| \mathbb{E}_{(x,u,x',u')} \left[ G^1(x, u, x', u'; \vartheta) - \widetilde{G}^1(x, u, x', u'; \vartheta) \right] \right| \cdot J(K_0)$$

$$+ \left\| \mathbb{E}_{(x,u,x',u')} \left[ G^2(x, u, x', u'; \vartheta) - \widetilde{G}^2(x, u, x', u'; \vartheta) \right] \right\|_2 \cdot \widetilde{R}_\Omega. \tag{C.25}$$

By the definitions of $G(x, u, x', u'; \vartheta)$ and $\widetilde{G}(x, u, x', u'; \vartheta)$ in (C.1) and (C.24), we have

$$G^1(x, u, x', u'; \vartheta) - \widetilde{G}^1(x, u, x', u'; \vartheta) = c(x, u) \cdot \mathbb{1}_{\mathcal{A}^c} \tag{C.26}$$

$$G^2(x, u, x', u'; \vartheta) - \widetilde{G}^2(x, u, x', u'; \vartheta) = G^2(x, u, x', u'; \vartheta) \cdot \mathbb{1}_{\mathcal{A}^c} + \phi(x', u')^\top \vartheta^2 \cdot \phi(x, u) \cdot \mathbb{1}_{\mathcal{A}} \cdot \mathbb{1}_{\mathcal{B}^c},$$

where we denote $\{ \left| \|\phi(x, u)\|_2^2 - \text{tr}(\widetilde{\Sigma}_K) \right| \leq C_1 \cdot \log T \cdot \|\widetilde{\Sigma}_K\| \}$ and $\{ \left| \|\phi(x, u)\|_2^2 - \text{tr}(\widetilde{\Sigma}_K) \right| \leq C_1 \cdot \log T \cdot \|\widetilde{\Sigma}_K\| \}$ by $\mathcal{A}$ and $\mathcal{B}$, respectively, and $\mathcal{A}^c, \mathcal{B}^c$ are the complement sets of $\mathcal{A}$ and $\mathcal{B}$.

For the first term on the right-hand side of (C.25), Cauchy-Schwarz inequality implies that

$$\left| \mathbb{E}_{(x,u,x',u')} \left[ G^1(x, u, x', u'; \vartheta) - \widetilde{G}^1(x, u, x', u'; \vartheta) \right] \right| \leq \sqrt{\mathbb{P}(\mathcal{A}^c)} \cdot \sqrt{\mathbb{E}[c^2(x, u)]}. \tag{C.27}$$

Since $c(x, u)$ is a quadratic form of a Gaussian random variable, by Lemma D.3, we have

$$\mathbb{E}[c^2(x, u)] = 2 \text{tr} \left[ \widetilde{\Sigma}_K \text{diag}(Q, R) \widetilde{\Sigma}_K \text{diag}(Q, R) \right] + \left\{ \text{tr} \left[ \widetilde{\Sigma}_K \text{diag}(Q, R) \right] \right\}^2$$

$$\leq 3(\|Q\|_{\text{fro}} + \|R\|_{\text{fro}})^2 \cdot \|\widetilde{\Sigma}_K\|_{\text{fro}}^2 \leq 3(\|Q\|_{\text{fro}} + \|R\|_{\text{fro}})^2 \cdot \left[ \sigma^2 \cdot k + (d + \|K\|_{\text{fro}}^2)^2 \cdot \|\Sigma_K\|^2 \right],$$

where the last inequality follows from (D.17). Besides, for the second term on the right-hand side of (C.25), combining (C.25), (C.26), triangle inequality, and Cauchy-Schwarz inequality, we have

$$\left\| \mathbb{E}_{(x,u,x',u')} \left[ G^2(x, u, x', u'; \vartheta) - \widetilde{G}^2(x, u, x', u'; \vartheta) \right] \right\|_2$$

$$\leq \left\{ \left\| \mathbb{E}_{(x,u,x',u')} [G^2(x, u, x', u'; \vartheta) \cdot \mathbb{1}_{\mathcal{A}^c}] \right\|_2 + \left\| \mathbb{E}_{(x,u,x',u')} [\phi(x', u')^\top \vartheta^2 \cdot \phi(x, u) \, \mathbb{1}_{\mathcal{B}^c}] \right\|_2 \right\}$$

$$\leq \left\{ \sqrt{\mathbb{P}(\mathcal{A}^c)} \cdot \sqrt{\mathbb{E} \left[ \|G^2(x, u, x', u'; \vartheta)\|_2^2 \right]} + \sqrt{\mathbb{P}(\mathcal{B}^c)} \cdot \sqrt{\mathbb{E} \left[ \|\phi(x, u) \cdot \phi(x', u')^\top \vartheta^2\|_2^2 \right]} \right\}. \tag{C.28}$$

For the expectations on the right-hand side of (C.28), using the inequality $(a + b)^2 \leq 2a^2 + 2b^2$, we have

$$\mathbb{E} \left[ \|G^2(x, u, x', u'; \vartheta)\|_2^2 \right]$$

$$\leq 2 \cdot \mathbb{E} \left\{ \left[ \vartheta^1 - c(x, u) + \phi(x, u)^\top \vartheta^2 \right]^2 \cdot \|\phi(x, u)\|_2^2 \right\} + 2 \cdot \mathbb{E} \left[ \|\phi(x, u) \cdot \phi(x', u')^\top \vartheta^2\|_2^2 \right]. \tag{C.29}$$

Further applying Cauchy-Schwarz inequality to (C.29), we have

$$\mathbb{E} \left\{ \left[ \vartheta^1 - c(x, u) + \phi(x, u)^\top \vartheta^2 \right]^2 \cdot \|\phi(x, u)\|_2^2 \right\}$$

$$\leq \left( \mathbb{E} \left\{ \left[ \vartheta^1 - c(x, u) + \phi(x, u)^\top \vartheta^2 \right]^4 \right\} \cdot \mathbb{E} \left[ \|\phi(x, u)\|_2^4 \right] \right)^{1/2}, \tag{C.30}$$

$$\mathbb{E} \left[ \|\phi(x, u) \cdot \phi(x', u')^\top \vartheta\|_2^2 \right] \leq \left( \mathbb{E} \left[ |\phi(x', u')^\top \vartheta|^4 \right] \cdot \mathbb{E} \left[ \|\phi(x, u)\|_2^4 \right] \right)^{1/2}. \tag{C.31}$$

Since the marginal distributions of $(x, u)$ and $(x', u')$ are both $N(0, \widetilde{\Sigma}_K)$, in (C.30) and (C.31) we bound the two terms in (C.29) using the fourth moments of $N(0, \widetilde{\Sigma}_K)$, which can be written as a polynomial of $J(K_0)$, $\|K\|_{\mathrm{fro}}$, $\|Q\|$, $\|R\|$, $\widetilde{R}_\Theta$, and $\widetilde{R}_\Omega$.

Meanwhile, recall that we have shown that $\mathbb{P}(\mathcal{A}^c) \leq T^{-6}$ and $\mathbb{P}(\mathcal{B}^c) \leq T^{-6}$. Thus, when $T$ is sufficiently large, by combining (C.25), (C.27), (C.28), and (C.29), we have $|F(\vartheta, \omega) - \widetilde{F}(\vartheta, \omega)| \leq 1/T$, which implies that

$$\left| \mathtt{Gap}(\widehat{\vartheta}, \widehat{\omega}) - \left[ \max_{\omega \in \mathcal{X}_\Omega} \widetilde{F}(\widehat{\vartheta}, \omega) - \min_{\vartheta \in \mathcal{X}_\Theta} \widetilde{F}(\vartheta, \widehat{\omega}) \right] \right|$$

$$\leq \max_{\omega \in \mathcal{X}_\Omega} \left| F(\widehat{\vartheta}, \omega) - \widetilde{F}(\widehat{\vartheta}, \omega) \right| + \max_{\vartheta \in \mathcal{X}_\Theta} \left| F(\vartheta, \widehat{\omega}) - \widetilde{F}(\vartheta, \widehat{\omega}) \right| \leq \frac{2}{T}. \tag{C.32}$$

Hereafter, we study the primal-dual gap in (C.13) conditioning on event $\mathcal{E}$. To simplify the notation, we define function $H(\vartheta, \omega; \phi, \phi')$ on $\mathcal{X}_\Theta \times \mathcal{X}_\Omega$ by

$$H(\vartheta, \omega; \phi, \phi') = \langle \widetilde{G}(x, u, x', u'; \vartheta), \omega \rangle - 1/2 \cdot \|\omega\|_2^2,$$

where the function $\widetilde{\phi}(x, u)$ is defined in (C.22), and we denote $\widetilde{\phi}(x, u)$ and $\widetilde{\phi}(x', u')$ by $\phi$ and $\phi'$, respectively. Using this definition, the objective function $\widetilde{F}(\vartheta, \omega)$ in (C.23) can be written as $\widetilde{F}(\vartheta, \omega) = \mathbb{E}_{(x, u, x', u')}[H(\vartheta, \omega; \phi, \phi')]$, where $(x, u)$ and $(x', u')$ are two consecutive state-action pairs. Note that $H(\vartheta, \omega; \phi, \phi')$ is a quadratic function of $(\vartheta, \omega)$ for all $\phi$ and $\phi'$. The partial gradients of $H(\vartheta, \omega; \phi, \phi')$ are given by

$$\nabla_{\vartheta^1} H(\vartheta, \omega; \phi, \phi') = \omega^1 + \widetilde{\phi}(x, u)^\top \omega^2, \tag{C.33}$$

$$\nabla_{\vartheta^2} H(\vartheta, \omega; \phi, \phi') = [\widetilde{\phi}(x, u)^\top \omega^2] \cdot [\widetilde{\phi}(x, u) - \widetilde{\phi}(x', u')], \tag{C.34}$$

$$\nabla_{\omega^1} H(\vartheta, \omega; \phi, \phi') = \vartheta^1 - \widetilde{c}(x, u) - \omega^1, \tag{C.35}$$

$$\nabla_{\omega^2} H(\vartheta, \omega; \phi, \phi') = \widetilde{G}^2(x, u, x', u'; \vartheta) - \omega^2. \tag{C.36}$$

By combining (C.22), (C.33), and (C.34), we can bound the norm of $\nabla_\vartheta H(\vartheta, \omega; \phi, \phi')$ by

$$\|\nabla_\vartheta H(\vartheta, \omega; \phi, \phi')\|_2 \leq |\omega^1 + \widetilde{\phi}(x, u)^\top \omega^2| + \|[\widetilde{\phi}(x, u)^\top \omega^2] \cdot [\widetilde{\phi}(x, u) - \widetilde{\phi}(x', u')]\|_2 \tag{C.37}$$

$$\leq |\omega^1| + 2\|\widetilde{\phi}(x, u)\|_2 \cdot \|\omega^2\|_2 \cdot [\|\widetilde{\phi}(x, u)\|_2 + \|\widetilde{\phi}(x', u')\|_2]$$

$$\leq J(K_0) + 16C_1^2 \cdot (1 + \|K\|_{\mathrm{fro}}^2)^2 \cdot \log^2 T \cdot [\sigma^2 + (1 + \|K\|_{\mathrm{fro}}^2) \cdot \|\Sigma_K\|]^2 \cdot \widetilde{R}_\Omega.$$

Here the second inequality holds when $\|\widetilde{\phi}(x, u)\|_2 \geq 1$ and the last inequality follows from (C.21). Similarly, combining triangle inequality, (C.35), and (C.36), we have

$$\|\nabla_\omega H(\vartheta, \omega; \phi, \phi')\|_2 \leq |\vartheta^1 - \widetilde{c}(x, u) - \omega^1| + + [(\|Q\|_{\mathrm{fro}} + \|R\|_{\mathrm{fro}}) \cdot \|\widetilde{\phi}(x, u)\|_2$$

$$+ (\|\widetilde{\phi}(x', u')\|_2 + \|\widetilde{\phi}(x, u)\|_2) \cdot \widetilde{R}_\Theta] \cdot \|\widetilde{\phi}(x, u)\|_2$$

$$\leq 2J(K_0) + 16C_1^2 \cdot \log^2 T \cdot [\sigma^2 + (1 + \|K\|_{\mathrm{fro}}^2) \cdot \|\Sigma_K\|]^2 \cdot \widetilde{R}_\Theta. \tag{C.38}$$

where the last equality holds since $\widetilde{R}_\Theta \geq \|Q\|_{\mathrm{fro}} + \|R\|_{\mathrm{fro}}$. Moreover, we have $\nabla_{\vartheta\vartheta}^2 H(\vartheta, \omega; \phi, \phi') = 0$ and $-\nabla_{\omega\omega}^2 H(\vartheta, \omega; \phi, \phi')$ is the identity matrix.

We utilize the following lemma, obtained from [69], to handle the dependence along the trajectory.

**Lemma C.3** (Geometrically $\beta$-mixing). Consider a linear dynamical system $X_{t+1} = LX_t + \varepsilon$, where $\{X_t\}_{t\geq 0} \subseteq \mathbb{R}^m$, $\varepsilon \sim N(0, \Psi)$ is the random noise, and $L \in \mathbb{R}^{m \times m}$ has spectral radius smaller than one. We denote by $\nu_t$ the marginal distribution of $X_t$ for all $t \geq 0$. Besides, the stationary distribution of this Markov chain is denoted by $N(0, \Sigma_\infty)$. For any integer $k \geq 1$, we define the $k$-th mixing coefficient as

$$\beta(k) = \sup_{t \geq 0} \mathbb{E}_{x \sim \nu_t} \left[ \left\| \mathbb{P}_{X_k}(\cdot \mid X_0 = x) - \mathbb{P}_{N(0, \Sigma_\infty)}(\cdot) \right\|_{\mathrm{TV}} \right].$$

Furthermore, for any $\rho \in (\rho(L), 1)$ and any $k \geq 1$, we have

$$\beta(k) \leq C_{\rho, L} \cdot \left[ \mathrm{tr}(\Sigma_\infty) + m \cdot (1 - \rho)^{-2} \right]^{1/2} \cdot \rho^k,$$

where $C_{\rho, L}$ is a constant that solely depends on $\rho$ and $A$. That is, $\{X_t\}_{t\geq 0}$ is geometrically $\beta$-mixing.

*Proof.* See Proposition 3.1 in [69] for a detailed proof. $\qquad\square$

Recall that under policy $\pi_K$, $\{(x_t, u_t)\}_{t \geq 0}$ form a linear dynamic system characterized by (D.13) and (D.14). Since $\rho(L) = \rho(A - BK) < 1$, Lemma C.3 implies that, for all $\rho \in (\rho(A - BK), 1)$, $(x_t, u_t)_{t \geq 0}$ is a geometrically $\beta$-mixing stochastic process with parameter $\rho$. The following theorem, adapted from Theorem 1 in [72], establishes the primal-dual gap for a convex-concave minimax optimization problem involving a geometrically $\beta$-mixing stochastic process.

**Theorem C.4** (Primal-dual gap for minimax optimization)**.** Let $\mathcal{X}$ and $\mathcal{Y}$ are bounded and closed convex sets such that $\|x - x'\|_2 \leq D$ for all $x, x' \in \mathcal{X}$ and $\|y - y'\|_2 \leq D$ for all $y, y' \in \mathcal{Y}$. Consider the gradient algorithm for stochastic minimax optimization problem

$$\min_{x \in \mathcal{X}} \max_{y \in \mathcal{Y}} F(x, y) = \mathbb{E}_{\xi \sim \pi_\xi}[\Phi(x, y; \xi)], \tag{C.39}$$

where $\xi$ is a random variable with distribution $\pi_\xi$ and $F(x, y)$ is convex in $x$ and concave in $y$. In addition, we assume that $\pi_\xi$ is the stationary distribution of a Markov chain $\{\xi_t\}_{t \geq 0}$ which is geometrically $\beta$-mixing with parameter $\rho \in (0, 1)$. Specifically, we assume that there exists a constant $C_\xi > 0$ such that, for all $k \geq 1$, the $k$-th mixing coefficient satisfy $\beta(k) \leq C_\xi \cdot \rho^k$. Furthermore, we consider the case where, almost surely for every $\xi \sim \pi_\xi$, $\Phi(x, y; \xi)$ is $L_1$-Lipschitz in both $x$ and $y$, $\nabla_x \Phi(x, y; \xi)$ is $L_2$-Lipschitz in $x$ for all $y \in \mathcal{Y}$, and $\nabla_y \Phi(x, y; \xi)$ is $L_2$-Lipschitz in $y$ for all $x \in \mathcal{X}$. Here, without loss of generality, we assume that $D, L_1, L_2 > 1$. Consider solving the optimization problem in (C.39) via $T$ iterations of the gradient-based updates

$$x_t = \Pi_{\mathcal{X}}\big[x_{t-1} - \alpha_t \nabla_x \Phi(x_{t-1}, y_{t-1}; \xi_{t-1})\big], \qquad y_t = \Pi_{\mathcal{Y}}\big[y_{t-1} + \alpha_t \cdot \nabla_y \Phi(x_{t-1}, y_{t-1}; \xi_{t-1})\big],$$

where $t \in [T]$, $\Pi_{\mathcal{X}}$ and $\Pi_{\mathcal{Y}}$ are projection operators, and $\{\alpha_t = \alpha/\sqrt{t}\}_{t \in [T]}$ are the stepsizes, where $\alpha > 0$ is a constant. Let

$$\widehat{x} = \frac{\sum_{t \in [T]} \alpha_t \cdot x_t}{\sum_{t \in [T]} \alpha_t}, \qquad \widehat{y} = \frac{\sum_{t \in [T]} \alpha_t \cdot y_t}{\sum_{t \in [T]} \alpha_t}$$

be the final output of the algorithm. Then, there exists an absolute constant $C > 0$ such that, for any $\delta \in (0, 1)$, with probability at least $1 - \delta$, the primal-dual gap satisfies

$$\max_{y \in \mathcal{Y}} F(\widehat{x}, y) - \min_{x \in \mathcal{X}} F(x, \widehat{y}) \leq \frac{C \cdot (D^2 + L_1^2 + L_1 L_2 D)}{\log(1/\rho)} \cdot \frac{\log^2 T + \log(1/\delta)}{\sqrt{T}} + \frac{C \cdot C_\xi L_1 D}{T}.$$

*Proof.* This theorem follows from Theorem 1 in [72], where we set $\alpha_t = \alpha/\sqrt{t}$ for all $t \geq 1$, and focus on the case where $\{\xi_t\}_{t \geq 0}$ is geometrically $\beta$-mixing. Under the mixing assumption, for any $k \geq 1$, the $k$-th mixing coefficient of $\{\xi_t\}_{t \geq 0}$ satisfies $\beta(k) \leq C_\xi \cdot \rho^k$. Then, for any $\delta, \eta \in (0, 1)$, Theorem 1 in [72] implies

$$\max_{y \in \mathcal{Y}} F(\widehat{x}, y) - \min_{x \in \mathcal{X}} F(x, \widehat{y}) \leq \left(\sum_{t=1}^T \alpha_t\right)^{-1} \Bigg(A_0 + A_1 \cdot \eta \cdot \sum_{t=1}^T \alpha_t + A_2 \sum_{t=1}^T \alpha_t^2 + \tag{C.40}$$

$$16DL_1 \cdot \left\{2\tau(\eta) \cdot \log[\tau(\eta)/\delta] \cdot \left[\sum_{t=1}^T \alpha_t^2 + \tau(\eta) \cdot \alpha_1\right]\right\}^{1/2}\Bigg),$$

where we define $\tau(\eta) = \log(\eta/C_\xi)/\log(\rho)$ and denote

$$A_0 = D^2 + 12D \cdot \alpha_1 \cdot \tau(\eta) \qquad A_1 = 4L_1 D \qquad A_2 = 10L_1^2 + (24L_1^2 + 8L_1 L_2 D) \cdot \tau(\eta).$$

Now we set $\alpha_t = \alpha/\sqrt{t}$ and $\eta = C_\xi/T$ in (C.40), which implies that $\tau(\eta) = \log T/\log(1/\rho)$. Moreover, note that for all $T \geq 1$, we have $2\sqrt{T+1} - 2 \leq \sum_{t=1}^T 1/\sqrt{t} \leq 2\sqrt{T} - 1$ and $\sum_{t=1}^T 1/t \leq \log T + 1$. The last term on the right-hand side of (C.40) can be upper bounded by

$$16DL_1 \cdot \left\{2\log T/\log(1/\rho) \cdot \log[\tau(\eta)/\delta] \cdot \left[\log T + 1 + \alpha \cdot \log T/\log(1/\rho)\right]\right\}^{1/2}$$

$$\leq 16DL_1 \cdot \left\{2\log T/\log(1/\rho) \cdot [\log\log T + \log(1/\delta)] \cdot \left[\log T + 1 + \alpha \cdot \log T/\log(1/\rho)\right]\right\}^{1/2}$$

$$\leq C \cdot DL_1 \cdot \log T/\log(1/\rho) \cdot \sqrt{\log\log T + \log(1/\delta)}, \tag{C.41}$$

where $C$ is an absolute constant. Moreover, for the first three terms, we have

$$A_0 = D^2 + 12D \cdot \alpha \cdot \log T / \log(1/\rho) \le C \cdot D^2 \log T / \log(1/\rho), \quad A_1 \cdot \eta \le C \cdot C_\xi L_1 D / T,$$

$$\text{(C.42)}$$

$$A_2 \cdot \sum_{t=1}^{T} \alpha_t^2 \le \left[ 10L_1^2 + (24L_1^2 + 8L_1 L_2 D) \cdot \log T / \log(1/\rho) \right] \cdot (\log T + 1)$$

$$\le C \cdot [L_1^2 + L_1 L_2 D] \cdot \log^2 T / \log(1/\rho). \quad \text{(C.43)}$$

Thus, combining (C.40), (C.41), (C.42), and (C.43), we obtain that

$$\max_{y \in \mathcal{Y}} F(\widehat{x}, y) - \min_{x \in \mathcal{X}} F(x, \widehat{y}) \le C \cdot \left[ (D^2 + L_1^2 + L_1 L_2 D) / \log(1/\rho) \cdot \log T \cdot \log(T/\delta) / \sqrt{T} + C_\xi L_1 D / T \right],$$

which concludes the proof of Theorem C.4. $\qquad\square$

In order to apply Theorem C.4 to the minimax optimization in (C.23), we only need to specify parameters $C_\xi$, $D$, $L_1$, and $L_2$. First, for any $\rho \in (\rho(A - BK), 1)$, by Lemma C.3, we can set

$$C_\xi = C_{\rho,L} \cdot \left[ \text{tr}(\widetilde{\Sigma}_K) + (d + k) \cdot (1 - \rho)^2 \right]^{1/2}$$

$$\le 2C_{\rho,L} \cdot \sqrt{d + k} \cdot \left\{ \left[ \sigma^2 + (1 + \|K\|_{\text{fro}}^2) \cdot \|\Sigma_K\| \right]^{1/2} + (1 - \rho)^{-1} \right\}. \quad \text{(C.44)}$$

Moreover, by the definitions of $\mathcal{X}_\Theta$ and $\mathcal{X}_\Omega$ in (4.1) and (4.2), respectively, we can set $D$ by

$$D^2 = 2[J(K_0)]^2 + \widetilde{R}_\Theta^2 + (1 + \|K\|_{\text{fro}}^2)^4 \cdot \widetilde{R}_\Omega^2. \quad \text{(C.45)}$$

Moreover, by (C.37), (C.38), and the form of $\nabla^2 G(\theta, \omega; \phi, \phi')$, we have

$$L_1 \le 16C_1^2 \cdot \log^2 T \cdot \left[ \sigma^2 + (1 + \|K\|_{\text{fro}}^2) \cdot \|\Sigma_K\| \right]^2 \cdot \left[ (1 + \|K\|_{\text{fro}}^2)^2 \cdot \widetilde{R}_\Omega + \widetilde{R}_\Theta \right], \quad L_2 = 1.$$

$$\text{(C.46)}$$

Combining Theorem C.4, (C.44), (C.45), and (C.46), we to obtain an upper bound for the primal-dual gap in (C.13). Specifically, for any $\rho \in (\rho(A - BK), 1)$ and any $\delta \in (0, 1)$, with probability at least $1 - \delta$, the primal-dual gap of the optimization problem in (C.23) is bounded by

$$C \cdot \log^4 T \cdot \left[ \sigma^2 + (1 + \|K\|_{\text{fro}}^2) \cdot \|\Sigma_K\| \right]^4 \cdot \left[ (1 + \|K\|_{\text{fro}}^2)^2 \cdot \widetilde{R}_\Omega + \widetilde{R}_\Theta \right]^2$$

$$\cdot \left( \frac{\log^2 T + \log(1/\delta)}{\log(1/\rho) \cdot \sqrt{T}} + \frac{\sqrt{d + k}}{(1 - \rho) \cdot T} \right). \quad \text{(C.47)}$$

where $C > 0$ is an absolute constant. Besides, we note that $\sigma$ is a constant and that $\|\Sigma_K\| \ge \sigma_{\min}(\Psi) > 0$. Finally, recall that, when event $\mathcal{E}$ holds, the primal-dual gap is equal to $\max_{\omega \in \mathcal{X}_\Omega} \widetilde{F}(\widehat{\vartheta}, \omega) - \min_{\vartheta \in \mathcal{X}_\Theta} \widetilde{F}(\vartheta, \widehat{\omega})$. Combining (C.32), (C.47) with $\delta = T^{-5}$, and the fact that $\mathbb{P}(\mathcal{E}) \ge 1 - 2T^{-5}$, we conclude that

$$\text{Gap}(\widehat{\vartheta}, \widehat{\omega}) \le C \cdot \log^4 T \cdot (1 + \|K\|_{\text{fro}}^2)^4 \cdot \|\Sigma_K\|^4 \cdot \left[ (1 + \|K\|_{\text{fro}}^2)^2 \cdot \widetilde{R}_\Omega + \widetilde{R}_\Theta \right]^2$$

$$\cdot \left( \frac{\log^2 T + \log(T^5)}{\log(1/\rho) \cdot \sqrt{T}} + \frac{\sqrt{d + k}}{(1 - \rho) \cdot T} \right) + \frac{2}{T}$$

$$\le C \cdot (1 + \|K\|_{\text{fro}}^2)^4 \cdot \|\Sigma_K\|^4 \cdot \left[ (1 + \|K\|_{\text{fro}}^2)^2 \cdot \widetilde{R}_\Omega + \widetilde{R}_\Theta \right]^2 \cdot \frac{\log^6 T}{(1 - \rho) \cdot \sqrt{T}} \quad \text{(C.48)}$$

holds with probability at least $1 - 3T^{-5} \ge 1 - T^{-4}$, where in the second inequality we use the fact that $1 - 1/x < \log x < x + 1$ holds for all $x > 0$, which implies that $1/\log(1/\rho) \le 1/(1 - \rho)$. This further implies that the first term on the right-hand side of the first inequality dominates the second term. The upper bound of $\text{Gap}(\widehat{\vartheta}, \widehat{\omega})$ in (C.48) concludes the last step of our proof. Finally, combining (C.17) and (C.48), we complete the proof of Theorem 4.2. $\qquad\square$

## C.2  Proof of Theorem 4.3

*Proof.* Our proof of the global convergence can be decomposed into two steps. In the first step, similar to the analysis in [26], we study the geometry of the average return $J(K)$, as a function of $K$. Specifically, we show that $J(K)$ is gradient dominated [51]. Note that we study the ergodic setting with system noise and stochastic policies. In contrast, [26] study the case where both the transition and the policy are deterministic. Thus, their analysis of the geometry of $J(K)$ cannot be directly applied to our problem. Motivated by their analysis, we follow the similar approach to with modifications for our setting. In addition, in the second step, we utilize the geometry of $J(K)$ to show the global convergence of the actor-critic algorithm. Specifically, combining Theorem 4.2, we show that, with high probability, Algorithm 1 constructs a sequence of policies that converges linearly to the optimal policy $\pi_{K^*}$.

**Step 1.** As shown in (3.8) in Proposition 3.1, we can write $J(K)$ as

$$J(K) = \operatorname{tr}(P_K \Psi_\sigma) + \sigma^2 \cdot \operatorname{tr}(R) = \mathbb{E}_{x \in N(0, \Psi_\sigma)}(x^\top P_K x) + \sigma^2 \cdot \operatorname{tr}(R).$$

In the following lemma, for two policies $\pi_K$ and $\pi_{K'}$, we bound the difference between $x^\top P_K x$ and $x^\top P_{K'} x$. Then, taking expectation with respect to $x \in N(0, \Psi_\sigma)$ yields the difference between $J(K)$ and $J(K')$.

**Lemma C.5.** Let $K$ and $K'$ be two stable policies such that both $\rho(A - BK)$ and $\rho(A - BK')$ are smaller than one. For any $x \in \mathbb{R}^d$, let $\{x'_t\}_{t \geq 0} \subseteq \mathbb{R}^d$ be the sequence of states satisfying $x'_0 = x$ and $x'_{t+1} = (A - BK')x'_t$ for all $t \geq 0$. Then it holds that

$$x^\top P_{K'} x - x^\top P_K x = \sum_{t \geq 0} A_{K,K'}(x'_t),$$

where the function $A_{K,K'} : \mathbb{R}^d \to \mathbb{R}^d$ is defined as

$$A_{K,K'}(x) = 2x^\top (K' - K)^\top E_K x + x^\top (K' - K)^\top (R + B^\top P_K B)(K' - K)x.$$

*Proof.* Note that both $P_K$ and $P_{K'}$ satisfy the Bellman equation specified in (3.4). Moreover, using the operator $\mathcal{T}_K^\top$ defined in (D.3), we have $P_{K'} = \mathcal{T}_{K'}^\top(Q + K'^\top R K')$, which is equivalent to

$$x^\top P_{K'} x = \sum_{t \geq 0} x^\top [(A - BK')^t]^\top (Q + K'^\top R K')[(A - BK')^t]x. \tag{C.49}$$

By the construction in Lemma (C.5), for all $t \geq 0$, we have $(A - BK')^t x = x'_t$. Thus, by (C.49) we have

$$x^\top P_{K'} x = \sum_{t \geq 0} x'^\top_t (Q + K'^\top R K')x'_t = \sum_{t \geq 0} \left(x'^\top_t Q x'_t + u'^\top_t R u'_t\right), \tag{C.50}$$

where we define $u'_t = -K' x'_t$ for all $t \geq 0$. Thus, by (C.50), we have the following telescoping sum:

$$x^\top P_{K'} x - x^\top P_K x = \sum_{t \geq 0} \left[(x'^\top_t Q x'_t + u'^\top_t R u'_t) + x'^\top_t P_K x'_t - x'^\top_t P_K x'_t\right] - x'^\top_0 P_K x'_0$$

$$= \sum_{t \geq 0} \left[(x'^\top_t Q x'_t + u'^\top_t R u'_t) + x'^\top_{t+1} P_K x'_{t+1} - x'^\top_t P_K x'_t\right]. \tag{C.51}$$

Thus, in (C.51) we write $x^\top P_{K'} x - x^\top P_K x$ as a summation where each term can be written as a quadratic function of $x_t$. To further simplify (C.51), for any $x \in \mathbb{R}^d$, we have

$$x^\top Q x + (-K'x)^\top R(-K'x) + [(A - BK')x]^\top P_K[(A - BK')x] - x^\top P_K x \tag{C.52}$$

$$= x^\top \left[Q + (K' - K + K)^\top R(K' - K + K)\right]x +$$

$$\qquad x^\top \left[A - BK - B(K' - K)\right]^\top P_K \left[A - BK - B(K' - K)\right]x - x^\top P_K x$$

$$= 2x^\top (K' - K)^\top \left[(R + B^\top P_K B)K - B^\top P_K A\right]x + x^\top (K' - K)^\top (R + B^\top P_K B)(K' - K)x.$$

$$= 2x^\top (K' - K)^\top E_K x + x^\top (K' - K)^\top (R + B^\top P_K B)(K' - K)x,$$

where $E_K = (R + B^\top P_K B)K - B^\top P_K A$. Finally, combining (C.51) and (C.52), we complete the proof of this lemma. $\qquad\square$

In the following lemma, we utilize Lemma C.5 to show that $J(K)$ is gradient dominated.

**Lemma C.6** (Gradient domination of $J(K)$). Let $K^*$ be an optimal policy. Suppose $K$ has finite cost. Then, it holds that

$$\sigma_{\min}(\Psi) \cdot \|R + B^\top P_K B\|^{-1} \cdot \mathrm{tr}(E_K^\top E_K) \leq J(K) - J(K^*)$$
$$\leq 1/\sigma_{\min}(R) \cdot \|\Sigma_{K^*}\| \cdot \mathrm{tr}(E_K^\top E_K). \qquad \text{(C.53)}$$

*Proof.* For the upper bound in (C.53), bu (3.8) we obtain that

$$J(K) - J(K^*) = \mathrm{tr}[(P_K - P_K^*)\Psi_\sigma] = \mathbb{E}_{x \sim N(0, \Psi_\sigma)}\big[x^\top (P_K - P_K^*)x\big], \qquad \text{(C.54)}$$

where $\Psi_\sigma = \Psi + \sigma^2 BB^\top$ does not involve $K$ or $K^*$. Applying Lemma C.5 to (C.54) with $K' = K^*$, we have

$$J(K) - J(K^*) = -\mathbb{E}_{x_0^* \sim N(0, \Psi_\sigma)}\Big[\sum_{t \geq 0} A_{K, K^*}(x_t^*)\Big], \qquad \text{(C.55)}$$

where we define $x_t^* = (A - BK^*)^t x_0^*$ for all $t \geq 0$. Besides, by direct computation, we have

$$\mathbb{E}_{x_0^* \sim N(0, \Psi_\sigma)}\Big[\sum_{t \geq 0} x_t^*(x_t^*)^\top\Big]$$
$$= \mathbb{E}_{x \sim N(0, \Psi_\sigma)}\Big\{\sum_{t \geq 0}(A - BK^*)^t xx^\top[(A - BK^*)^t]^\top\Big\} = \mathcal{T}_{K^*}(\Psi_\sigma) = \Sigma_{K^*}, \qquad \text{(C.56)}$$

where the operator $\mathcal{T}_K$ is defined in (D.3).

Meanwhile, by the definition of $A_{K, K'}$, for any $x \in \mathbb{R}^d$, by completing the squares we have

$$A_{K, K'}(x) = 2x^\top (K' - K)^\top E_K x + x^\top (K' - K)^\top (R + B^\top P_K B)(K' - K)x$$
$$= \mathrm{tr}\Big\{xx^\top\big[K' - K + (R + B^\top P_K B)^{-1}E_K\big]^\top (R + B^\top P_K B)\big[K' - K + (R + B^\top P_K B)^{-1}E_K\big]\Big\}$$
$$\quad - \mathrm{tr}\big[xx^\top E_K^\top (R + B^\top P_K B)^{-1}E_K\big]$$
$$\geq -\mathrm{tr}\big[xx^\top E_K^\top (R + B^\top P_K B)^{-1}E_K\big], \qquad \text{(C.57)}$$

where the equality is attained by $K' = K - (R + B^\top P_K B)^{-1}E_K$.

Thus, combining (C.55), (C.56), and (C.57), we obtain that

$$J(K) - J(K^*) \leq \mathrm{tr}\big[\Sigma_{K^*}E_K^\top (R + B^\top P_K B)^{-1}E_K\big] \leq \|\Sigma_{K^*}\| \cdot \mathrm{tr}\big[\Sigma_{K^*}E_K^\top (R + B^\top P_K B)^{-1}E_K\big]$$
$$\leq \|\Sigma_{K^*}\| \cdot \|(R + B^\top P_K B)^{-1}\| \cdot \mathrm{tr}(E_K^\top E_K). \qquad \text{(C.58)}$$

Notice that $R + B^\top P_K B \succeq R$ implies $(R + B^\top P_K B)^{-1} \preceq R^{-1}$. Therefore, by (C.58) we obtain that $J(K) - J(K^*) \leq 1/\sigma_{\min}(R) \cdot \|\Sigma_{K^*}\| \cdot \mathrm{tr}(E_K^\top E_K)$, which establishes the upper bound in (C.53).

Furthermore, for the lower bound, since $K' = K - (R + B^\top P_K B)^{-1}E_K$ attains the lower bound in (C.57) and $K^*$ is the optimal policy, similar to (C.55) and (C.56), we have

$$J(K) - J(K^*) \geq J(K) - J(K') = -\mathbb{E}_{x_0^* \sim N(0, \Psi_\sigma)}\Big[\sum_{t \geq 0} A_{K, K'}(x_t')\Big]$$
$$= \mathrm{tr}\big[\Sigma_{K'}E_K^\top (R + B^\top P_K B)^{-1}E_K\big] \geq \sigma_{\min}(\Psi) \cdot \|R + B^\top P_K B\|^{-1} \cdot \mathrm{tr}(E_K^\top E_K),$$

where in the first equality we define $x_t' = (A - BK')^t$ for all $t \geq 0$, and the last inequality follows from the fact that $\Sigma_{K'} \succeq \Psi \succeq \sigma_{\min}(\Psi) \cdot I_d$. Therefore, we conclude the proof of Lemma C.6. $\qquad \square$

Notice that $K = K^*$ achieves the minimum of $J(K)$. Lemma C.6 implies that

$$J(K) - J(K^*) \leq \lambda \cdot \langle E_K, E_K \rangle,$$

where $\lambda = 1/\sigma_{\min}(R) \cdot \|\Sigma_{K^*}\|$. That is, the difference of the objective can be bounded by the norm of the natural gradient. Therefore, updating the policy parameter $K$ in the direction of natural gradient $E_K$ yields decreases the objective value. Therefore, we conclude the first step.

**Step 2.** In the second part of the proof, equipped with Lemma C.6, we establish the global convergence of the natural actor-critic algorithm. Recall that we assume that the initial policy $\pi_{K_0}$ is stable, which implies that $J(K_0)$ is finite. Moreover, according to Algorithm 1, the policy parameters are updated via

$$K_{t+1} = K_t - \gamma \cdot \widehat{E}_{K_t}, \qquad \widehat{E}_{K_t} = \widehat{\Theta}_t^{22} K_t - \widehat{\Theta}_t^{21}, \tag{C.59}$$

where $\widehat{\Theta}_t$ is the estimator of $\Theta_{K_t}$ returned by Algorithm 2.

We use mathematical induction to show that $\{J(K_t)\}_{t \geq 0}$ is a monotone decreasing sequence. Suppose $J(K_t) \leq J(K_0)$. We define $K'_{t+1} = K_t - \gamma \cdot E_{K_t}$, i.e., $K'_{t+1}$ is obtained by a single step of natural policy gradient, starting from $K_t$. In the sequel, we use $J(K'_{t+1})$ to connect $J(K_t)$ and $J(K_{t+1})$. By Lemma C.5, we have

$$
\begin{aligned}
J(K'_{t+1}) - J(K_t) &= \mathbb{E}_{x \sim N(0,\Psi_\sigma)}[x^\top (P_{K'_{t+1}} - P_{K_t})x] \\
&= -2\gamma \cdot \mathrm{tr}\big(\Sigma_{K'_{t+1}} \cdot E_{K_t}^\top E_{K_t}\big) + \gamma^2 \cdot \mathrm{tr}\big[\Sigma_{K'_{t+1}} \cdot E_{K_t}^\top (R + B^\top P_{K_t} B) E_{K_t}\big] \\
&= -2\gamma \cdot \mathrm{tr}\big(\Sigma_{K'_{t+1}} \cdot E_{K_t}^\top E_{K_t}\big) + \gamma^2 \cdot \|R + B^\top P_{K_t} B\| \cdot \mathrm{tr}\big(\Sigma_{K'_{t+1}} \cdot E_{K_t}^\top E_{K_t}\big).
\end{aligned}
\tag{C.60}
$$

When $\gamma$ is sufficiently small such that

$$\gamma \cdot \big[\|R\| + \sigma_{\min}^{-1}(\Psi) \cdot \|B\|^2 \cdot J(K_0)\big] \leq 1, \tag{C.61}$$

by triangle inequality, we have

$$\gamma \cdot \|R + B^\top P_{K_t} B\| \leq \gamma \cdot \big[\|R\| + \|B\|^2 \cdot \|P_{K_t}\|\big] \leq \gamma \cdot \big[\|R\| + \sigma_{\min}^{-1}(\Psi) \cdot \|B\|^2 \cdot J(K_0)\big] < 1, \tag{C.62}$$

where the second inequality follows from Lemma C.1 and the induction assumption that $J(K_t) \leq J(K_0)$, and the last inequality follows from (C.61). Thus, combining (C.60) and (C.62), we have

$$
\begin{aligned}
J(K'_{t+1}) - J(K_t) &\leq -\gamma \cdot \mathrm{tr}\big(\Sigma_{K'_{t+1}} \cdot E_{K_t}^\top E_{K_t}\big) \leq -\gamma \cdot \sigma_{\min}(\Psi) \cdot \mathrm{tr}\big(E_{K_t}^\top E_{K_t}\big), \\
&\leq -\gamma \cdot \sigma_{\min}(\Psi) \cdot \sigma_{\min}(R) \cdot \|\Sigma_{K^*}\|^{-1} \cdot \big[J(K_t) - J(K^*)\big].
\end{aligned}
\tag{C.63}
$$

where the third inequality follows from the fact that $\Sigma_{K'_{t+1}} \succeq \Psi$, and the last inequality follows from Lemma C.6. Note that (C.63) implies that $J(K'_{t+1}) \leq J(K_t) \leq J(K_0)$.

Furthermore, by the difference between $J(K_{t+1})$ and $J(K'_{t+1})$ can be bounded by

$$
\begin{aligned}
\big|J(K_{t+1}) - J(K'_{t+1})\big| &= \big|\mathrm{tr}\big[(P_{K_{t+1}} - P_{K'_{t+1}}) \cdot \Psi_\sigma\big]\big| \leq \|\Psi_\sigma\|_{\mathrm{fro}} \cdot \big\|P_{K_{t+1}} - P_{K'_{t+1}}\big\| \\
&\leq \big[\|\Psi\|_{\mathrm{fro}} \cdot + \sigma^2 \cdot \|B\|_{\mathrm{fro}}^2\big] \cdot \big\|P_{K_{t+1}} - P_{K'_{t+1}}\big\|.
\end{aligned}
\tag{C.64}
$$

Now we utilize the following Lemma, obtained from [26], to construct and upper bound for $\|P_{K_{t+1}} - P_{K'_{t+1}}\|$.

**Lemma C.7** (Perturbation of $P_K$)**.** Suppose $\pi_{K'}$ is a small perturbation of $\pi_K$ in the sense that

$$\|K' - K\| \leq \sigma_{\min}(\Psi)/4 \cdot \|\Sigma_K\|^{-1}\|B\|^{-1} \cdot (\|A - BK\| + 1)^{-1}, \tag{C.65}$$

then we have

$$
\begin{aligned}
\|P_{K'} - P_K\| &\leq 6\sigma_{\min}^{-1}(\Psi) \cdot \|\Sigma_K\| \cdot \|K\| \cdot \|R\| \\
&\quad \cdot \big(\|K\| \cdot \|B\| \cdot \|A - BK\| + \|K\| \cdot \|B\| + 1\big) \cdot \|K - K'\|.
\end{aligned}
\tag{C.66}
$$

*Proof.* This lemma is a slight modification of Lemma 24 in [26]. Here we sketch the proof. See [26, Lemmas 17 and 24] for a detailed proof.

Recall that we define operator $\mathcal{T}_K$ in (D.3). The operator norm of $\mathcal{T}_K$ is defined as $\|\mathcal{T}_K\| \leq \sup_\Omega \|\mathcal{T}_K(\Omega)\|/\|\Omega\|$, where the supremum is taken over all symmetric matrices. As shown in Lemma 17 in [26], we have $\|\mathcal{T}_K\| \leq \sigma_{\min}^{-1}(\Psi) \cdot \|\Sigma_K\|$. Moreover, under the condition in (C.65), in the proof of Lemma 24 in [26], it is shown that

$$\|P_{K'} - P_K\| \leq 6\|\mathcal{T}_K\| \cdot \|K\| \cdot \|R\| \cdot \big(\|K\| \cdot \|B\| \cdot \|A - BK\| + \|K\| \cdot \|B\| + 1\big) \cdot \|K - K'\|.$$

Combining this with the upper bound on $\|\mathcal{T}_K\|$, we conclude the proof. $\qquad\square$

To use this lemma, we need to verify (C.65). That is,

$$4\|K_{t+1} - K'_{t+1}\| \cdot (1 + \|A - BK'_{t+1}\|) \cdot \|B\| \cdot \|\Sigma_{K'_{t+1}}\| \leq \sigma_{\min}(\Psi). \qquad (C.67)$$

By the definition of $K_{t+1}$ and $K'_{t+1}$, we have

$$\|K_{t+1} - K'_{t+1}\| = \gamma \cdot \|\widehat{E}_{K_t} - E_{K_t}\| \leq \gamma \cdot \|\widehat{\Theta}_t - \Theta_{K_t}\|_{\mathrm{fro}} \cdot (1 + \|K_t\|), \qquad (C.68)$$

where $\widehat{E}_{K_t}$ is defined in (C.59). Plugging (C.68) into the left-hand side of (C.67), we obtain that

$$4\|K_{t+1} - K'_{t+1}\| \cdot (1 + \|A - BK'_{t+1}\|) \cdot \|B\| \cdot \|\Sigma_{K'_{t+1}}\|$$
$$\leq 4\gamma \cdot \|\widehat{\Theta}_t - \Theta_{K_t}\|_{\mathrm{fro}} \cdot (1 + \|K_t\|) \cdot (1 + \|A - BK'_{t+1}\|) \cdot \|B\| \cdot \|\Sigma_{K'_{t+1}}\|. \qquad (C.69)$$

Utilizing Lemma (C.1) and the fact that $J(K'_{t+1}) \leq J(K_0)$, we have

$$\|\Sigma_{K'_{t+1}}\| \leq J(K'_{t+1})/\sigma_{\min}(Q) \leq J(K_0)/\sigma_{\min}(Q). \qquad (C.70)$$

In addition, by triangle inequality, we have

$$\|A - BK'_{t+1}\| \leq \|A - BK_t\| + \gamma \cdot \|B\| \cdot \|E_{K_t}\|$$
$$\leq \|A - BK_t\| + \gamma \cdot \|B\| \cdot \|\Theta_{K_t}\| \cdot (1 + \|K_t\|). \qquad (C.71)$$

By the definition of $\Theta_K$ in (3.7), we have

$$\|\Theta_{K_t}\| \leq \|Q\| + \|R\| + (\|A\|_{\mathrm{fro}} + \|B\|_{\mathrm{fro}})^2 \cdot \|P_{K_t}\|$$
$$\leq \|Q\| + \|R\| + (\|A\|_{\mathrm{fro}} + \|B\|_{\mathrm{fro}})^2 \cdot J(K_0)/\sigma_{\min}(\Psi), \qquad (C.72)$$

where the last inequality follows from Lemma (C.1) and the induction assumption. Furthermore, by triangle inequality, it holds that

$$\|K_{t+1}\| \leq \|K_t\| + \gamma \cdot \|E_{K_t}\| \leq \|K_t\| + \gamma \cdot \|\Theta_{K_t}\| \cdot (1 + \|K_t\|)$$
$$\leq \|K_t\| + \gamma \cdot \big[\|Q\| + \|R\| + (\|A\|_{\mathrm{fro}} + \|B\|_{\mathrm{fro}})^2 \cdot J(K_0)/\sigma_{\min}(\Psi)\big] \cdot (1 + \|K_t\|). \tag{C.73}$$

In the sequel, we set

$$\gamma = \big[\|R\| + \sigma_{\min}^{-1}(\Psi) \cdot \|B\|^2 \cdot J(K_0)\big]^{-1}. \qquad (C.74)$$

Note that we assume that $\|Q\|$, $\|R\|$, $\|A\|$, $\|B\|$, $\sigma_{\min}(Q)$, $\sigma_{\min}(R)$ are all constants. Combining (C.69), (C.70), (C.71), and (C.72), we conclude that there exists a polynomial $\Upsilon_1(\cdot, \cdot)$ such that

$$4\|K_{t+1} - K'_{t+1}\| \cdot (1 + \|A - BK'_{t+1}\|) \cdot \|B\| \cdot \|\Sigma_{K'_{t+1}}\| \leq \Upsilon_1\big[\|K_t\|, J(K_0)\big] \cdot \|\widehat{\Theta}_t - \Theta_{K_t}\|_{\mathrm{fro}}. \tag{C.75}$$

Furthermore, for the right-hand side of (C.66), combining (C.68), (C.69), (C.70), (C.71), (C.72), and (C.73). we conclude that there exists a polynomial $\Upsilon_2(\cdot, \cdot)$ such that

$$\big[\|\Psi\|_{\mathrm{fro}} \cdot + \sigma^2 \cdot \|B\|_{\mathrm{fro}}^2\big] \cdot 6\sigma_{\min}^{-1}(\Psi) \cdot \|\Sigma_{K'_{t+1}}\| \cdot \|K_{t+1'}\| \cdot \|R\|$$
$$\cdot \big(\|K'_{t+1}\| \cdot \|B\| \cdot \|A - BK'_{t+1}\| + \|K'_{t+1}\| \cdot \|B\| + 1\big) \cdot \|K_{t+1} - K'_{t+1}\|$$
$$\leq \Upsilon_2\big[\|K_t\|, J(K_0)\big] \cdot \|\widehat{\Theta}_t - \Theta_{K_t}\|_{\mathrm{fro}}. \tag{C.76}$$

Meanwhile, in Theorem 4.2 we have shown that, there exists a polynomial $\Upsilon_3(\cdot, \cdot)$ such that, for $T$ sufficiently large, Algorithm 2 with $T$ iterations returns an estimator $\widehat{\Theta}_t$ for $\Theta_{K_t}$ such that

$$\|\widehat{\Theta}_t - \Theta_{K_t}\|_{\mathrm{fro}} \leq \frac{\Upsilon_3\big[\|K_t\|, J(K_0)\big]}{\kappa_{K_t}^* \cdot \sqrt{(1-\rho)}} \cdot \frac{\log^3 T}{T^{1/4}} \qquad (C.77)$$

holds with probability at least $1 - T^{-4}$, where $\rho \in (\rho(A - BK_t), 1)$ and $\kappa_{K_t}^*$ is specified in Lemma 3.2, which depends only on $\rho$, $\sigma$, and $\sigma_{\min}(\Psi)$. Notice that $\log^3 T \cdot T^{-1/4} \leq T^{-1/5}$ for $T$ sufficiently

large. Therefore, in the GTD algorithm for estimating $\Theta_{K_t}$, we set the number of iterations $T_t$ sufficiently large such that

$$\Upsilon_1\big[\|K_t\|, J(K_0)\big] \cdot \Upsilon_3\big[\|K_t\|, J(K_0)\big] \cdot {\kappa_{K_t}^*}^{-1} \cdot (1-\rho)^{-1/2} \cdot T_t^{-1/5} \leq \sigma_{\min}(\Psi),$$

$$\Upsilon_2\big[\|K_t\|, J(K_0)\big] \cdot \Upsilon_3\big[\|K_t\|, J(K_0)\big] \cdot {\kappa_{K_t}^*}^{-1} \cdot (1-\rho)^{-1/2} \cdot T_t^{-1/5}$$
$$\leq \epsilon/2 \cdot \sigma_{\min}(\Psi) \cdot \sigma_{\min}(R) \cdot \|\Sigma_{K^*}\|^{-1} \tag{C.78}$$

hold simultaneously. For such a $T_t$, combining (C.75) and (C.77), we conclude that (C.67) holds. Lemma C.7 implies that (C.66) is true. Combining (C.64), (C.66), (C.76), and (C.77), we conclude that

$$\big|J(K_{t+1}) - J(K'_{t+1})\big| \leq \epsilon/2 \cdot \sigma_{\min}(\Psi) \cdot \sigma_{\min}(R) \cdot \|\Sigma_{K^*}\|^{-1} \tag{C.79}$$

holds with probability at least $1 - T_t^{-4}$. Thus, when $J(K_t) - J(K^*) > \epsilon$, combining (C.63) and (C.79) we have

$$J(K_{t+1}) - J(K_t) \leq -\epsilon/2 \cdot \gamma\sigma_{\min}(\Psi) \cdot \sigma_{\min}(R) \cdot \|\Sigma_{K^*}\|^{-1} < 0.$$

Therefore, we have shown that, as long as $J(K_t) - J(K^*) \geq \epsilon$, $J(K_{t+1}) < J(K_t)$ holds with probability at least $1 - T_t^{-1/4}$.

Meanwhile, (C.63) implies that,

$$J(K'_{t+1}) - J(K^*) \leq \big[1 - \gamma \cdot \sigma_{\min}(\Psi) \cdot \sigma_{\min}(R) \cdot \|\Sigma_{K^*}\|^{-1}\big] \cdot \big[J(K_t) - J(K^*)\big]$$

By (C.79), when $J(K_t) - J(K^*) \geq \epsilon$, with probability $1 - T_t^{-4}$, we have

$$J(K_{t+1}) - J(K^*) \leq \big[1 - \gamma/2 \cdot \sigma_{\min}(\Psi) \cdot \sigma_{\min}(R) \cdot \|\Sigma_{K^*}\|^{-1}\big] \cdot \big[J(K_t) - J(K^*)\big],$$

which shows that, in terms of the policy parameter, natural actor-critic algorithm converges linearly. Specifically, with

$$N \geq 2\|\Sigma_{K^*}\|/\gamma \cdot \sigma_{\min}^{-1}(\Psi) \cdot \sigma_{\min}^{-1}(R) \cdot \log\big\{2[J(K_0) - J(K^*)]/\epsilon\big\} \tag{C.80}$$

policy updates, we have $J(K_N) - J(K^*) \leq \epsilon$ with high-probability, where $\gamma$ is specified in (C.74).

Finally, it remains to determine $T_t$ for all $t \in [N]$. Notice that $T_t$ satisfies the two inequalities in (C.78). Thus, we set

$$T_t \geq \Upsilon_4[\|K_t\|, J(K_0)] \cdot {\kappa_{K_t}^*}^{-5} \cdot (\Xi_{K_t}) \cdot \big[1 - \rho(A - BK_t)\big]^{-5/2} \cdot \epsilon^{-5}$$

for some polynomial function $\Upsilon_4(\cdot, \cdot)$. With such a $T_t$, the fail probability $T_t^{-4} \leq \epsilon^{-20}$. Notice that the total number of iterations depends on $\epsilon$ only through $\log(1/\epsilon)$. Thus, the total fail probability can be bounded by $\epsilon^{10}$. Therefore, we conclude the proof. ☐

# D   Proofs of the Auxiliary Results

In this section, we provides the proofs for Proposition 3.1 and Lemma 3.2.

## D.1   Proof of Proposition 3.1

*Proof.* We first establish (3.8). Note that under $\pi_K$, we can write $u_t$ as $-Kx_t + \sigma \cdot \eta_t$, where $\eta_t \sim N(0, I_d)$. This implies that, for all $\geq 0$, we have

$$\mathbb{E}[c(x_t, u_t) \,|\, x_t] = x_t^\top Q x_t + \mathbb{E}_{\eta_t \sim N(0, I_d)}[(-Kx_t + \sigma \cdot \eta_t)^\top R(-Kx_t + \sigma \cdot \eta_t)]$$
$$= x_t^\top (Q + K^\top RK)x_t + \sigma^2 \cdot \mathrm{tr}(R). \tag{D.1}$$

Thus, combining (D.1) and the definition of $J(K)$ in (2.1), we have

$$J(K) = \lim_{T \to \infty} \mathbb{E}\bigg\{\frac{1}{T}\sum_{t \geq 0}^T \mathbb{E}[c(x_t, u_t) \,|\, x_t]\bigg\} = \lim_{T \to \infty} \mathbb{E}\bigg\{\frac{1}{T}\sum_{t \geq 0}^T [x_t^\top (Q + K^\top RK)x_t + \sigma^2 \cdot \mathrm{tr}(R)]\bigg\}$$
$$= \mathbb{E}_{x \sim \nu_K}[x^\top (Q + K^\top RK)x] + \sigma^2 \cdot \mathrm{tr}(R) = \mathrm{tr}\big[(Q + K^\top RK)\Sigma_K\big] + \sigma^2 \cdot \mathrm{tr}(R), \tag{D.2}$$

where the third inequality in (D.2) holds because the limiting distribution of $\{x_t\}_{t\geq 0}$ is $\nu_K$.

It remains to establish the second equality in (3.8). To this end, for $K \in \mathbb{R}^{k \times d}$ such that $\rho(A - BK) < 1$, we define operators we define $\mathcal{T}_K$ and $\mathcal{T}_K^\top$ by

$$\mathcal{T}_K(\Omega) = \sum_{t \geq 0} (A - BK)^t \Omega \big[(A - BK)^t\big]^\top, \qquad \mathcal{T}_K^\top(\Omega) = \sum_{t \geq 0} \big[(A - BK)^t\big]^\top \Omega (A - BK)^t,$$
(D.3)

where $\Omega \in \mathbb{R}^{d \times d}$ is positive definite. By definition, $\mathcal{T}_K(\Omega)$ and $\mathcal{T}_K^\top(\Omega)$ satisfy Lyapunov equations

$$\mathcal{T}_K(\Omega) = \Omega + (A - BK)\mathcal{T}_K(\Omega)(A - BK)^\top, \tag{D.4}$$

$$\mathcal{T}_K^\top(\Omega) = \Omega + (A - BK)^\top \mathcal{T}_K^\top(\Omega)(A - BK), \tag{D.5}$$

respectively. Moreover, for any positive definite matrices $\Omega_1, \Omega_2$, since $\rho(A - BK) < 1$, we have

$$\text{tr}[\Omega_1 \cdot \mathcal{T}_K(\Omega_2)] = \sum_{t \geq 0} \text{tr}\big\{\Omega_1 (A - BK)^t \Omega_2 [(A - BK)^t]^\top\big\}$$
$$= \sum_{t \geq 0} \text{tr}\big\{[(A - BK)^t]^\top \Omega_1 (A - BK)^t \Omega_2\big\} = \text{tr}[\mathcal{T}_K^\top(\Omega_1) \cdot \Omega_2]. \tag{D.6}$$

Meanwhile, by combining (3.3), (3.4), (D.4), and (D.5), we have $\Sigma_K = \mathcal{T}_K(\Psi_\sigma)$ and $P_K = \mathcal{T}_K^\top(Q + K^\top RK)$. Thus, (D.6) implies that

$$\text{tr}\big[(Q + K^\top RK) \cdot \Sigma_K\big] = \text{tr}\big[(Q + K^\top RK) \cdot \mathcal{T}_K(\Psi_\sigma)\big] = \text{tr}\big[\mathcal{T}_K^\top(Q + K^\top RK) \cdot \Psi_\sigma\big] = \text{tr}(P_K \Psi_\sigma).$$

Combining this equation with (D.2), we establish the second equation of (3.8).

In the following, we establish the value functions. In the setting of LQR, the state-value function $V_K$ is given by

$$V_K(x) = \sum_{t=0}^{\infty} \big\{\mathbb{E}[c(x_t, u_t) \,|\, x_0 = x, u_t = -Kx_t + \sigma \cdot \eta_t] - J(K)\big\}$$
$$= \sum_{t=0}^{\infty} \big\{\mathbb{E}[x_t^\top (Q + K^\top RK)x_t] + \sigma^2 \cdot \text{tr}(R) - J(K)\big\}. \tag{D.7}$$

Combining the linear dynamics in (3.2) and (D.7), we see that $V_K$ is a quadratic function, which is denoted by $V_k(x) = x^\top P_K x + \alpha_K$, where both $P_K$ and $\alpha_K$ depends on $K$. Note that $V_K$ satisfies the Bellman equation

$$V_K(x) = \mathbb{E}_{u \sim \pi_K}[c(x, u)] - J(K) + \mathbb{E}[V_K(x') \,|\, x],$$

where $x'$ is the next state given $(x, u)$. Thus, for any $x \in \mathbb{R}^d$, we have

$$x^\top P_K x = x(Q + K^\top RK)x + x^\top (A - BK)^\top P_K (A - BK)x.$$

Thus, $P_K$ is the unique positive definite solution to the Bellman equation in (3.4). Meanwhile, since $\mathbb{E}_{x \sim \nu_K}[V_K(x)] = 0$, we have $\alpha_K = -\text{tr}(P_K \Sigma_K)$. Hence, we establish (3.5).

Furthermore, for any state-action pair $(x, u)$, we have

$$Q_K(x, u) = c(x, u) - J(K) + \mathbb{E}[V_K(x') \,|\, x, u]$$
$$= c(x, u) - J(K) + (Ax + Bu)^\top P_K (Ax + Bu) + \text{tr}(P_K \Psi) - \text{tr}(P_K \Sigma_K)$$
$$= x^\top Qx + u^\top Ru + (Ax + Bu)^\top P_K (Ax + Bu) - \sigma^2 \cdot \text{tr}(R + P_K BB^\top) - \text{tr}(P_K \Sigma_K),$$

where $x'$ in the first equality is the next state following $(x, u)$, and the last equality follows from (3.8) and the fact that $\Psi_\sigma = \Psi + \sigma^2 \cdot BB^\top$. Thus, we prove (3.6).

It remains to derive the policy gradient $\nabla_K J(K)$. By (3.8), we have

$$\nabla_K J(K) = 2RK\Sigma_K + \nabla_K \text{tr}(Q_0 \cdot \Sigma_K)\big|_{Q_0 = Q + K^\top RK}, \tag{D.8}$$

where the second term denotes that we first take compute the gradient $\nabla_K \text{tr}[Q_0 \Sigma_K]$ with respect to $K$ and then set $Q_0 = Q + K^\top RK$. Recall that we can write $\Sigma_K = \mathcal{T}_K(\Psi_\sigma)$. The following lemma enables us to compute the gradient involving $\mathcal{T}_K$.

**Lemma D.1.** Let $W$ and $\Psi$ be two positive definite matrices. Then it holds that

$$\nabla_K \operatorname{tr}\big[W \cdot \mathcal{T}_K(\Psi)\big] = -2B^\top \mathcal{T}_K^\top(W)(A - BK)\mathcal{T}_K(\Psi).$$

*Proof.* To simplify the notation, we define operator $\mathcal{F}_K$ by

$$\mathcal{F}_K^\top(\Omega) = (A - BK)^\top \Omega(A - BK)$$

and let $\mathcal{F}_K^{\top,t}$ be the $t$-th composition of $\mathcal{F}_K$. Thus, by the definition of $\mathcal{T}_K^\top$ and $\mathcal{F}_K^\top$, we have

$$\mathcal{T}_K^\top(\Omega) = \sum_{t \geq 0} \mathcal{F}_K^{\top,t}(\Omega).$$

Moreover, by (D.4) we have

$$\operatorname{tr}\big[W \cdot \mathcal{T}_K(\Psi)\big] = \operatorname{tr}(W\Psi) + \operatorname{tr}\big[(A - BK)^\top W(A - BK) \cdot \mathcal{T}_K(\Psi)\big],$$

which implies that

$$\nabla_K \operatorname{tr}\big[W \cdot \mathcal{T}_K(\Psi)\big] = -2B^\top W(A - BK)\mathcal{T}_K(\Psi) + \nabla_K \operatorname{tr}[W_1 \mathcal{T}_K(\Psi)]\Big|_{W_1 = \mathcal{F}_K(\Omega)}. \tag{D.9}$$

For any $k \geq 1$, by recursively applying (D.9) for $k$ times, we have

$$\nabla_K \operatorname{tr}\big[W \cdot \mathcal{T}_K(\Psi)\big]$$
$$= -2B^\top \left[\sum_{t=0}^{k} \mathcal{F}_K^{\top,t}(W)\right](A - BK)\mathcal{T}_K(\Psi) + \nabla_K \operatorname{tr}[W_1 \mathcal{T}_K(\Psi)]\Big|_{W_1 = \mathcal{F}_K^{(k+1)}(\Omega)}. \tag{D.10}$$

Meanwhile, since $\rho(A - BK) < 1$, we have

$$\lim_{k \to \infty} \operatorname{tr}\big[\mathcal{F}_K^{\top,k}(W)\mathcal{T}_K(\Psi)\big] \leq \lim_{k \to \infty} \|W\| \cdot \operatorname{tr}[\mathcal{T}_K(\Psi)] \cdot \rho(A - BK)^{2k} = 0.$$

Thus, by letting $k$ on the right-hand side of (D.10) go to infinity, we obtain

$$\nabla_K \operatorname{tr}\big[W \cdot \mathcal{T}_K(\Psi)\big] = -2B^\top \left[\sum_{t=0}^{\infty} \mathcal{F}_K^{\top,t}(W)\right](A - BK)\mathcal{T}_K(\Psi) = -2B^\top \mathcal{T}_K^\top(W)(A - BK)\mathcal{T}_K(\Psi).$$

Therefore, we conclude the proof of the lemma. $\qquad\square$

By the above lemma, since $\Sigma_K = \mathcal{T}_K(\Psi_\sigma)$, we have

$$\nabla_K \operatorname{tr}(Q_0 \cdot \Sigma_K)\big|_{Q_0 = Q + K^\top RK} = \nabla_K \operatorname{tr}\big[Q_0 \cdot \mathcal{T}_K(\Psi_\sigma)\big]\big|_{Q_0 = Q + K^\top RK}$$
$$= -2B^\top \mathcal{T}_K^\top(Q + K^\top RK)(A - BK)\mathcal{T}_K(\Psi_\sigma) = -2B^\top P_K(A - BK)\Sigma_K, \tag{D.11}$$

where we use the fact that $P_K = \mathcal{T}_K^\top(Q + K^\top RK)$. Therefore, combining (D.8) and (D.11), we establish (3.9), which completes the proof of Proposition 3.1. $\qquad\square$

## D.2 Proof of Lemma 3.2

We present a stronger lemma than Lemma 3.2, whose proof automatically validates Lemma 3.2.

**Lemma D.2.** Suppose $\rho(A - BK) < 1$. Let $N(0, \widetilde{\Sigma}_K)$ be the stationary distribution of the state-action pair $(x, u)$ when following policy $\pi_K$. Then for $\Xi_K$ defined in (3.15), we have

$$\Xi_K = \big(\widetilde{\Sigma}_K \otimes_s \widetilde{\Sigma}_K\big) - \big(\widetilde{\Sigma}_K L^\top\big) \otimes_s \big(\widetilde{\Sigma}_K L^\top\big) = \big(\widetilde{\Sigma}_K \otimes_s \widetilde{\Sigma}_K\big)\big(I - L^\top \otimes_s L^\top\big). \tag{D.12}$$

Moreover, $\Xi_K$ is a invertible matrix whose operator norm is bounded by $2[\sigma^2 + (1 + \|K\|_{\text{fro}}^2) \cdot \|\Sigma_K\|]$. There exists a positive number $\kappa_K^*$ such that the minimum singular value of the matrix in the left-hand side of (3.16) is lower bounded by a constant $\kappa_K^* > 0$, where $\kappa_K^*$ only depends on $\rho(A - BK)$, $\sigma$, and $\sigma_{\min}(\Psi)$. Furthermore, since $\Xi_K$ is invertible, the linear equation in (3.16) has unique solution $\vartheta_K^*$, whose first and second components are $J(K)$ and $\operatorname{svec}(\Theta_K)$, respectively.

*Proof.* Throughout the proof of Lemma D.2, for any state-action pair $(x, u) \in \mathbb{R}^{d+k}$, we denote the next state-action pair following policy $\pi_K$ by $(x', u')$. Then we can write

$$x' = Ax + Bu + \epsilon, \qquad u' = -Kx' + \sigma \cdot \eta = -KAx - KBu - K\epsilon + \sigma \cdot \eta, \qquad \text{(D.13)}$$

where $\epsilon \sim N(0, \Psi)$ and $\eta \in N(0, I_k)$. For notational simplicity, we denote $(x, u)$ and $(x', u')$ by $z$ and $z'$, respectively. Thus, we can write $z' = Lz + \varepsilon$, where we define

$$L = \begin{pmatrix} A & B \\ -KA & -KB \end{pmatrix} = \begin{pmatrix} I_d \\ -K \end{pmatrix} (A \quad B), \qquad \varepsilon = \begin{pmatrix} \epsilon \\ -K\epsilon + \sigma \cdot \eta \end{pmatrix}. \qquad \text{(D.14)}$$

Since it holds that $\rho(MN) = \rho(NM)$ for any two matrices $M$ and $N$ [32, Theorem 1.3.22], we have $\rho(L) = \rho(A - BK) < 1$. Meanwhile, by definition, $\varepsilon \in \mathbb{R}^{d+k}$ is a centered Gaussian random variable with covariance

$$\begin{pmatrix} \Psi & -\Psi K^\top \\ -K\Psi & K\Psi K^\top + \sigma^2 \cdot I_k \end{pmatrix}, \qquad \text{(D.15)}$$

which is denoted by $\widetilde{\Psi}_\sigma$ for notational simplicity. In addition, for $x \sim \nu_K$ and $u \sim \pi_K(\cdot \,|\, x)$, we denote the joint distribution of $z = (x, u)$ by $\widetilde{\nu}_K$, which is a centered Gaussian distribution in $\mathbb{R}^{d \times k}$. Since $x \sim N(0, \Sigma_K)$ and $u = -Kx + \sigma \cdot I_k$, we can write $\widetilde{\nu}_K$ as $N(0, \widetilde{\Sigma}_K)$, where $\widetilde{\Sigma}_K \in \mathbb{R}^{(d+k) \times (d+k)}$ can be written as

$$\widetilde{\Sigma}_K = \begin{pmatrix} \Sigma_K & -\Sigma_K K^\top \\ -K\Sigma_K & K\Sigma_K K^\top + \sigma^2 \cdot I_k \end{pmatrix} = \begin{pmatrix} 0 & 0 \\ 0 & \sigma^2 \cdot I_k \end{pmatrix} + \begin{pmatrix} I_d \\ -K \end{pmatrix} \Sigma_K \begin{pmatrix} I_d \\ -K \end{pmatrix}^\top. \qquad \text{(D.16)}$$

Thus, by triangle inequality we have

$$\left\|\widetilde{\Sigma}_K\right\|_{\text{fro}} \le \sigma^2 \cdot k + \|\Sigma_K\| \cdot (d + \|K\|_{\text{fro}}^2), \qquad \left\|\widetilde{\Sigma}_K\right\| \le \sigma^2 + (1 + \|K\|_{\text{fro}}^2) \cdot \|\Sigma_K\|, \qquad \text{(D.17)}$$

where in (D.17) we use the fact that $\|AB\|_{\text{fro}} \le \|A\|_{\text{fro}} \cdot \|B\|$.

Furthermore, since $L$ defined in (D.14) satisfy $\rho(L) < 1$, $\widetilde{\Sigma}_K$ is the unique positive definite solution to the Lyapunov equation

$$\widetilde{\Sigma}_K = L\widetilde{\Sigma}_K L^\top + \widetilde{\Psi}_K, \qquad \text{(D.18)}$$

where $\widetilde{\Psi}_K$ is defined in (D.15). Moreover, the feature mapping can be written as $\phi(x, u) = \phi(z) = \text{svec}(zz^\top)$, which implies that

$$\begin{aligned} \phi(x, u) - \phi(x', u') &= \text{svec}\big[zz^\top - (Lz + \varepsilon)(Lz + \varepsilon)^\top\big] \\ &= \text{svec}\big(zz^\top - Lzz^\top L^\top - Lz\varepsilon^\top - \varepsilon z^\top L^\top - \varepsilon\varepsilon^\top\big). \end{aligned}$$

Hence, since $\varepsilon$ is independent of $z$, by the definition of $\Xi_K$ in (3.15), we have

$$\Xi_K = \mathbb{E}_{z \sim \widetilde{\nu}_K}[\phi(z)\,\text{svec}(xx^\top - Lxx^\top L^\top - \widetilde{\Psi}_\sigma)^\top].$$

Now let $M$ and $N$ by any two matrices, by direct computation, we have

$$\begin{aligned} \text{svec}(M)^\top \Xi_K \,\text{svec}(N) &= \mathbb{E}_{z \sim \widetilde{\nu}_K}\big[\langle zz^\top, M\rangle \cdot \langle zz^\top - Lzz^\top L^\top - \widetilde{\Psi}_\sigma, N\rangle\big] \\ &= \mathbb{E}_{z \sim \widetilde{\nu}_K}\big[z^\top Mzz^\top (N - L^\top NL)z\big] - \mathbb{E}_{z \sim \widetilde{\nu}_K}[z^\top Mz] \cdot \langle\widetilde{\Psi}_\sigma, N\rangle \\ &= \mathbb{E}_{g \sim N(0, I_{d+k})}\big[g^\top \widetilde{\Sigma}_K^{1/2} M\widetilde{\Sigma}_K^{1/2} gg^\top \widetilde{\Sigma}_K^{1/2}(N - L^\top NL)\widetilde{\Sigma}_K^{1/2} g\big] - \langle\widetilde{\Sigma}_K, M\rangle \cdot \langle\widetilde{\Psi}_\sigma, N\rangle, \end{aligned}$$
$$\text{(D.19)}$$

where $\widetilde{\Sigma}_K^{1/2}$ is the square root of $\widetilde{\Sigma}_K$ defined in (D.18). We utilize the following Lemma to compute the expectation of the product of quadratic forms of Gaussian random variables.

**Lemma D.3.** Let $g \sim N(0, I_d)$ be the standard Gaussian random variable in $\mathbb{R}^d$ and let $A_1, A_2$ be two symmetric matrices. Then we have

$$\mathbb{E}[g^\top A_1 g \cdot g^\top A_2 g] = 2\,\text{tr}(A_1 A_2) + \text{tr}(A_1) \cdot \text{tr}(A_2).$$

*Proof.* See, e.g., [48, 43] for a detailed proof. □

Applying this lemma to (D.19), we have

$$
\begin{aligned}
\operatorname{svec}(M)^\top \Xi_K \operatorname{svec}(N) \\
&= 2\operatorname{tr}\big[\widetilde{\Sigma}_K^{1/2} M \widetilde{\Sigma}_K^{1/2} \cdot \widetilde{\Sigma}_K^{1/2}(N - L^\top N L)\widetilde{\Sigma}_K^{1/2}\big] \\
&\quad + \operatorname{tr}\big(\widetilde{\Sigma}_K^{1/2} M \widetilde{\Sigma}_K^{1/2}\big) \cdot \operatorname{tr}\big[\widetilde{\Sigma}_K^{1/2}(N - L^\top N L)\widetilde{\Sigma}_K^{1/2}\big] - \langle \widetilde{\Sigma}_K, M\rangle \cdot \langle \widetilde{\Psi}_\sigma, N\rangle \\
&= 2\big\langle M, \widetilde{\Sigma}_K(N - L^\top N L)\widetilde{\Sigma}_K\big\rangle + \big\langle M, \widetilde{\Sigma}_K\big\rangle \cdot \big[\big\langle N - L^\top N L, \widetilde{\Sigma}_K\big\rangle - \big\langle\widetilde{\Psi}_\sigma, N\big\rangle\big]. \quad \text{(D.20)}
\end{aligned}
$$

Note that $\widetilde{\Sigma}_K$ satisfy the Lyapunov equation in (D.18), which implies that

$$
\big\langle N - L^\top N L, \widetilde{\Sigma}_K\big\rangle = \big\langle N, \widetilde{\Sigma}_K\big\rangle - \big\langle N, L\widetilde{\Sigma}_K L^\top\big\rangle = \big\langle N, \widetilde{\Psi}_\sigma\big\rangle.
$$

Thus, by (D.20) we have

$$
\begin{aligned}
\operatorname{svec}(M)^\top \Xi_K \operatorname{svec}(N) &= 2\big\langle M, \widetilde{\Sigma}_K(N - L^\top N L)\widetilde{\Sigma}_K\big\rangle = 2\operatorname{svec}(M)^\top \operatorname{svec}\big[\widetilde{\Sigma}_K(N - L^\top N L)\widetilde{\Sigma}_K\big] \\
&= 2\operatorname{svec}(M)^\top \big(\widetilde{\Sigma}_K \otimes_s \widetilde{\Sigma}_K - \widetilde{\Sigma}_K L^\top \otimes_s \widetilde{\Sigma}_K L^\top\big)\operatorname{svec}(N)^\top \\
&= 2\operatorname{svec}(M)^\top \big[\big(\widetilde{\Sigma}_K \otimes_s \widetilde{\Sigma}_K\big)(I - L^\top \otimes L^\top)\big]\operatorname{svec}(N),
\end{aligned}
$$

where the last equality follows from the fact that

$$
(A \otimes_s B)(C \otimes_s D) = 1/2 \cdot (AC \otimes_s BD + AD \otimes_s BC)
$$

holds for any matrices $A, B, C, D$. Thus, we have established (D.12). Since $\rho(L) = \rho(A - BK) < 1$, $I - L^\top \otimes L^\top$ is positive definite, which implies that $\Xi_K$ is invertible.

Now we consider the linear equation in (3.16). Since $\Xi_K$ is invertible,

$$
\widetilde{\Xi}_K = \begin{pmatrix} 1 & 0 \\ \mathbb{E}_{(x,u)}[\phi(x,u)] & \Xi_K \end{pmatrix} = \begin{pmatrix} 1 & 0 \\ \operatorname{svec}(\widetilde{\Sigma}_K) & \Xi_K \end{pmatrix} \quad \text{(D.21)}
$$

is also invertible. Thus, (3.16) has unique solution $\vartheta_K^*$. Moreover, to bound the smallest singular value of $\widetilde{\Xi}_K$, we note that the inverse of $\widetilde{\Xi}_K$ can be written as

$$
\widetilde{\Xi}_K^{-1} = \begin{pmatrix} 1 & 0 \\ -\Xi_K^{-1}\operatorname{svec}(\widetilde{\Sigma}_K) & \Xi_K^{-1} \end{pmatrix},
$$

whose operator norm is bounded via

$$
\big\|\widetilde{\Xi}_K^{-1}\big\|^2 \leq 1 + \big\|\Xi_K^{-1}\operatorname{svec}(\widetilde{\Sigma}_K)\big\|_2^2 + \|\Xi_K^{-1}\|^2. \quad \text{(D.22)}
$$

By (D.12), we have

$$
\begin{aligned}
\Xi_K^{-1}\operatorname{svec}(\widetilde{\Sigma}_K) &= (I - L^\top \otimes_s L^\top)^{-1}(\widetilde{\Sigma}_K \otimes_s \widetilde{\Sigma}_K)^{-1}\operatorname{svec}(\widetilde{\Sigma}_K) \\
&= (I - L^\top \otimes_s L^\top)^{-1}(\widetilde{\Sigma}_K^{-1} \otimes_s \widetilde{\Sigma}_K^{-1})\operatorname{svec}(\widetilde{\Sigma}_K) = (I - L^\top \otimes_s L^\top)^{-1}\operatorname{svec}(\widetilde{\Sigma}_K^{-1}). \quad \text{(D.23)}
\end{aligned}
$$

The following lemma characterizes the eigenvalues of symmetric Kronecker matrices.

**Lemma D.4** (Lemma 7.2 in [2]). Let $A$ and $B$ be two matrices in $\mathbb{R}^{m \times m}$ that can be diagonalized simultaneously. Moreover, let $\lambda_1, \ldots, \lambda_m$ and $\mu_1, \ldots, \mu_m$ be the eigenvalues of $A$ and $B$, respectively. Then, the eigenvalues of $A \otimes_s B$ are given by $\{1/2 \cdot (\lambda_i\mu_j + \lambda_j\mu_i), i, j \in [m]\}$.

By Lemma D.4, the spectral radius of $L^\top \otimes_s L^\top$ is bounded by $\rho^2(L) = \rho^2(A - BK) < 1$. By (D.23) we have

$$
\big\|\Xi_K^{-1}\operatorname{svec}(\widetilde{\Sigma}_K)\big\|_2 \leq \big[1 - \rho^2(L)\big]^{-1} \cdot \|\widetilde{\Sigma}_K^{-1}\|_F \leq \sqrt{d+k} \cdot \big[1 - \rho^2(L)\big]^{-1} \cdot \|\widetilde{\Sigma}_K^{-1}\|. \quad \text{(D.24)}
$$

Besides, by (D.12) we have

$$
\|\Xi_K^{-1}\| \leq \big\|(I - L^\top \otimes_s L^\top)^{-1}\big\| \cdot \big\|\widetilde{\Sigma}_K^{-1} \otimes_s \widetilde{\Sigma}_K^{-1}\big\| \leq \big[1 - \rho^2(L)\big]^{-1} \cdot \big\|\widetilde{\Sigma}_K^{-1}\big\|^2. \quad \text{(D.25)}
$$

Notice that $\|\widetilde{\Sigma}_K^{-1}\| = 1/\sigma_{\min}(\widetilde{\Sigma}_K)$. Hence, combining (D.22), (D.24), and (D.25) we conclude that

$$
\big\|\widetilde{\Xi}_K^{-1}\big\|^2 \leq 1 + (d+k) \cdot \big[1 - \rho(L)^2\big]^{-2} \cdot [\sigma_{\min}(\widetilde{\Sigma}_K)]^{-2} + \big[1 - \rho(L)^2\big]^{-2} \cdot [\sigma_{\min}(\widetilde{\Sigma}_K)]^{-4},
$$

which implies that

$$\sigma_{\min}(\widetilde{\Xi}_K) \geq \frac{[1 - \rho^2(A - BK)] \cdot [\sigma_{\min}(\widetilde{\Sigma}_K)]^2}{\left(1 + [1 - \rho^2(A - BK)]^2 \cdot [\sigma_{\min}(\widetilde{\Sigma}_K)]^4 + (d + k) \cdot [\sigma_{\min}(\widetilde{\Sigma}_K)]^2\right)^{1/2}} > 0.$$

Moreover, to see that $\sigma_{\min}(\widetilde{\Sigma}_K)$ only depends on $\sigma$ and $\sigma_{\min}(\Psi)$, for any $a \in \mathbb{R}^d$ and $b \in \mathbb{R}^k$, we have

$$\begin{pmatrix} a \\ b \end{pmatrix}^\top \widetilde{\Sigma}_K \begin{pmatrix} a \\ b \end{pmatrix} = \mathbb{E}_{(x,u)\sim\widetilde{\nu}_K}[(a^\top x + b^\top u)^2] = \mathbb{E}_{x\sim\nu_K, \eta\sim N(0,I_k)}\left\{[(a - K^\top b)x + \sigma \cdot \eta]^2\right\}$$

$$\geq \sigma^2 \cdot \|b\|_2^2 + \sigma_{\min}(\Psi) \cdot \|a - K^\top b\|_2^2 \geq (\sigma^2 - \sigma_{\min}(\Psi) \cdot \|K\|^2) \cdot \|b\|_2^2 + \sigma_{\min}(\Psi) \cdot \|a\|_2^2.$$

Thus, suppose $\sigma^2$ is sufficiently large such that $\sigma^2 - \sigma_{\min}(\Psi) \cdot \|K\|^2 > 0$, $\sigma_{\min}(\widetilde{\Sigma}_K)$ is lower bounded by $\min\{\sigma^2 - \sigma_{\min}(\Psi) \cdot \|K\|^2, \sigma_{\min}(\Psi)\}$. Therefore, we can find a constant $\kappa_K^*$ depending only on $\rho(A - KB)$, $\sigma$, and $\sigma_{\min}(\Psi)$ such that $\sigma_{\min}(\widetilde{\Xi}_K) \geq \kappa_K^*$.

Finally, to obtain an upper bound on $\|\Xi_K\|$, by triangle inequality and Lemma D.4 we have

$$\|\Xi_K\| \leq \|\widetilde{\Sigma}_K \otimes_s \widetilde{\Sigma}_K\| \cdot \left(1 + \|L^\top \otimes_s L^\top\|\right) \leq \|\widetilde{\Sigma}_K\|^2 \cdot \left(1 + \|L\|^2\right) \leq 2\|\widetilde{\Sigma}_K\|^2,$$

where we use the fact that $\rho(L) < 1$. Applying (D.17) to the inequality above, we obtain that

$$\|\Xi_K\| \leq 2[\sigma^2 + (1 + \|K\|_{\mathrm{fro}}^2) \cdot \|\Sigma_K\|],$$

which concludes the proof. $\qquad\square$

[Supplementary Material 2]

# A  Algorithms

In this section, we present the details of the actor-critic algorithm.

---

**Algorithm 1** Natural Actor-Critic Algorithm for Linear Quadratic Regulator

---

**Input:** Initial policy $\pi_{K_0}$ such that $\rho(A - BK_0) < 1$, stepsizes $\gamma$ for policy update, and a policy evaluation algorithm.

**Initialization:** Set the current policy $\pi_K$ by letting $K \leftarrow K_0$.

**while** updating current policy **do**

    **Critic step.** Estimate $\Theta_K$ in (3.7) via a policy evaluation algorithm, e.g., the on-policy GTD algorithm (Algorithm 2), which returns an estimator $\widehat{\Theta}$ of $\Theta_K$.

    **Actor step.** Update the policy parameter by $K \leftarrow K - \gamma \cdot (\widehat{\Theta}^{22}K - \widehat{\Theta}^{21})$.

**end while**

**Output:** The final policy $\pi_K$, matrix $\widehat{\Theta}$ that estimates $\Theta_K$, and $\widehat{J}$ that approximates $J(K)$.

---

---

**Algorithm 2** On-Policy Gradient-Based Temporal-Difference Algorithm for Policy Evaluation

---

**Input:** Policy $\pi_K$, number of iterations $T$, and stepsizes $\{\alpha_t\}_{t \in [T]}$.

**Output:** Estimator $\widehat{\Theta}$ of $\Theta_K$ in (3.7).

Initialize the primal and dual variables by $\vartheta_0 \in \mathcal{X}_\Theta$ and $\omega_0 \in \mathcal{X}_\Omega$, respectively.

Sample the initial state $x_0 \in \mathbb{R}^d$ from the stationary distribution $\rho_K$. Take action $u_0 \sim \pi_K(\cdot \,|\, x_0)$ and obtain the reward $c_0$ and the next state $x_1$.

**for** $t = 1, 2, \ldots, T$ **do**

    Take action $u_t$ according to policy $\pi_K$, observe the reward $c_t$ and the next state $x_{t+1}$.

    Compute the TD-error $\delta_t = \vartheta_{t-1}^1 - c_{t-1} + [\phi(x_{t-1}, u_{t-1}) - \phi(x_t, u_t)]^\top \vartheta_{t-1}^2$.

    Update $\vartheta^1$ by $\vartheta_t^1 = \vartheta_{t-1}^1 - \alpha_t \cdot [\omega_{t-1}^1 + \phi(x_{t-1}, u_{t-1})^\top \omega_{t-1}^2]$.

    Update $\vartheta^2$ by $\vartheta_t^2 = \vartheta_{t-1}^2 - \alpha_t \cdot [\phi(x_{t-1}, u_{t-1}) - \phi(x_t, u_t)] \cdot \phi(x_{t-1}, u_{t-1})^\top \omega_{t-1}^2$.

    Update $\omega^1$ by $\omega_t^1 = (1 - \alpha_t) \cdot \omega_t^1 + \alpha_t \cdot (\vartheta_{t-1}^1 - c_{t-1})$.

    Update $\omega^2$ by $\omega_t^2 = (1 - \alpha_t) \cdot \omega_t^2 + \alpha_t \cdot \delta_t \cdot \phi(x_{t-1})$.

    Project $\vartheta_t$ and $\omega_t$ to $\mathcal{X}_\Theta$ and $\mathcal{X}_\Omega$, respectively.

**end for**

Define $\widehat{\vartheta} = (\widehat{\vartheta}^1, \widehat{\vartheta}^2) = (\sum_{t=1}^T \alpha_t \cdot \vartheta_t)/(\sum_{t=1}^T \alpha_t)$ and $\widehat{\omega} = (\sum_{t=1}^T \alpha_t \cdot \omega_t)/(\sum_{t=1}^T \alpha_t)$.

Return $\widehat{\vartheta}^1$ and $\widehat{\Theta} = \mathrm{smat}(\widehat{\vartheta}^2)$ as the estimators of $J(K)$ and $\Theta_K$, respectively.

---

# B  Extension to the Off-Policy Setting

Recall that in our natural actor-critic algorithm, the critic can apply any policy evaluation algorithm to estimate $\widehat{\Theta}_K$. When using an off-policy method, we obtain an off-policy actor-critic algorithm. In this section, we extend Algorithm 2 to the off-policy setting using importance sampling. Specifically, let $\pi_b$ be the behavior policy and suppose it induces a stationary distribution $\nu_b$ over the state space $\mathbb{R}^d$. Moreover, let $\pi_K$ be the policy of interest and let $\tau_K(x, u) = \pi_K(u \,|\, x)/\pi_b(u \,|\, x)$ be the importance sampling ratio. Then, the Bellman equation in (3.14) can be written as

$$\langle \phi(x, u), \theta_K^* \rangle = c(x, u) - J(K) + \langle \mathbb{E}[\phi(x', u') \cdot \tau_K(x', u') \,|\, x, u], \theta_K^* \rangle, \tag{B.1}$$

where $(x, u) \in \mathbb{R}^{d+k}$, $x'$ is the next state given $(x, u)$, and $u' \sim \pi_b(\cdot \,|\, x)$. In the following, we denote by $\mathbb{E}_{(x,u)}$ the expectation with respect to $x \sim \nu_b$ and $u \sim \pi_b(\cdot \,|\, x)$. Similar to $\Xi_K$ and $b_K$ defined in (3.15), for the off-policy setting, we define

$$\overline{\Xi}_K = \mathbb{E}_{(x,u)}\big\{\phi(x, u)\big[\phi(x, u) - \tau_K(x', u') \cdot \phi(x', u')\big]^\top\big\}, \qquad \overline{b}_K = \mathbb{E}_{(x,u)}\big[c(x, u)\phi(x, u)\big],$$

$$\overline{h}_K = \mathbb{E}_{(x,u)}[\phi(x, u) - \tau_K(x', u') \cdot \phi(x', u')], \qquad \overline{g}_K = \mathbb{E}_{(x,u)}[\phi(x, u)], \qquad \overline{a}_K = \mathbb{E}_{(x,u)}[c(x, u)].$$

Based on (B.1) and direct computation, it can be shown that $\vartheta_K^* = (J(K), \mathrm{svec}(\Theta_K^\top)^\top)^\top$ is the solution to linear equation

$$\Lambda_K \vartheta = \lambda_K, \qquad \Lambda_K = \begin{pmatrix} 1 & \overline{h}_K^\top \\ \overline{g}_K & \overline{\Xi}_K \end{pmatrix}, \qquad \vartheta = \begin{pmatrix} \vartheta^1 \\ \vartheta^2 \end{pmatrix}, \qquad \lambda_K = \begin{pmatrix} \overline{a}_K \\ \overline{b}_K \end{pmatrix}. \tag{B.2}$$

Similar to the derivations in §3.2, we propose to estimate $\vartheta_K^*$ by solving a minimax optimization problem:

$$\min_{\vartheta \in \mathcal{X}_\Theta} \max_{\omega \in \mathcal{X}_\Omega} \overline{F}(\vartheta, \omega) = \omega^\top [\Lambda_K \vartheta - \lambda_K] - 1/2 \cdot \|\omega\|_2^2. \tag{B.3}$$

Here, similar to Assumption (4.1), we let $\mathcal{X}_\Theta$ and $\mathcal{X}_\Omega$ be Euclidean balls given by $\mathcal{X}_\Theta$ and $\mathcal{X}_\Omega$ in (3.18) be defined as $\mathcal{X}_\Theta = \{\vartheta \colon \|\vartheta^1\|_2 \leq \widetilde{R}_\Theta\}$ and $\mathcal{X}_\Omega = \{\omega \colon \|\omega\|_2 \leq \widetilde{R}_\Omega\}$, where $\widetilde{R}_\Theta$ and $\widetilde{R}_\Omega$ are chosen appropriately. Notice that both $\overline{F}(\vartheta, \omega)$ and its gradient can be estimated unbiasedly using transitions sampled from the behavior policy. Solving (B.3) using stochastic gradient method, we obtain the off-policy GTD algorithm for the ergodic setting. See Algorithm 3 for the details. Combining this policy evaluation method with Algorithm 1, we establish the off-policy on-line natural actor-critic algorithm.

---

**Algorithm 3** Off-Policy Gradient-Based Temporal-Difference Algorithm for Policy Evaluation

---

**Input:** Policy $\pi_K$, number of iterations $T$, and stepsizes $\{\alpha_t\}_{t \in [T]}$, the behavior policy $\pi_b$ and its stationary distribution $\nu_b$.
**Output:** Estimators $\widehat{J}$ and $\widehat{\Theta}$ of $J(K)$ in (3.8) and $\Theta_K$ in (3.7), respectively.
Initialize the primal and dual variables by $\vartheta_0 \in \mathcal{X}_\Theta$ and $\omega_0 \in \mathcal{X}_\Omega$, respectively.
Sample the initial state $x_0 \in \mathbb{R}^d$ from the stationary distribution $\nu_b$. Take action $u_0 \sim \pi_b(\cdot \,|\, x_0)$ and obtain the reward $c_0$ and the next state $x_1$.
**for** $t = 1, 2, \ldots, T$ **do**
  Take action $u_t$ according to policy $\pi_K$, observe the reward $c_t$ and the next state $x_{t+1}$.
  Compute the TD-error $\delta_t = \vartheta_{t-1}^1 - c_{t-1} + [\phi(x_{t-1}, u_{t-1}) - \tau_K(x_t, u_t) \cdot \phi(x_t, u_t)]^\top \vartheta_{t-1}^2$.
  Update the primal variable $\vartheta$ by

$$\vartheta_t^1 = \vartheta_{t-1}^1 - \alpha_t \cdot [\omega_{t-1}^1 + \phi(x_{t-1}, u_{t-1})^\top \omega_{t-1}^2],$$
$$\vartheta_t^2 = \vartheta_{t-1}^2 - \alpha_t \cdot [\phi(x_{t-1}, u_{t-1}) - \tau_K(x_t, u_t) \cdot \phi(x_t, u_t)] \cdot [\phi(x_{t-1}, u_{t-1})^\top \omega_{t-1}^2 + \omega_{t-1}^1].$$

  Update the dual variable $\omega$ by

$$\omega_t^1 = (1 - \alpha_t) \cdot \omega_t^1 + \alpha_t \cdot \{\vartheta_{t-1}^1 + [\phi(x_{t-1}, u_{t-1}) - \tau_K(x_t, u_t) \cdot \phi(x_t, u_t)]^\top \vartheta_{t-1}^2 - c_{t-1}\},$$
$$\omega_t^2 = (1 - \alpha_t) \cdot \omega_t^2 + \alpha_t \cdot \delta_t \cdot \phi(x_{t-1}).$$

  Project $\vartheta_t$ and $\omega_t$ to $\mathcal{X}_\Theta$ and $\mathcal{X}_\Omega$, respectively.
**end for**
Define $\widehat{\vartheta} = (\widehat{\vartheta}^1, \widehat{\vartheta}^2) = (\sum_{t=1}^T \alpha_t \cdot \vartheta_t)/(\sum_{t=1}^T \alpha_t)$ and $\widehat{\omega} = (\sum_{t=1}^T \alpha_t \cdot \omega_t)/(\sum_{t=1}^T \alpha_t)$.
Return $\widehat{\vartheta}^1$ and $\widehat{\Theta} = \mathrm{smat}(\widehat{\vartheta}^2)$ as the estimators of $J(K)$ and $\Theta_K$, respectively.

---

Similar to Theorem 4.2, we have the following theorem that shows that Algorithm 3 converges at a sublinear rate to the desired solution $\vartheta_K^*$.

**Theorem B.1** (off-policy GTD). Let $\widehat{\vartheta}^1$ and $\widehat{\Theta}$ be the output of Algorithm 3 based on $T$ iterations. We set the stepsize to be $\alpha_t = \alpha/\sqrt{t}$ with $\alpha > 0$ being a constant. We assume that $\Lambda_K$ in (B.2) is invertible and that its minimum singular value is lower bounded by a constant $\kappa_k^* > 0$. Moreover, we assume that the Markov chain induced by the behavioral policy $\pi_b$ is geometrically $\beta$-mixing with parameter $\rho \in (0, 1)$. Let $\nu_b$ be the stationary distribution of this induced Markov chain. We assume that, for $(x, u) \sim \nu_b$, both $\phi(x, u)$ and $\tau_K(x, u)$ are sub-exponential random variables. Then, when the number of iterations $T$ is sufficiently large, with probability at least $1 - T^{-4}$, we have

$$\|\widehat{\Theta} - \Theta_K\|_{\mathrm{fro}}^2 \leq \frac{\Upsilon[\widetilde{R}_\Theta, \widetilde{R}_\Omega, J(K_0), \|K\|_{\mathrm{fro}}, \sigma_{\min}^{-1}(Q)]}{\kappa_K^{*\,2} \cdot (1 - \rho)} \cdot \frac{\log^6 T}{\sqrt{T}},$$

where $\Upsilon[\widetilde{R}_\Theta, \widetilde{R}_\Omega, J(K_0), \|K\|_{\mathrm{fro}}, \sigma_{\min}^{-1}(Q)]$ is a polynomial of $\widetilde{R}_\Omega$, $\widetilde{R}_\Omega$, $J(K_0)$, $\|K\|_{\mathrm{fro}}$, and $1/\sigma_{\min}(Q)$.

*Proof.* The proof of this theorem is parallel to that of Theorem 4.2, thus here we only sketch the proof for brevity.

The proof can be completed in three steps. In the first step, we show that $(\vartheta, \omega) = (\vartheta_K^*, 0)$ is the saddle point of the optimization problem in (B.3). To simplify the notation, we define a vector-valued function $G(x, u, x', u'; \vartheta)$ by

$$G^1(x, u, x', u'; \vartheta) = \vartheta^1 - c(x, u) + \langle \phi(x, u) - \tau_K(x', u') \cdot \phi(x', u'), \vartheta^2 \rangle,$$

$$G^2(x, u, x', u'; \vartheta) = \vartheta^1 \cdot \phi(x, u) + \left\{ \left[ \phi(x, u) - \phi(x', u') \cdot \tau_K(x', u') \right]^\top \vartheta^2 - c(x, u) \right\} \cdot \phi(x, u).$$
(B.4)

By definition, for all $(\vartheta, \omega)$, $\overline{F}(\vartheta, \omega)$ in (B.3) can be equivalently written as

$$\overline{F}(\vartheta, \omega) = \left\langle \mathbb{E}_{(x,u,x',u')}[G(x, u, x', u'; \vartheta)], \omega \right\rangle - 1/2 \cdot \|\omega\|_2^2.$$
(B.5)

Thus, for any $\vartheta$, the solution to the unconstrained maximization problem $\max_\omega F(\theta, \omega)$ is

$$w(\vartheta) = \mathbb{E}_{(x,u,x',u')}[G(x, u, x', u'; \vartheta)].$$

Recall that $c(x, u) = \langle \phi(x, u), \mathrm{svec}[\mathrm{diag}(Q, R)] \rangle$. Since both $\phi(x, u)$ and $\tau_K(x, u) \cdot \phi(x, u)$ are sub-exponential random vectors, when $\widetilde{R}_\Theta$ and $\widetilde{R}_\Omega$ are chosen properly, we can show that $\vartheta_K^* \in \mathcal{X}_\Theta$ and that $w(\vartheta) \in \mathcal{X}_\Omega$ for all $\vartheta \in \mathcal{X}_\Theta$. Notice that $w(\vartheta_K^*) = 0$. Thus, $(\vartheta_K^*, 0)$ is the solution to the minimax optimization problem in (B.3).

Then, in the second step, we relate the estimation error $\|\widehat{\Theta} - \Theta_K\|_{\mathrm{fro}}^2$ to the primal-dual gap

$$\mathrm{Gap}(\widehat{\vartheta}, \widehat{\omega}) = \max_{\omega \in \mathcal{X}_\Omega} \overline{F}(\widehat{\vartheta}, \omega) - \min_{\vartheta \in \mathcal{X}_\Theta} \overline{F}(\vartheta, \widehat{\omega}).$$
(B.6)

Similar to the derivations in (C.14)–(C.17), since the minimum singular value of $\Lambda_K$ is lower bounded by $\kappa_K^*$, it holds that

$$|\widehat{\vartheta}^1 - J(K)|^2 + \|\widehat{\Theta} - \Theta_K\|_{\mathrm{fro}}^2 \leq \kappa_K^{*-2} \cdot \mathrm{Gap}(\widehat{\vartheta}, \widehat{\omega}).$$
(B.7)

Thus, it suffices to bound the primal-dual gap in (B.6), which is achieved in the last step.

Specifically, we would like to utilize Theorem C.4 obtained from [72]. Since this theorem requires bounded iterates and Lipschitz gradient, similar to the third step in §C.1, we truncate the feature vector $\phi(x, u)$. In particular, we define

$$\mathcal{E} = \bigcap_{0 \leq t \leq T} \left\{ \|\phi(x_t, u_t)\|_2^2 \leq C_K \cdot \log T, \|\tau_K(x_t, u_t) \cdot \phi(x_t, u_t)\|_2^2 \leq C_K \cdot \log T \right\},$$
(B.8)

where $C_b$ is a constant specified by the stationary distribution $\nu_b$. Since both is $\phi(x, u)$ and $\phi(x, u) \cdot \tau_K(x, u)$ are sub-exponential random vector when $(x, u) \sim \nu_b$, it can be shown that $\mathcal{E}$ holds with probability at least $1 - 2T^{-5}$. Then we define truncated random vectors

$$\widetilde{\phi}(x, u) = \phi(x, u) \cdot \mathbb{1}_\mathcal{E}, \qquad \widetilde{\varphi}_K(x, u) = \phi(x, u) \cdot \tau_K(x, u) \cdot \mathbb{1}_\mathcal{E}$$

and the truncated minimax optimization problem,

$$\min_{\vartheta \in \mathcal{X}_\Theta} \max_{\omega \in \mathcal{X}_\Omega} \widetilde{F}(\vartheta, \omega) = \left\langle \mathbb{E}_{(x,u,x',u')}[\widetilde{G}(x, u, x', u'; \vartheta)], \omega \right\rangle - 1/2 \cdot \|\omega\|_2^2,$$
(B.9)

where we define $\widetilde{G}(x, u, x', u'; \vartheta)$ by

$$\widetilde{G}^1(x, u, x', u'; \vartheta) = \vartheta^1 - \widetilde{c}(x, u) + \langle \widetilde{\phi}(x, u) - \widetilde{\varphi}_K(x', u'), \vartheta^2 \rangle,$$

$$G^2(x, u, x', u'; \vartheta) = \vartheta^1 \cdot \widetilde{\phi}(x, u) + \left\{ \left[ \widetilde{\phi}(x, u) - \widetilde{\varphi}_K(x', u') \right]^\top \vartheta^2 - \widetilde{c}(x, u) \right\} \cdot \widetilde{\phi}(x, u).$$
(B.10)

Here we let $\widetilde{c}(x, u) = \langle \widetilde{\phi}(x, u), \mathrm{svec}[\mathrm{diag}(Q, R)] \rangle$ in (B.10). Similar to the derivations from (C.25) to (C.31), we can show that $\sup_{\vartheta, \omega} |\overline{F}(\vartheta, \omega) - \widetilde{F}(\vartheta, \omega)| \leq 1/T$, which implies that

$$\left| \mathrm{Gap}(\widehat{\vartheta}, \widehat{\omega}) - \left[ \max_{\omega \in \mathcal{X}_\Omega} \widetilde{F}(\widehat{\vartheta}, \omega) - \min_{\vartheta \in \mathcal{X}_\Theta} \widetilde{F}(\vartheta, \widehat{\omega}) \right] \right| \leq 2/T.$$
(B.11)

Since $\widetilde{F}$ defined in (B.9) have Lipschitz gradients, by Theorem C.4, we have

$$\max_{\omega \in \mathcal{X}_\Omega} \widetilde{F}(\widehat{\vartheta}, \omega) - \min_{\vartheta \in \mathcal{X}_\Theta} \widetilde{F}(\vartheta, \widehat{\omega}) \leq \frac{C_K \cdot \log^6 T}{(1 - \rho) \cdot \sqrt{T}}$$
(B.12)

with probability at least $1 - T^{-5}$, where $C_K$ is a constant that depends polynomially on $\widetilde{R}_\Theta$, $\widetilde{R}_\Omega$, $J(K_0)$, $\|K\|_{\mathrm{fro}}$, and $1/\sigma_{\min}(Q)$. Finally, combining (B.7), (B.11), and (B.12), we conclude the proof of this theorem. $\qquad \square$

# C  Proofs of the Main Results

In this section, we provide the proofs of the main results, namely, Theorems 4.2 and 4.3, which are proved in §C.1 and §C.2, respectively. The proofs of the supporting results are deferred to the appendix.

## C.1  Proof of Theorem 4.2

*Proof.* Our proof can be decomposed into three steps. In the first step, we show that, with $\mathcal{X}_\Theta$ and $\mathcal{X}_\Omega$ given in (4.1) and (4.2), $(\vartheta, \omega) = (\vartheta_K^*, 0)$ is the solution to the minimax optimization problem in (3.18). Then, in the second step, we show that the primal-dual gap of this optimization problem yields an upper bound for the estimation error $\|\widehat{\Theta} - \Theta_K\|_{\mathrm{fro}}^2$, where $\widehat{\Theta} = \mathrm{smat}(\widehat{\vartheta}^2)$ is the estimator of $\Theta_K$ returned by the GTD algorithm. Finally, in the last step, we study the performance of such a minimax optimization problem, which enables us to establish the error of policy evaluation.

**Step 1.** In the first step, we show that $(\vartheta, \omega) = (\vartheta_K^*, 0)$ is the saddle point of the optimization problem in (3.18). To simplify the notation, we define a vector-valued function $G(x, u, x', u'; \vartheta)$ by

$$
\begin{aligned}
G^1(x, u, x', u'; \vartheta) &= \vartheta^1 - c(x, u), \\
G^2(x, u, x', u'; \vartheta) &= \vartheta^1 \cdot \phi(x, u) + \left\{ \left[ \phi(x, u) - \phi(x', u') \right]^\top \vartheta^2 - c(x, u) \right\} \cdot \phi(x, u).
\end{aligned}
\tag{C.1}
$$

By definition, $G(x, u, x', u'; \vartheta)$ is of the same shape as $\vartheta$ and $\omega$. Moreover, for all $(\vartheta, \omega)$, $F(\vartheta, \omega)$ in (3.18) can be equivalently written as

$$
F(\vartheta, \omega) = \left\langle \mathbb{E}_{(x, u, x', u')}[G(x, u, x', u'; \vartheta)], \omega \right\rangle - 1/2 \cdot \|\omega\|_2^2.
\tag{C.2}
$$

Thus, for any $\vartheta$, the solution to the unconstrained maximization problem $\max_\omega F(\theta, \omega)$ is

$$
w(\vartheta) = \mathbb{E}_{(x, u, x', u')}[G(x, u, x', u'; \vartheta)].
\tag{C.3}
$$

In the following, we show that $\vartheta_K^* \in \mathcal{X}_\Theta$. Moreover, we also prove that, for any $\vartheta \in \mathcal{X}_\Theta$, $w(\vartheta)$ in (C.3) belongs to $\mathcal{X}_\Omega$, where $\mathcal{X}_\Theta$ and $\mathcal{X}_\Omega$ are defined in (4.1) and (4.2), respectively. Since $w(\vartheta_K^*) = 0$, it holds that $(\vartheta_K^*, 0)$ is the solution to the minimax optimization problem in (3.18).

Recall that we assume $J(K) \leq J(K_0)$, where $K_0$ is the initial policy that is stable. Thus, $J(K_0)$ is finite. By the definition of $\vartheta_K^*$, to show $\vartheta_K^* \in \mathcal{X}_\Theta$, it suffices to bound $\|\Theta_K\|_{\mathrm{fro}}$. By the definition of $\Theta_K$ in (3.7), we have

$$
\Theta_K = \begin{pmatrix} Q + A^\top P_K A & A^\top P_K B \\ B^\top P_K A & R + B^\top P_K B \end{pmatrix} = \begin{pmatrix} Q & \\ & R \end{pmatrix} + \begin{pmatrix} A^\top \\ B^\top \end{pmatrix} P_K \begin{pmatrix} A & B \end{pmatrix},
$$

which implies that

$$
\|\Theta_K\|_{\mathrm{fro}} \leq (\|Q\|_{\mathrm{fro}} + \|R\|_{\mathrm{fro}}) + (\|A\|_{\mathrm{fro}}^2 + \|B\|_{\mathrm{fro}}^2) \cdot \|P_K\|_{\mathrm{fro}}.
\tag{C.4}
$$

Now we apply the following lemma to obtain an upper bound on $\|P_K\|_{\mathrm{fro}}$.

**Lemma C.1.** When $\pi_K$ is a stable policy, we have

$$
\|\Sigma_K\| \leq J(K)/\sigma_{\min}(Q), \qquad \|P_K\| \leq J(K)/\sigma_{\min}(\Psi),
$$

where $\sigma_{\min}(\cdot)$ denotes the minimal eigenvalue of a matrix.

*Proof.* By (3.8) in Proposition 3.1, we have

$$
\begin{aligned}
J(K) &\geq \mathrm{tr}[(Q + K^\top R K)\Sigma_K] \geq \sigma_{\min}(Q) \cdot \mathrm{tr}(\Sigma_K) \geq \sigma_{\min}(Q) \cdot \|\Sigma_K\|, \\
J(K) &\geq \mathrm{tr}(P_K \Psi_\sigma) \geq \sigma_{\min}(\Psi_\sigma) \cdot \mathrm{tr}(P_K) \geq \|P_K\| \geq J(K)/\sigma_{\min}(\Psi),
\end{aligned}
$$

where we use the fact that $\Psi_\sigma \succeq \Psi$. Therefore, we conclude the proof. $\qquad\square$

Applying Lemma C.1 to (C.4), we have

$$
\|\Theta_K\|_{\mathrm{fro}} \leq (\|Q\|_{\mathrm{fro}} + \|R\|_{\mathrm{fro}}) + (\|A\|_{\mathrm{fro}}^2 + \|B\|_{\mathrm{fro}}^2) \cdot \sqrt{d} \cdot J(K)/\sigma_{\min}(\Psi).
\tag{C.5}
$$

Combining (C.5) and the definition of $\widetilde{R}_\Theta$ in (4.3) we conclude that $\vartheta_K^* \in \mathcal{X}_\Theta$.

Furthermore, it remains to show that the vector in (C.3) belongs to $\mathcal{X}_\Omega$ for all $\vartheta \in \mathcal{X}_\Theta$. We consider the two components of $G(x, u, x', u'; \vartheta)$ separately. By (C.1), we have

$$\left| \mathbb{E}_{(x,u,x',u')}[G^1(x, u, x', u'; \vartheta)] \right| = |\vartheta^1 - J(K)| \leq J(K_0), \tag{C.6}$$

where the second inequality follows from the fact that $0 \leq \vartheta^1 \leq J(K_0)$. Moreover, by (C.1), for the second component of $G(x, u, x', u'; \vartheta)$, we have

$$\mathbb{E}_{(x,u,x',u')}[G^2(x, u, x', u'; \vartheta)] = \vartheta^1 \cdot \mathbb{E}_{(x,u)}[\phi(x, u)] + \Xi_K \vartheta^2 - b_K, \tag{C.7}$$

where $\Xi_K$ and $b_K$ are defined in (3.15). By Lemma D.2, we have

$$\|\Xi_K \vartheta^2\|_2 \leq \|\Xi_K\| \cdot \|\vartheta^2\|_2 \leq 4(1 + \|K\|_{\mathrm{fro}}^2)^2 \cdot \|\Sigma_K\|^2 \cdot \widetilde{R}_\Theta. \tag{C.8}$$

Moreover, for any positive definite matrix $\Gamma$, we have

$$b_K^\top \operatorname{svec}(\Gamma) = \mathbb{E}_{(x,u)}\{\langle \phi(x, u), \operatorname{svec}[\operatorname{diag}(Q, R)]\rangle \cdot \langle \phi(x, u), \operatorname{smat}(\Gamma)\rangle\}, \tag{C.9}$$

where $\operatorname{diag}(Q, R)$ is the block diagonal matrix constructed by $Q$ and $R$. Note that the joint distribution of $(x, u)$ is the Gaussian distribution $N(0, \widetilde{\Sigma}_K)$, where $\widetilde{\Sigma}_K$ is defined in (D.16). Thus, $b_K^\top \operatorname{svec}(\Gamma)$ can be written as the product of two quadratic forms of Gaussian random variables. Applying Lemma D.3 to (C.9), we obtain that

$$b_K^\top \operatorname{svec}(\Gamma) = 2\langle \widetilde{\Sigma}_K \operatorname{diag}(Q, R)\widetilde{\Sigma}_K, \Gamma \rangle \cdot + \langle \widetilde{\Sigma}_K, \operatorname{diag}(Q, R)\rangle \cdot \langle \widetilde{\Sigma}_K, \Gamma \rangle,$$

which implies that

$$\|b_K\|_2 \leq 3(\|Q\|_{\mathrm{fro}} + \|R\|_{\mathrm{fro}}) \cdot \|\widetilde{\Sigma}_K\|^2. \tag{C.10}$$

In addition, the first term on the right-hand side of (C.7) is bounded by

$$\left\| \vartheta^1 \cdot \mathbb{E}_{(x,u)}[\phi(x, u)] \right\|_2 \leq J(K_0) \cdot \left\| \widetilde{\Sigma}_K \right\|_{\mathrm{fro}}. \tag{C.11}$$

Finally, combining (C.8), (C.10), (C.11), and the upper bounds in (D.17), we have

$$\left\| \mathbb{E}_{(x,u,x',u')}[G^2(x, u, x', u'; \vartheta)] \right\|_2$$
$$\leq 2(d + \|K\|_{\mathrm{fro}}^2) \cdot \|\Sigma_K\| + 4(1 + \|K\|_{\mathrm{fro}}^2)^2 \cdot \|\Sigma_K\|^2 \cdot \widetilde{R}_\Theta$$
$$\qquad + 12(\|Q\|_{\mathrm{fro}} + \|R\|_{\mathrm{fro}}) \cdot (d + \|K\|_{\mathrm{fro}}^2)^2 \cdot \|\Sigma_K\|^2$$
$$\leq C \cdot (1 + \|K\|_{\mathrm{fro}}^2)^2 \cdot \widetilde{R}_\Theta \cdot \sigma_{\min}^{-2}(Q) \cdot [J(K_0)]^2, \tag{C.12}$$

where $C > 0$ is an absolute constant.

Hence, combining (4.4), (C.6) and (C.12), we conclude that $w(\vartheta) \in \mathcal{X}_\Omega$ for all $\vartheta \in \mathcal{X}_\Theta$. Therefore, we have shown that $(\vartheta_K^*, 0)$ is the saddle point of the optimization problem in (3.18), which concludes the first step of the proof.

**Step 2.** In the following, we relate the estimation error $\|\widehat{\Theta} - \Theta_K\|_{\mathrm{fro}}^2$ to the performance of the optimization in (3.18). Specifically, we consider the primal-dual gap

$$\texttt{Gap}(\widehat{\vartheta}, \widehat{\omega}) = \max_{\omega \in \mathcal{X}_\Omega} F(\widehat{\vartheta}, \omega) - \min_{\vartheta \in \mathcal{X}_\Theta} F(\vartheta, \widehat{\omega}), \tag{C.13}$$

which characterizes the closeness between $(\widehat{\vartheta}, \widehat{\phi})$ and the optimal solution $(\vartheta_K^*, 0)$, quantified by the objective value.

Recall that $w(\vartheta)$ defined in (C.3) is the optimal dual variable for each $\theta \in \mathcal{X}_\Theta$. Hence, for any $\omega \in \mathcal{X}_\Omega$, it holds that

$$\min_{\vartheta \in \mathcal{X}_\Theta} F(\vartheta, \omega) \leq \min_{\theta \in \mathcal{X}_\Theta} \max_{\omega \in \mathcal{X}_\Omega} F(\theta, \omega)$$
$$\leq \min_{\vartheta \in \mathcal{X}_\Omega} \{[\vartheta^1 - J(K)]^2 + \|\vartheta^1 \cdot \mathbb{E}_{(x,u)}[\phi(x, u)] + \Xi_K \vartheta^2 - b_K\|_2^2\} = 0. \tag{C.14}$$

Thus, for $\widehat{\vartheta}$ returned by the GTD algorithm, we have

$$\{[\widehat{\vartheta}^1 - J(K)]^2 + \|\widehat{\vartheta}^1 \cdot \mathbb{E}_{(x,u)}[\phi(x, u)] + \Xi_K \widehat{\vartheta}^2 - b_K\|_2^2\} = \max_{\omega \in \mathcal{X}_\Omega} F(\widehat{\vartheta}, \omega)$$
$$= \max_{\omega \in \mathcal{X}_\Omega} F(\widehat{\vartheta}, \omega) - \min_{\vartheta \in \mathcal{X}_\Theta} F(\vartheta, \widehat{\omega}) + \min_{\vartheta \in \mathcal{X}_\Theta} F(\vartheta, \widehat{\omega}) \leq \texttt{Gap}(\widehat{\vartheta}, \widehat{\omega}), \tag{C.15}$$

where the last inequality follows from (C.14).

Furthermore, by direct computation, we can bound the left-hand side of (C.15) via

$$\left\| \begin{pmatrix} 1 & 0 \\ \mathbb{E}_{(x,u)}[\phi(x,u)] & \Xi_K \end{pmatrix} (\widehat{\vartheta} - \vartheta_K^*) \right\|_2^2$$
$$\geq {\kappa_K^*}^2 \cdot \|\widehat{\vartheta} - \vartheta_K^*\|_2^2 = \kappa_K^* \cdot \big[\|\widehat{\Theta} - \Theta_K\|_{\mathrm{fro}}^2 + |\widehat{\vartheta}^1 - J(K)|^2\big], \tag{C.16}$$

where we utilize the fact that $\vartheta_K^*$ is the solution to the linear equation in (3.16) and $\kappa_K^*$ is specified in Lemma 3.2. Therefore, combining (C.15) and (C.16), we have

$$|\widehat{\vartheta}^1 - J(K)|^2 + \|\widehat{\Theta} - \Theta_K\|_{\mathrm{fro}}^2 \leq {\kappa_K^*}^{-2} \cdot \texttt{Gap}(\widehat{\vartheta}, \widehat{\omega}), \tag{C.17}$$

which establishes the connection between $\|\widehat{\Theta} - \Theta_K\|_{\mathrm{fro}}^2$ and the primal-dual gap in (C.13).

**Step 3.** In the last step, we construct an upper bound for the primal-dual gap. By (C.17), this yields an upper bound for the error of parameter estimation.

Note that the distribution of the state-action pair $(x, u)$ have unbounded support. We first construct an event such that $\{\phi(x_t, u_t)\}_{t=0}^T$ are bounded conditioning on this event. To this end, we establish an upper bound for tail probability of the $\|\phi(x, u)\|_2$ using the Hansen-Wright inequality stated as follows.

**Lemma C.2** (Hansen-Wright inequality). For any integer $m > 0$, let $A$ be a matrix in $\mathbb{R}^{m \times m}$ and let $\eta \sim N(0, I_m)$ be the standard Gaussian random variable in $\mathbb{R}^m$. Then, there exists an absolute constant $C > 0$ such that, for any $t \geq 0$, we have

$$\mathbb{P}\big[\big|\eta^\top A \eta - \mathbb{E}(\eta^\top A \eta)\big| > t\big] \leq 2 \cdot \exp\big[-C \cdot \min(t^2 \cdot \|A\|_{\mathrm{fro}}^{-2}, \ t \cdot \|A\|^{-1})\big]$$

*Proof.* See [55] for a detailed proof. $\qquad\square$

Applying Lemma C.2 to $(x, u) \sim N(0, \widetilde{\Sigma}_K)$ with $\widetilde{\Sigma}_K$ defined in (D.16), we obtain

$$\mathbb{P}\big[\big|\|x\|_2^2 + \|u\|_2^2 - \mathrm{tr}(\widetilde{\Sigma}_K)\big| > t\big] \leq 2 \cdot \exp\big[-C \cdot \min\big(t^2 \cdot \|\widetilde{\Sigma}_K\|_{\mathrm{fro}}^{-2}, \ t \cdot \|\widetilde{\Sigma}_K\|^{-1}\big)\big]. \tag{C.18}$$

Setting $t = C_1 \cdot \log T \cdot \|\widetilde{\Sigma}_K\|$ in (C.18) with constant $C_1$ sufficiently large, it holds that

$$t^2 \cdot \|\widetilde{\Sigma}_K\|_{\mathrm{fro}}^{-2} = \|\widetilde{\Sigma}_K\|_{\mathrm{fro}}^{-2} \cdot C_1^2 \cdot \log^2 T \cdot \|\widetilde{\Sigma}_K\|^2 \geq C_1^2 \cdot (d+k)^{-1} \cdot \log^2 T \geq t \cdot \|\widetilde{\Sigma}_K\|^{-1}, \tag{C.19}$$

where the first inequality follows from the relation between the operator and Frobenius norms, and the second inequality holds when $\log T \geq C_1^{-1} \cdot (d+k)$. For ease of presentation, for any $t \in \{0, 1, \dots, T\}$, we define

$$\mathcal{E}_t = \Big\{\big|\|x_t\|_2^2 + \|u_t\|_2^2 - \mathrm{tr}(\widetilde{\Sigma}_K)\big| \leq C_1 \cdot \log T \cdot \|\widetilde{\Sigma}_K\|\Big\}, \tag{C.20}$$

and write $\mathcal{E} = \bigcap_{0 \leq t \leq T} \mathcal{E}_t$. Combining (C.18) and (C.19), we obtain that $\mathcal{E}_t$ holds with probability at least $1 - T^{-6}$. Thus, by taking a union bound for $\{(x_t, u_t)\}_{t=0}^T$, we have $\mathbb{P}(\mathcal{E}) \geq 1 - 2T^{-5}$. Moreover, combining (C.20) and (D.17) further implies that, on event $\mathcal{E}$, we have

$$\max_{0 \leq t \leq T} \big\{\|x_t\|_2^2 + \|u_t\|_2^2\big\} \leq C_1 \cdot \log T \cdot \|\widetilde{\Sigma}_K\| + \mathrm{tr}(\widetilde{\Sigma}_K) \leq \big(C_1 \cdot \log T + d + k\big) \cdot \|\widetilde{\Sigma}_K\|$$
$$\leq 2C_1 \cdot \log T \cdot \|\widetilde{\Sigma}_K\| \leq 2C_1 \cdot \log T \cdot \big[\sigma^2 + (1 + \|K\|_{\mathrm{fro}}^2) \cdot \|\Sigma_K\|\big]. \tag{C.21}$$

In the sequel, we study the stochastic optimization problem in (3.18) with the restriction that $\mathcal{E}$ holds. Specifically, for any state-action pair $(x, u)$, we define the truncated feature function as

$$\widetilde{\phi}(x, u) = \phi(x, u) \cdot \mathbb{1}\Big\{\big|\|\phi(x,u)\|_2^2 - \mathrm{tr}(\widetilde{\Sigma}_K)\big| \leq C_1 \cdot \log T \cdot \|\widetilde{\Sigma}_K\|\Big\}. \tag{C.22}$$

By this definition, for any $t \in \{0, \dots, t\}$, we have $\widetilde{\phi}(x_t, u_t) = \phi(x_t, u_t) \cdot \mathbb{1}_{\mathcal{E}_t}$. Now we replace $\phi(x, u)$ by $\widetilde{\phi}(x, u)$ in (3.18) and consider the following minimax optimization problem:

$$\min_{\vartheta \in \mathcal{X}_\Theta} \max_{\omega \in \mathcal{X}_\Omega} \widetilde{F}(\vartheta, \omega) = \big\langle \mathbb{E}_{(x,u,x',u')}\big[\widetilde{G}(x, u, x', u'; \vartheta)\big], \omega \big\rangle - 1/2 \cdot \|\omega\|_2^2, \tag{C.23}$$

where, similar to $G(x, u, x', u'; \vartheta)$ in (C.1), we define $\widetilde{G}(x, u, x', u'; \vartheta)$ by

$$\widetilde{G}^1(x, u, x', u'; \vartheta) = \vartheta^1 - \widetilde{c}(x, u),$$
$$\widetilde{G}^2(x, u, x', u'; \vartheta) = \vartheta^1 \cdot \widetilde{\phi}(x, u) + \left\{ \left[ \widetilde{\phi}(x, u) - \widetilde{\phi}(x', u') \right]^\top \vartheta^2 - \widetilde{c}(x, u) \right\} \cdot \widetilde{\phi}(x, u). \tag{C.24}$$

Here we denote $\widetilde{c}(x, u) = \langle \widetilde{\phi}(x, u), \mathrm{svec}[\mathrm{diag}(Q, R)] \rangle$ in (C.24) to simplify the notation.

We remark that, when $\mathcal{E}$ is true, $(\widehat{\vartheta}, \widehat{\omega})$ is also the solution returned by the gradient-based algorithm for the minimax optimization problem in (C.23). As a result, when $\mathcal{E}$ holds, the primal-dual gap of (C.23) is equal to $\max_{\omega \in \mathcal{X}_\Omega} \widetilde{F}(\widehat{\vartheta}, \omega) - \min_{\vartheta \in \mathcal{X}_\Theta} \widetilde{F}(\vartheta, \widehat{\omega})$.

In the following, we characterize the difference between the objective functions in (3.18) and (C.23). For any $(\vartheta, \omega) \in \mathcal{X}_\Theta \times \mathcal{X}_\Omega$, by (C.2) and (C.23) we have

$$\left| F(\vartheta, \omega) - \widetilde{F}(\vartheta, \omega) \right| = \left| \langle \mathbb{E}_{(x, u, x', u')} [ G(x, u, x', u'; \vartheta) - \widetilde{G}(x, u, x', u'; \vartheta) ], \omega \rangle \right|$$
$$\leq \left| \mathbb{E}_{(x, u, x', u')} [ G^1(x, u, x', u'; \vartheta) - \widetilde{G}^1(x, u, x', u'; \vartheta) ] \right| \cdot J(K_0)$$
$$+ \left\| \mathbb{E}_{(x, u, x', u')} [ G^2(x, u, x', u'; \vartheta) - \widetilde{G}^2(x, u, x', u'; \vartheta) ] \right\|_2 \cdot \widetilde{R}_\Omega. \tag{C.25}$$

By the definitions of $G(x, u, x', u'; \vartheta)$ and $\widetilde{G}(x, u, x', u'; \vartheta)$ in (C.1) and (C.24), we have

$$G^1(x, u, x', u'; \vartheta) - \widetilde{G}^1(x, u, x', u'; \vartheta) = c(x, u) \cdot \mathbb{1}_{\mathcal{A}^c} \tag{C.26}$$
$$G^2(x, u, x', u'; \vartheta) - \widetilde{G}^2(x, u, x', u'; \vartheta) = G^2(x, u, x', u'; \vartheta) \cdot \mathbb{1}_{\mathcal{A}^c} + \phi(x', u')^\top \vartheta^2 \cdot \phi(x, u) \cdot \mathbb{1}_{\mathcal{A}} \cdot \mathbb{1}_{\mathcal{B}^c},$$

where we denote $\{ | \|\phi(x, u)\|_2^2 - \mathrm{tr}(\widetilde{\Sigma}_K) | \leq C_1 \cdot \log T \cdot \|\widetilde{\Sigma}_K\| \}$ and $\{ | \|\phi(x, u)\|_2^2 - \mathrm{tr}(\widetilde{\Sigma}_K) | \leq C_1 \cdot \log T \cdot \|\widetilde{\Sigma}_K\| \}$ by $\mathcal{A}$ and $\mathcal{B}$, respectively, and $\mathcal{A}^c$, $\mathcal{B}^c$ are the complement sets of $\mathcal{A}$ and $\mathcal{B}$.

For the first term on the right-hand side of (C.25), Cauchy-Schwarz inequality implies that

$$\left| \mathbb{E}_{(x, u, x', u')} [ G^1(x, u, x', u'; \vartheta) - \widetilde{G}^1(x, u, x', u'; \vartheta) ] \right| \leq \sqrt{\mathbb{P}(\mathcal{A}^c)} \cdot \sqrt{\mathbb{E}[c^2(x, u)]}. \tag{C.27}$$

Since $c(x, u)$ is a quadratic form of a Gaussian random variable, by Lemma D.3, we have

$$\mathbb{E}[c^2(x, u)] = 2 \mathrm{tr} \left[ \widetilde{\Sigma}_K \mathrm{diag}(Q, R) \widetilde{\Sigma}_K \mathrm{diag}(Q, R) \right] + \left\{ \mathrm{tr} \left[ \widetilde{\Sigma}_K \mathrm{diag}(Q, R) \right] \right\}^2$$
$$\leq 3 (\|Q\|_{\mathrm{fro}} + \|R\|_{\mathrm{fro}})^2 \cdot \|\widetilde{\Sigma}_K\|_{\mathrm{fro}}^2 \leq 3 (\|Q\|_{\mathrm{fro}} + \|R\|_{\mathrm{fro}})^2 \cdot \left[ \sigma^2 \cdot k + (d + \|K\|_{\mathrm{fro}}^2)^2 \cdot \|\Sigma_K\|^2 \right],$$

where the last inequality follows from (D.17). Besides, for the second term on the right-hand side of (C.25), combining (C.25), (C.26), triangle inequality, and Cauchy-Schwarz inequality, we have

$$\left\| \mathbb{E}_{(x, u, x', u')} [ G^2(x, u, x', u'; \vartheta) - \widetilde{G}^2(x, u, x', u'; \vartheta) ] \right\|_2$$
$$\leq \left\{ \left\| \mathbb{E}_{(x, u, x', u')} [ G^2(x, u, x', u'; \vartheta) \cdot \mathbb{1}_{\mathcal{A}^c} ] \right\|_2 + \left\| \mathbb{E}_{(x, u, x', u')} [ \phi(x', u')^\top \vartheta^2 \cdot \phi(x, u) \, \mathbb{1}_{\mathcal{B}^c} ] \right\|_2 \right\}$$
$$\leq \left\{ \sqrt{\mathbb{P}(\mathcal{A}^c)} \cdot \sqrt{\mathbb{E} [ \| G^2(x, u, x', u'; \vartheta) \|_2^2 ]} + \sqrt{\mathbb{P}(\mathcal{B}^c)} \cdot \sqrt{\mathbb{E} [ \| \phi(x, u) \cdot \phi(x', u')^\top \vartheta^2 \|_2^2 ]} \right\}. \tag{C.28}$$

For the expectations on the right-hand side of (C.28), using the inequality $(a + b)^2 \leq 2a^2 + 2b^2$, we have

$$\mathbb{E} [ \| G^2(x, u, x', u'; \vartheta) \|_2^2 ]$$
$$\leq 2 \cdot \mathbb{E} \left\{ \left[ \vartheta^1 - c(x, u) + \phi(x, u)^\top \vartheta^2 \right]^2 \cdot \|\phi(x, u)\|_2^2 \right\} + 2 \cdot \mathbb{E} [ \| \phi(x, u) \cdot \phi(x', u')^\top \vartheta^2 \|_2^2 ]. \tag{C.29}$$

Further applying Cauchy-Schwarz inequality to (C.29), we have

$$\mathbb{E} \left\{ \left[ \vartheta^1 - c(x, u) + \phi(x, u)^\top \vartheta^2 \right]^2 \cdot \|\phi(x, u)\|_2^2 \right\}$$
$$\leq \left( \mathbb{E} \left\{ \left[ \vartheta^1 - c(x, u) + \phi(x, u)^\top \vartheta^2 \right]^4 \right\} \cdot \mathbb{E} [ \|\phi(x, u)\|_2^4 ] \right)^{1/2}, \tag{C.30}$$

$$\mathbb{E} [ \| \phi(x, u) \cdot \phi(x', u')^\top \vartheta \|_2^2 ] \leq \left( \mathbb{E} [ | \phi(x', u')^\top \vartheta |^4 ] \cdot \mathbb{E} [ \|\phi(x, u)\|_2^4 ] \right)^{1/2}. \tag{C.31}$$

Since the marginal distributions of $(x, u)$ and $(x', u')$ are both $N(0, \widetilde{\Sigma}_K)$, in (C.30) and (C.31) we bound the two terms in (C.29) using the fourth moments of $N(0, \widetilde{\Sigma}_K)$, which can be written as a polynomial of $J(K_0)$, $\|K\|_{\mathrm{fro}}$, $\|Q\|$, $\|R\|$, $\widetilde{R}_\Theta$, and $\widetilde{R}_\Omega$.

Meanwhile, recall that we have shown that $\mathbb{P}(\mathcal{A}^c) \leq T^{-6}$ and $\mathbb{P}(\mathcal{B}^c) \leq T^{-6}$. Thus, when $T$ is sufficiently large, by combining (C.25), (C.27), (C.28), and (C.29), we have $|F(\vartheta, \omega) - \widetilde{F}(\vartheta, \omega)| \leq 1/T$, which implies that

$$\left| \mathtt{Gap}(\widehat{\vartheta}, \widehat{\omega}) - \left[ \max_{\omega \in \mathcal{X}_\Omega} \widetilde{F}(\widehat{\vartheta}, \omega) - \min_{\vartheta \in \mathcal{X}_\Theta} \widetilde{F}(\vartheta, \widehat{\omega}) \right] \right|$$

$$\leq \max_{\omega \in \mathcal{X}_\Omega} \left| F(\widehat{\vartheta}, \omega) - \widetilde{F}(\widehat{\vartheta}, \omega) \right| + \max_{\vartheta \in \mathcal{X}_\Theta} \left| F(\vartheta, \widehat{\omega}) - \widetilde{F}(\vartheta, \widehat{\omega}) \right| \leq \frac{2}{T}. \qquad \text{(C.32)}$$

Hereafter, we study the primal-dual gap in (C.13) conditioning on event $\mathcal{E}$. To simplify the notation, we define function $H(\vartheta, \omega; \phi, \phi')$ on $\mathcal{X}_\Theta \times \mathcal{X}_\Omega$ by

$$H(\vartheta, \omega; \phi, \phi') = \langle \widetilde{G}(x, u, x', u'; \vartheta), \omega \rangle - 1/2 \cdot \|\omega\|_2^2,$$

where the function $\widetilde{\phi}(x, u)$ is defined in (C.22), and we denote $\widetilde{\phi}(x, u)$ and $\widetilde{\phi}(x', u')$ by $\phi$ and $\phi'$, respectively. Using this definition, the objective function $\widetilde{F}(\vartheta, \omega)$ in (C.23) can be written as $\widetilde{F}(\vartheta, \omega) = \mathbb{E}_{(x, u, x', u')}[H(\vartheta, \omega; \phi, \phi')]$, where $(x, u)$ and $(x', u')$ are two consecutive state-action pairs. Note that $H(\vartheta, \omega; \phi, \phi')$ is a quadratic function of $(\vartheta, \omega)$ for all $\phi$ and $\phi'$. The partial gradients of $H(\vartheta, \omega; \phi, \phi')$ are given by

$$\nabla_{\vartheta^1} H(\vartheta, \omega; \phi, \phi') = \omega^1 + \widetilde{\phi}(x, u)^\top \omega^2, \qquad \text{(C.33)}$$

$$\nabla_{\vartheta^2} H(\vartheta, \omega; \phi, \phi') = [\widetilde{\phi}(x, u)^\top \omega^2] \cdot [\widetilde{\phi}(x, u) - \widetilde{\phi}(x', u')], \qquad \text{(C.34)}$$

$$\nabla_{\omega^1} H(\vartheta, \omega; \phi, \phi') = \vartheta^1 - \widetilde{c}(x, u) - \omega^1, \qquad \text{(C.35)}$$

$$\nabla_{\omega^2} H(\vartheta, \omega; \phi, \phi') = \widetilde{G}^2(x, u, x', u'; \vartheta) - \omega^2. \qquad \text{(C.36)}$$

By combining (C.22), (C.33), and (C.34), we can bound the norm of $\nabla_\vartheta H(\vartheta, \omega; \phi, \phi')$ by

$$\left\| \nabla_\vartheta H(\vartheta, \omega; \phi, \phi') \right\|_2 \leq |\omega^1 + \widetilde{\phi}(x, u)^\top \omega^2| + \left\| [\widetilde{\phi}(x, u)^\top \omega^2] \cdot [\widetilde{\phi}(x, u) - \widetilde{\phi}(x', u')] \right\|_2 \qquad \text{(C.37)}$$

$$\leq |\omega^1| + 2\|\widetilde{\phi}(x, u)\|_2 \cdot \|\omega^2\|_2 \cdot \left[ \|\widetilde{\phi}(x, u)\|_2 + \|\widetilde{\phi}(x', u')\|_2 \right]$$

$$\leq J(K_0) + 16 C_1^2 \cdot (1 + \|K\|_{\mathrm{fro}}^2)^2 \cdot \log^2 T \cdot \left[ \sigma^2 + (1 + \|K\|_{\mathrm{fro}}^2) \cdot \|\Sigma_K\| \right]^2 \cdot \widetilde{R}_\Omega.$$

Here the second inequality holds when $\|\widetilde{\phi}(x, u)\|_2 \geq 1$ and the last inequality follows from (C.21). Similarly, combining triangle inequality, (C.35), and (C.36), we have

$$\left\| \nabla_\omega H(\vartheta, \omega; \phi, \phi') \right\|_2 \leq |\vartheta^1 - \widetilde{c}(x, u) - \omega^1| + + \left[ (\|Q\|_{\mathrm{fro}} + \|R\|_{\mathrm{fro}}) \cdot \|\widetilde{\phi}(x, u)\|_2 \right.$$

$$+ \left. (\|\widetilde{\phi}(x', u')\|_2 + \|\widetilde{\phi}(x, u)\|_2) \cdot \widetilde{R}_\Theta \right] \cdot \|\widetilde{\phi}(x, u)\|_2$$

$$\leq 2 J(K_0) + 16 C_1^2 \cdot \log^2 T \cdot \left[ \sigma^2 + (1 + \|K\|_{\mathrm{fro}}^2) \cdot \|\Sigma_K\| \right]^2 \cdot \widetilde{R}_\Theta. \qquad \text{(C.38)}$$

where the last equality holds since $\widetilde{R}_\Theta \geq \|Q\|_{\mathrm{fro}} + \|R\|_{\mathrm{fro}}$. Moreover, we have $\nabla^2_{\vartheta\vartheta} H(\vartheta, \omega; \phi, \phi') = 0$ and $-\nabla^2_{\omega\omega} H(\vartheta, \omega; \phi, \phi')$ is the identity matrix.

We utilize the following lemma, obtained from [69], to handle the dependence along the trajectory.

**Lemma C.3** (Geometrically $\beta$-mixing)**.** Consider a linear dynamical system $X_{t+1} = L X_t + \varepsilon$, where $\{X_t\}_{t \geq 0} \subseteq \mathbb{R}^m$, $\varepsilon \sim N(0, \Psi)$ is the random noise, and $L \in \mathbb{R}^{m \times m}$ has spectral radius smaller than one. We denote by $\nu_t$ the marginal distribution of $X_t$ for all $t \geq 0$. Besides, the stationary distribution of this Markov chain is denoted by $N(0, \Sigma_\infty)$. For any integer $k \geq 1$, we define the $k$-th mixing coefficient as

$$\beta(k) = \sup_{t \geq 0} \mathbb{E}_{x \sim \nu_t} \left[ \left\| \mathbb{P}_{X_k}(\cdot \mid X_0 = x) - \mathbb{P}_{N(0, \Sigma_\infty)}(\cdot) \right\|_{\mathrm{TV}} \right].$$

Furthermore, for any $\rho \in (\rho(L), 1)$ and any $k \geq 1$, we have

$$\beta(k) \leq C_{\rho, L} \cdot \left[ \mathrm{tr}(\Sigma_\infty) + m \cdot (1 - \rho)^{-2} \right]^{1/2} \cdot \rho^k,$$

where $C_{\rho, L}$ is a constant that solely depends on $\rho$ and $A$. That is, $\{X_t\}_{t \geq 0}$ is geometrically $\beta$-mixing.

*Proof.* See Proposition 3.1 in [69] for a detailed proof. $\qquad\square$

Recall that under policy $\pi_K$, $\{(x_t, u_t)\}_{t \geq 0}$ form a linear dynamic system characterized by (D.13) and (D.14). Since $\rho(L) = \rho(A - BK) < 1$, Lemma C.3 implies that, for all $\rho \in (\rho(A - BK), 1)$, $(x_t, u_t)_{t \geq 0}$ is a geometrically $\beta$-mixing stochastic process with parameter $\rho$. The following theorem, adapted from Theorem 1 in [72], establishes the primal-dual gap for a convex-concave minimax optimization problem involving a geometrically $\beta$-mixing stochastic process.

**Theorem C.4** (Primal-dual gap for minimax optimization)**.** Let $\mathcal{X}$ and $\mathcal{Y}$ are bounded and closed convex sets such that $\|x - x'\|_2 \leq D$ for all $x, x' \in \mathcal{X}$ and $\|y - y'\|_2 \leq D$ for all $y, y' \in \mathcal{Y}$. Consider the gradient algorithm for stochastic minimax optimization problem

$$\min_{x \in \mathcal{X}} \max_{y \in \mathcal{Y}} F(x, y) = \mathbb{E}_{\xi \sim \pi_\xi}[\Phi(x, y; \xi)], \tag{C.39}$$

where $\xi$ is a random variable with distribution $\pi_\xi$ and $F(x, y)$ is convex in $x$ and concave in $y$. In addition, we assume that $\pi_\xi$ is the stationary distribution of a Markov chain $\{\xi_t\}_{t \geq 0}$ which is geometrically $\beta$-mixing with parameter $\rho \in (0, 1)$. Specifically, we assume that there exists a constant $C_\xi > 0$ such that, for all $k \geq 1$, the $k$-th mixing coefficient satisfy $\beta(k) \leq C_\xi \cdot \rho^k$. Furthermore, we consider the case where, almost surely for every $\xi \sim \pi_\xi$, $\Phi(x, y; \xi)$ is $L_1$-Lipschitz in both $x$ and $y$, $\nabla_x \Phi(x, y; \xi)$ is $L_2$-Lipschitz in $x$ for all $y \in \mathcal{Y}$, and $\nabla_y \Phi(x, y; \xi)$ is $L_2$-Lipschitz in $y$ for all $x \in \mathcal{X}$. Here, without loss of generality, we assume that $D, L_1, L_2 > 1$. Consider solving the optimization problem in (C.39) via $T$ iterations of the gradient-based updates

$$x_t = \Pi_{\mathcal{X}}\big[x_{t-1} - \alpha_t \nabla_x \Phi(x_{t-1}, y_{t-1}; \xi_{t-1})\big], \qquad y_t = \Pi_{\mathcal{Y}}\big[y_{t-1} + \alpha_t \cdot \nabla_y \Phi(x_{t-1}, y_{t-1}; \xi_{t-1})\big],$$

where $t \in [T]$, $\Pi_{\mathcal{X}}$ and $\Pi_{\mathcal{Y}}$ are projection operators, and $\{\alpha_t = \alpha/\sqrt{t}\}_{t \in [T]}$ are the stepsizes, where $\alpha > 0$ is a constant. Let

$$\widehat{x} = \frac{\sum_{t \in [T]} \alpha_t \cdot x_t}{\sum_{t \in [T]} \alpha_t}, \qquad \widehat{y} = \frac{\sum_{t \in [T]} \alpha_t \cdot y_t}{\sum_{t \in [T]} \alpha_t}$$

be the final output of the algorithm. Then, there exists an absolute constant $C > 0$ such that, for any $\delta \in (0, 1)$, with probability at least $1 - \delta$, the primal-dual gap satisfies

$$\max_{y \in \mathcal{Y}} F(\widehat{x}, y) - \min_{x \in \mathcal{X}} F(x, \widehat{y}) \leq \frac{C \cdot (D^2 + L_1^2 + L_1 L_2 D)}{\log(1/\rho)} \cdot \frac{\log^2 T + \log(1/\delta)}{\sqrt{T}} + \frac{C \cdot C_\xi L_1 D}{T}.$$

*Proof.* This theorem follows from Theorem 1 in [72], where we set $\alpha_t = \alpha/\sqrt{t}$ for all $t \geq 1$, and focus on the case where $\{\xi_t\}_{t \geq 0}$ is geometrically $\beta$-mixing. Under the mixing assumption, for any $k \geq 1$, the $k$-th mixing coefficient of $\{\xi_t\}_{t \geq 0}$ satisfies $\beta(k) \leq C_\xi \cdot \rho^k$. Then, for any $\delta, \eta \in (0, 1)$, Theorem 1 in [72] implies

$$\max_{y \in \mathcal{Y}} F(\widehat{x}, y) - \min_{x \in \mathcal{X}} F(x, \widehat{y}) \leq \left(\sum_{t=1}^{T} \alpha_t\right)^{-1} \left(A_0 + A_1 \cdot \eta \cdot \sum_{t=1}^{T} \alpha_t + A_2 \sum_{t=1}^{T} \alpha_t^2 + \tag{C.40}\right.$$

$$\left. 16 D L_1 \cdot \left\{ 2\tau(\eta) \cdot \log[\tau(\eta)/\delta] \cdot \left[\sum_{t=1}^{T} \alpha_t^2 + \tau(\eta) \cdot \alpha_1\right]\right\}^{1/2}\right),$$

where we define $\tau(\eta) = \log(\eta/C_\xi)/\log(\rho)$ and denote

$$A_0 = D^2 + 12 D \cdot \alpha_1 \cdot \tau(\eta) \qquad A_1 = 4 L_1 D \qquad A_2 = 10 L_1^2 + (24 L_1^2 + 8 L_1 L_2 D) \cdot \tau(\eta).$$

Now we set $\alpha_t = \alpha/\sqrt{t}$ and $\eta = C_\xi/T$ in (C.40), which implies that $\tau(\eta) = \log T / \log(1/\rho)$. Moreover, note that for all $T \geq 1$, we have $2\sqrt{T+1} - 2 \leq \sum_{t=1}^{T} 1/\sqrt{t} \leq 2\sqrt{T} - 1$ and $\sum_{t=1}^{T} 1/t \leq \log T + 1$. The last term on the right-hand side of (C.40) can be upper bounded by

$$16 D L_1 \cdot \left\{ 2 \log T / \log(1/\rho) \cdot \log[\tau(\eta)/\delta] \cdot \left[\log T + 1 + \alpha \cdot \log T / \log(1/\rho)\right]\right\}^{1/2}$$

$$\leq 16 D L_1 \cdot \left\{ 2 \log T / \log(1/\rho) \cdot [\log\log T + \log(1/\delta)] \cdot \left[\log T + 1 + \alpha \cdot \log T / \log(1/\rho)\right]\right\}^{1/2}$$

$$\leq C \cdot D L_1 \cdot \log T / \log(1/\rho) \cdot \sqrt{\log\log T + \log(1/\delta)}, \tag{C.41}$$

where $C$ is an absolute constant. Moreover, for the first three terms, we have

$$A_0 = D^2 + 12D \cdot \alpha \cdot \log T / \log(1/\rho) \leq C \cdot D^2 \log T / \log(1/\rho), \quad A_1 \cdot \eta \leq C \cdot C_\xi L_1 D / T,$$
(C.42)

$$A_2 \cdot \sum_{t=1}^{T} \alpha_t^2 \leq \left[ 10L_1^2 + (24L_1^2 + 8L_1L_2D) \cdot \log T / \log(1/\rho) \right] \cdot (\log T + 1)$$

$$\leq C \cdot [L_1^2 + L_1 L_2 D] \cdot \log^2 T / \log(1/\rho).$$
(C.43)

Thus, combining (C.40), (C.41), (C.42), and (C.43), we obtain that

$$\max_{y \in \mathcal{Y}} F(\widehat{x}, y) - \min_{x \in \mathcal{X}} F(x, \widehat{y}) \leq C \cdot \left[ (D^2 + L_1^2 + L_1 L_2 D) / \log(1/\rho) \cdot \log T \cdot \log(T/\delta) / \sqrt{T} + C_\xi L_1 D / T \right],$$

which concludes the proof of Theorem C.4. □

In order to apply Theorem C.4 to the minimax optimization in (C.23), we only need to specify parameters $C_\xi$, $D$, $L_1$, and $L_2$. First, for any $\rho \in (\rho(A - BK), 1)$, by Lemma C.3, we can set

$$C_\xi = C_{\rho,L} \cdot \left[ \text{tr}(\widetilde{\Sigma}_K) + (d+k) \cdot (1-\rho)^2 \right]^{1/2}$$

$$\leq 2C_{\rho,L} \cdot \sqrt{d+k} \cdot \left\{ \left[ \sigma^2 + (1 + \|K\|_{\text{fro}}^2) \cdot \|\Sigma_K\| \right]^{1/2} + (1-\rho)^{-1} \right\}.$$
(C.44)

Moreover, by the definitions of $\mathcal{X}_\Theta$ and $\mathcal{X}_\Omega$ in (4.1) and (4.2), respectively, we can set $D$ by

$$D^2 = 2[J(K_0)]^2 + \widetilde{R}_\Theta^2 + (1 + \|K\|_{\text{fro}}^2)^4 \cdot \widetilde{R}_\Omega^2.$$
(C.45)

Moreover, by (C.37), (C.38), and the form of $\nabla^2 G(\theta, \omega; \phi, \phi')$, we have

$$L_1 \leq 16C_1^2 \cdot \log^2 T \cdot \left[ \sigma^2 + (1 + \|K\|_{\text{fro}}^2) \cdot \|\Sigma_K\| \right]^2 \cdot \left[ (1 + \|K\|_{\text{fro}}^2)^2 \cdot \widetilde{R}_\Omega + \widetilde{R}_\Theta \right], \quad L_2 = 1.$$
(C.46)

Combining Theorem C.4, (C.44), (C.45), and (C.46), we to obtain an upper bound for the primal-dual gap in (C.13). Specifically, for any $\rho \in (\rho(A - BK), 1)$ and any $\delta \in (0, 1)$, with probability at least $1 - \delta$, the primal-dual gap of the optimization problem in (C.23) is bounded by

$$C \cdot \log^4 T \cdot \left[ \sigma^2 + (1 + \|K\|_{\text{fro}}^2) \cdot \|\Sigma_K\| \right]^4 \cdot \left[ (1 + \|K\|_{\text{fro}}^2)^2 \cdot \widetilde{R}_\Omega + \widetilde{R}_\Theta \right]^2$$

$$\cdot \left( \frac{\log^2 T + \log(1/\delta)}{\log(1/\rho) \cdot \sqrt{T}} + \frac{\sqrt{d+k}}{(1-\rho) \cdot T} \right).$$
(C.47)

where $C > 0$ is an absolute constant. Besides, we note that $\sigma$ is a constant and that $\|\Sigma_K\| \geq \sigma_{\min}(\Psi) > 0$. Finally, recall that, when event $\mathcal{E}$ holds, the primal-dual gap is equal to $\max_{\omega \in \mathcal{X}_\Omega} \widetilde{F}(\widehat{\vartheta}, \omega) - \min_{\vartheta \in \mathcal{X}_\Theta} \widetilde{F}(\vartheta, \widehat{\omega})$. Combining (C.32), (C.47) with $\delta = T^{-5}$, and the fact that $\mathbb{P}(\mathcal{E}) \geq 1 - 2T^{-5}$, we conclude that

$$\text{Gap}(\widehat{\vartheta}, \widehat{\omega}) \leq C \cdot \log^4 T \cdot (1 + \|K\|_{\text{fro}}^2)^4 \cdot \|\Sigma_K\|^4 \cdot \left[ (1 + \|K\|_{\text{fro}}^2)^2 \cdot \widetilde{R}_\Omega + \widetilde{R}_\Theta \right]^2$$

$$\cdot \left( \frac{\log^2 T + \log(T^5)}{\log(1/\rho) \cdot \sqrt{T}} + \frac{\sqrt{d+k}}{(1-\rho) \cdot T} \right) + \frac{2}{T}$$

$$\leq C \cdot (1 + \|K\|_{\text{fro}}^2)^4 \cdot \|\Sigma_K\|^4 \cdot \left[ (1 + \|K\|_{\text{fro}}^2)^2 \cdot \widetilde{R}_\Omega + \widetilde{R}_\Theta \right]^2 \cdot \frac{\log^6 T}{(1-\rho) \cdot \sqrt{T}}$$
(C.48)

holds with probability at least $1 - 3T^{-5} \geq 1 - T^{-4}$, where in the second inequality we use the fact that $1 - 1/x < \log x < x + 1$ holds for all $x > 0$, which implies that $1/\log(1/\rho) \leq 1/(1-\rho)$. This further implies that the first term on the right-hand side of the first inequality dominates the second term. The upper bound of $\text{Gap}(\widehat{\vartheta}, \widehat{\omega})$ in (C.48) concludes the last step of our proof. Finally, combining (C.17) and (C.48), we complete the proof of Theorem 4.2. □

## C.2 Proof of Theorem 4.3

*Proof.* Our proof of the global convergence can be decomposed into two steps. In the first step, similar to the analysis in [26], we study the geometry of the average return $J(K)$, as a function of $K$. Specifically, we show that $J(K)$ is gradient dominated [51]. Note that we study the ergodic setting with system noise and stochastic policies. In contrast, [26] study the case where both the transition and the policy are deterministic. Thus, their analysis of the geometry of $J(K)$ cannot be directly applied to our problem. Motivated by their analysis, we follow the similar approach to with modifications for our setting. In addition, in the second step, we utilize the geometry of $J(K)$ to show the global convergence of the actor-critic algorithm. Specifically, combining Theorem 4.2, we show that, with high probability, Algorithm 1 constructs a sequence of policies that converges linearly to the optimal policy $\pi_{K^*}$.

**Step 1.** As shown in (3.8) in Proposition 3.1, we can write $J(K)$ as

$$J(K) = \mathrm{tr}(P_K \Psi_\sigma) + \sigma^2 \cdot \mathrm{tr}(R) = \mathbb{E}_{x \in N(0, \Psi_\sigma)}(x^\top P_K x) + \sigma^2 \cdot \mathrm{tr}(R).$$

In the following lemma, for two policies $\pi_K$ and $\pi_{K'}$, we bound the difference between $x^\top P_K x$ and $x^\top P_{K'} x$. Then, taking expectation with respect to $x \in N(0, \Psi_\sigma)$ yields the difference between $J(K)$ and $J(K')$.

**Lemma C.5.** Let $K$ and $K'$ be two stable policies such that both $\rho(A - BK)$ and $\rho(A - BK')$ are smaller than one. For any $x \in \mathbb{R}^d$, let $\{x'_t\}_{t \geq 0} \subseteq \mathbb{R}^d$ be the sequence of states satisfying $x'_0 = x$ and $x'_{t+1} = (A - BK')x'_t$ for all $t \geq 0$. Then it holds that

$$x^\top P_{K'} x - x^\top P_K x = \sum_{t \geq 0} A_{K,K'}(x'_t),$$

where the function $A_{K,K'} \colon \mathbb{R}^d \to \mathbb{R}^d$ is defined as

$$A_{K,K'}(x) = 2x^\top (K' - K)^\top E_K x + x^\top (K' - K)^\top (R + B^\top P_K B)(K' - K)x.$$

*Proof.* Note that both $P_K$ and $P_{K'}$ satisfy the Bellman equation specified in (3.4). Moreover, using the operator $\mathcal{T}_K^\top$ defined in (D.3), we have $P_{K'} = \mathcal{T}_{K'}^\top (Q + K'^\top R K')$, which is equivalent to

$$x^\top P_{K'} x = \sum_{t \geq 0} x^\top [(A - BK')^t]^\top \big(Q + K'^\top R K'\big)[(A - BK')^t]x. \tag{C.49}$$

By the construction in Lemma (C.5), for all $t \geq 0$, we have $(A - BK')^t x = x'_t$. Thus, by (C.49) we have

$$x^\top P_{K'} x = \sum_{t \geq 0} x'^\top_t \big(Q + K'^\top R K'\big)x'_t = \sum_{t \geq 0} \big(x'^\top_t Q x'_t + u'^\top_t R u'_t\big), \tag{C.50}$$

where we define $u'_t = -K' x'_t$ for all $t \geq 0$. Thus, by (C.50), we have the following telescoping sum:

$$x^\top P_{K'} x - x^\top P_K x = \sum_{t \geq 0} \big[(x'^\top_t Q x'_t + u'^\top_t R u'_t) + x'^\top_t P_K x'_t - x'^\top_t P_K x'_t\big] - x'^\top_0 P_K x'_0$$

$$= \sum_{t \geq 0} \big[(x'^\top_t Q x'_t + u'^\top_t R u'_t) + x'^\top_{t+1} P_K x'_{t+1} - x'^\top_t P_K x'_t\big]. \tag{C.51}$$

Thus, in (C.51) we write $x^\top P_{K'} x - x^\top P_K x$ as a summation where each term can be written as a quadratic function of $x_t$. To further simplify (C.51), for any $x \in \mathbb{R}^d$, we have

$$x^\top Q x + (-K'x)^\top R(-K'x) + [(A - BK')x]^\top P_K [(A - BK')x] - x^\top P_K x \tag{C.52}$$

$$= x^\top \big[Q + (K' - K + K)^\top R(K' - K + K)\big]x +$$

$$\quad x^\top \big[A - BK - B(K' - K)\big]^\top P_K \big[A - BK - B(K' - K)\big]x - x^\top P_K x$$

$$= 2x^\top (K' - K)^\top \big[(R + B^\top P_K B)K - B^\top P_K A\big]x + x^\top (K' - K)^\top (R + B^\top P_K B)(K' - K)x.$$

$$= 2x^\top (K' - K)^\top E_K x + x^\top (K' - K)^\top (R + B^\top P_K B)(K' - K)x,$$

where $E_K = (R + B^\top P_K B)K - B^\top P_K A$. Finally, combining (C.51) and (C.52), we complete the proof of this lemma. $\qquad\square$

In the following lemma, we utilize Lemma C.5 to show that $J(K)$ is gradient dominated.

**Lemma C.6** (Gradient domination of $J(K)$)**.** Let $K^*$ be an optimal policy. Suppose $K$ has finite cost. Then, it holds that

$$\sigma_{\min}(\Psi) \cdot \|R + B^\top P_K B\|^{-1} \cdot \operatorname{tr}(E_K^\top E_K) \leq J(K) - J(K^*)$$
$$\leq 1/\sigma_{\min}(R) \cdot \|\Sigma_{K^*}\| \cdot \operatorname{tr}(E_K^\top E_K). \qquad (C.53)$$

*Proof.* For the upper bound in (C.53), bu (3.8) we obtain that

$$J(K) - J(K^*) = \operatorname{tr}[(P_K - P_K^*)\Psi_\sigma] = \mathbb{E}_{x \sim N(0, \Psi_\sigma)}[x^\top (P_K - P_K^*)x], \qquad (C.54)$$

where $\Psi_\sigma = \Psi + \sigma^2 BB^\top$ does not involve $K$ or $K^*$. Applying Lemma C.5 to (C.54) with $K' = K^*$, we have

$$J(K) - J(K^*) = -\mathbb{E}_{x_0^* \sim N(0, \Psi_\sigma)}\left[\sum_{t \geq 0} A_{K, K^*}(x_t^*)\right], \qquad (C.55)$$

where we define $x_t^* = (A - BK^*)^t x_0^*$ for all $t \geq 0$. Besides, by direct computation, we have

$$\mathbb{E}_{x_0^* \sim N(0, \Psi_\sigma)}\left[\sum_{t \geq 0} x_t^*(x_t^*)^\top\right]$$
$$= \mathbb{E}_{x \sim N(0, \Psi_\sigma)}\left\{\sum_{t \geq 0}(A - BK^*)^t xx^\top [(A - BK^*)^t]^\top\right\} = \mathcal{T}_{K^*}(\Psi_\sigma) = \Sigma_{K^*}, \qquad (C.56)$$

where the operator $\mathcal{T}_K$ is defined in (D.3).

Meanwhile, by the definition of $A_{K, K'}$, for any $x \in \mathbb{R}^d$, by completing the squares we have

$$A_{K, K'}(x) = 2x^\top (K' - K)^\top E_K x + x^\top (K' - K)^\top (R + B^\top P_K B)(K' - K)x$$
$$= \operatorname{tr}\left\{xx^\top [K' - K + (R + B^\top P_K B)^{-1} E_K]^\top (R + B^\top P_K B)[K' - K + (R + B^\top P_K B)^{-1} E_K]\right\}$$
$$- \operatorname{tr}[xx^\top E_K^\top (R + B^\top P_K B)^{-1} E_K]$$
$$\geq -\operatorname{tr}[xx^\top E_K^\top (R + B^\top P_K B)^{-1} E_K], \qquad (C.57)$$

where the equality is attained by $K' = K - (R + B^\top P_K B)^{-1} E_K$.

Thus, combining (C.55), (C.56), and (C.57), we obtain that

$$J(K) - J(K^*) \leq \operatorname{tr}[\Sigma_{K^*} E_K^\top (R + B^\top P_K B)^{-1} E_K] \leq \|\Sigma_{K^*}\| \cdot \operatorname{tr}[\Sigma_{K^*} E_K^\top (R + B^\top P_K B)^{-1} E_K]$$
$$\leq \|\Sigma_{K^*}\| \cdot \|(R + B^\top P_K B)^{-1}\| \cdot \operatorname{tr}(E_K^\top E_K). \qquad (C.58)$$

Notice that $R + B^\top P_K B \succeq R$ implies $(R + B^\top P_K B)^{-1} \preceq R^{-1}$. Therefore, by (C.58) we obtain that $J(K) - J(K^*) \leq 1/\sigma_{\min}(R) \cdot \|\Sigma_{K^*}\| \cdot \operatorname{tr}(E_K^\top E_K)$, which establishes the upper bound in (C.53).

Furthermore, for the lower bound, since $K' = K - (R + B^\top P_K B)^{-1} E_K$ attains the lower bound in (C.57) and $K^*$ is the optimal policy, similar to (C.55) and (C.56), we have

$$J(K) - J(K^*) \geq J(K) - J(K') = -\mathbb{E}_{x_0^* \sim N(0, \Psi_\sigma)}\left[\sum_{t \geq 0} A_{K, K'}(x_t')\right]$$
$$= \operatorname{tr}[\Sigma_{K'} E_K^\top (R + B^\top P_K B)^{-1} E_K] \geq \sigma_{\min}(\Psi) \cdot \|R + B^\top P_K B\|^{-1} \cdot \operatorname{tr}(E_K^\top E_K),$$

where in the first equality we define $x_t' = (A - BK')^t$ for all $t \geq 0$, and the last inequality follows from the fact that $\Sigma_{K'} \succeq \Psi \succeq \sigma_{\min}(\Psi) \cdot I_d$. Therefore, we conclude the proof of Lemma C.6. $\square$

Notice that $K = K^*$ achieves the minimum of $J(K)$. Lemma C.6 implies that

$$J(K) - J(K^*) \leq \lambda \cdot \langle E_K, E_K \rangle,$$

where $\lambda = 1/\sigma_{\min}(R) \cdot \|\Sigma_{K^*}\|$. That is, the difference of the objective can be bounded by the norm of the natural gradient. Therefore, updating the policy parameter $K$ in the direction of natural gradient $E_K$ yields decreases the objective value. Therefore, we conclude the first step.

**Step 2.** In the second part of the proof, equipped with Lemma C.6, we establish the global convergence of the natural actor-critic algorithm. Recall that we assume that the initial policy $\pi_{K_0}$ is stable, which implies that $J(K_0)$ is finite. Moreover, according to Algorithm 1, the policy parameters are updated via

$$K_{t+1} = K_t - \gamma \cdot \widehat{E}_{K_t}, \qquad \widehat{E}_{K_t} = \widehat{\Theta}_t^{22} K_t - \widehat{\Theta}_t^{21}, \tag{C.59}$$

where $\widehat{\Theta}_t$ is the estimator of $\Theta_{K_t}$ returned by Algorithm 2.

We use mathematical induction to show that $\{J(K_t)\}_{t \geq 0}$ is a monotone decreasing sequence. Suppose $J(K_t) \leq J(K_0)$. We define $K'_{t+1} = K_t - \gamma \cdot E_{K_t}$, i.e., $K'_{t+1}$ is obtained by a single step of natural policy gradient, starting from $K_t$. In the sequel, we use $J(K'_{t+1})$ to connect $J(K_t)$ and $J(K_{t+1})$. By Lemma C.5, we have

$$\begin{aligned}
J(K'_{t+1}) - J(K_t) &= \mathbb{E}_{x \sim N(0, \Psi_\sigma)}[x^\top (P_{K'_{t+1}} - P_{K_t}) x] \\
&= -2\gamma \cdot \mathrm{tr}\big(\Sigma_{K'_{t+1}} \cdot E_{K_t}^\top E_{K_t}\big) + \gamma^2 \cdot \mathrm{tr}\big[\Sigma_{K'_{t+1}} \cdot E_{K_t}^\top (R + B^\top P_{K_t} B) E_{K_t}\big] \\
&= -2\gamma \cdot \mathrm{tr}\big(\Sigma_{K'_{t+1}} \cdot E_{K_t}^\top E_{K_t}\big) + \gamma^2 \cdot \|R + B^\top P_{K_t} B\| \cdot \mathrm{tr}\big(\Sigma_{K'_{t+1}} \cdot E_{K_t}^\top E_{K_t}\big).
\end{aligned} \tag{C.60}$$

When $\gamma$ is sufficiently small such that

$$\gamma \cdot \big[\|R\| + \sigma_{\min}^{-1}(\Psi) \cdot \|B\|^2 \cdot J(K_0)\big] \leq 1, \tag{C.61}$$

by triangle inequality, we have

$$\gamma \cdot \|R + B^\top P_{K_t} B\| \leq \gamma \cdot \big[\|R\| + \|B\|^2 \cdot \|P_{K_t}\|\big] \leq \gamma \cdot \big[\|R\| + \sigma_{\min}^{-1}(\Psi) \cdot \|B\|^2 \cdot J(K_0)\big] < 1, \tag{C.62}$$

where the second inequality follows from Lemma C.1 and the induction assumption that $J(K_t) \leq J(K_0)$, and the last inequality follows from (C.61). Thus, combining (C.60) and (C.62), we have

$$\begin{aligned}
J(K'_{t+1}) - J(K_t) &\leq -\gamma \cdot \mathrm{tr}\big(\Sigma_{K'_{t+1}} \cdot E_{K_t}^\top E_{K_t}\big) \leq -\gamma \cdot \sigma_{\min}(\Psi) \cdot \mathrm{tr}\big(E_{K_t}^\top E_{K_t}\big), \\
&\leq -\gamma \cdot \sigma_{\min}(\Psi) \cdot \sigma_{\min}(R) \cdot \|\Sigma_{K^*}\|^{-1} \cdot \big[J(K_t) - J(K^*)\big].
\end{aligned} \tag{C.63}$$

where the third inequality follows from the fact that $\Sigma_{K'_{t+1}} \succeq \Psi$, and the last inequality follows from Lemma C.6. Note that (C.63) implies that $J(K'_{t+1}) \leq J(K_t) \leq J(K_0)$.

Furthermore, by the difference between $J(K_{t+1})$ and $J(K'_{t+1})$ can be bounded by

$$\begin{aligned}
\big|J(K_{t+1}) - J(K'_{t+1})\big| &= \big|\mathrm{tr}\big[(P_{K_{t+1}} - P_{K'_{t+1}}) \cdot \Psi_\sigma\big]\big| \leq \|\Psi_\sigma\|_{\mathrm{fro}} \cdot \big\|P_{K_{t+1}} - P_{K'_{t+1}}\big\| \\
&\leq \big[\|\Psi\|_{\mathrm{fro}} \cdot + \sigma^2 \cdot \|B\|_{\mathrm{fro}}^2\big] \cdot \big\|P_{K_{t+1}} - P_{K'_{t+1}}\big\|.
\end{aligned} \tag{C.64}$$

Now we utilize the following Lemma, obtained from [26], to construct and upper bound for $\|P_{K_{t+1}} - P_{K'_{t+1}}\|$.

**Lemma C.7** (Perturbation of $P_K$)**.** Suppose $\pi_{K'}$ is a small perturbation of $\pi_K$ in the sense that

$$\|K' - K\| \leq \sigma_{\min}(\Psi)/4 \cdot \|\Sigma_K\|^{-1} \|B\|^{-1} \cdot (\|A - BK\| + 1)^{-1}, \tag{C.65}$$

then we have

$$\begin{aligned}
\|P_{K'} - P_K\| \leq{} &6\sigma_{\min}^{-1}(\Psi) \cdot \|\Sigma_K\| \cdot \|K\| \cdot \|R\| \\
&\cdot \big(\|K\| \cdot \|B\| \cdot \|A - BK\| + \|K\| \cdot \|B\| + 1\big) \cdot \|K - K'\|.
\end{aligned} \tag{C.66}$$

*Proof.* This lemma is a slight modification of Lemma 24 in [26]. Here we sketch the proof. See [26, Lemmas 17 and 24] for a detailed proof.

Recall that we define operator $\mathcal{T}_K$ in (D.3). The operator norm of $\mathcal{T}_K$ is defined as $\|\mathcal{T}_K\| \leq \sup_\Omega \|\mathcal{T}_K(\Omega)\|/\|\Omega\|$, where the supremum is taken over all symmetric matrices. As shown in Lemma 17 in [26], we have $\|\mathcal{T}_K\| \leq \sigma_{\min}^{-1}(\Psi) \cdot \|\Sigma_K\|$. Moreover, under the condition in (C.65), in the proof of Lemma 24 in [26], it is shown that

$$\|P_{K'} - P_K\| \leq 6\|\mathcal{T}_K\| \cdot \|K\| \cdot \|R\| \cdot \big(\|K\| \cdot \|B\| \cdot \|A - BK\| + \|K\| \cdot \|B\| + 1\big) \cdot \|K - K'\|.$$

Combining this with the upper bound on $\|\mathcal{T}_K\|$, we conclude the proof. $\qquad\square$

To use this lemma, we need to verify (C.65). That is,

$$4\|K_{t+1} - K'_{t+1}\| \cdot (1 + \|A - BK'_{t+1}\|) \cdot \|B\| \cdot \|\Sigma_{K'_{t+1}}\| \le \sigma_{\min}(\Psi). \qquad \text{(C.67)}$$

By the definition of $K_{t+1}$ and $K'_{t+1}$, we have

$$\|K_{t+1} - K'_{t+1}\| = \gamma \cdot \|\widehat{E}_{K_t} - E_{K_t}\| \le \gamma \cdot \|\widehat{\Theta}_t - \Theta_{K_t}\|_{\text{fro}} \cdot (1 + \|K_t\|), \qquad \text{(C.68)}$$

where $\widehat{E}_{K_t}$ is defined in (C.59). Plugging (C.68) into the left-hand side of (C.67), we obtain that

$$4\|K_{t+1} - K'_{t+1}\| \cdot (1 + \|A - BK'_{t+1}\|) \cdot \|B\| \cdot \|\Sigma_{K'_{t+1}}\|$$
$$\le 4\gamma \cdot \|\widehat{\Theta}_t - \Theta_{K_t}\|_{\text{fro}} \cdot (1 + \|K_t\|) \cdot (1 + \|A - BK'_{t+1}\|) \cdot \|B\| \cdot \|\Sigma_{K'_{t+1}}\|. \qquad \text{(C.69)}$$

Utilizing Lemma (C.1) and the fact that $J(K'_{t+1}) \le J(K_0)$, we have

$$\|\Sigma_{K'_{t+1}}\| \le J(K'_{t+1})/\sigma_{\min}(Q) \le J(K_0)/\sigma_{\min}(Q). \qquad \text{(C.70)}$$

In addition, by triangle inequality, we have

$$\|A - BK'_{t+1}\| \le \|A - BK_t\| + \gamma \cdot \|B\| \cdot \|E_{K_t}\|$$
$$\le \|A - BK_t\| + \gamma \cdot \|B\| \cdot \|\Theta_{K_t}\| \cdot (1 + \|K_t\|). \qquad \text{(C.71)}$$

By the definition of $\Theta_K$ in (3.7), we have

$$\|\Theta_{K_t}\| \le \|Q\| + \|R\| + (\|A\|_{\text{fro}} + \|B\|_{\text{fro}})^2 \cdot \|P_{K_t}\|$$
$$\le \|Q\| + \|R\| + (\|A\|_{\text{fro}} + \|B\|_{\text{fro}})^2 \cdot J(K_0)/\sigma_{\min}(\Psi), \qquad \text{(C.72)}$$

where the last inequality follows from Lemma (C.1) and the induction assumption. Furthermore, by triangle inequality, it holds that

$$\|K_{t+1}\| \le \|K_t\| + \gamma \cdot \|E_{K_t}\| \le \|K_t\| + \gamma \cdot \|\Theta_{K_t}\| \cdot (1 + \|K_t\|)$$
$$\le \|K_t\| + \gamma \cdot \left[\|Q\| + \|R\| + (\|A\|_{\text{fro}} + \|B\|_{\text{fro}})^2 \cdot J(K_0)/\sigma_{\min}(\Psi)\right] \cdot (1 + \|K_t\|). \qquad \text{(C.73)}$$

In the sequel, we set

$$\gamma = \left[\|R\| + \sigma_{\min}^{-1}(\Psi) \cdot \|B\|^2 \cdot J(K_0)\right]^{-1}. \qquad \text{(C.74)}$$

Note that we assume that $\|Q\|$, $\|R\|$, $\|A\|$, $\|B\|$, $\sigma_{\min}(Q)$, $\sigma_{\min}(R)$ are all constants. Combining (C.69), (C.70), (C.71), and (C.72), we conclude that there exists a polynomial $\Upsilon_1(\cdot, \cdot)$ such that

$$4\|K_{t+1} - K'_{t+1}\| \cdot (1 + \|A - BK'_{t+1}\|) \cdot \|B\| \cdot \|\Sigma_{K'_{t+1}}\| \le \Upsilon_1\left[\|K_t\|, J(K_0)\right] \cdot \|\widehat{\Theta}_t - \Theta_{K_t}\|_{\text{fro}}. \qquad \text{(C.75)}$$

Furthermore, for the right-hand side of (C.66), combining (C.68), (C.69), (C.70), (C.71), (C.72), and (C.73). we conclude that there exists a polynomial $\Upsilon_2(\cdot, \cdot)$ such that

$$\left[\|\Psi\|_{\text{fro}} \cdot + \sigma^2 \cdot \|B\|_{\text{fro}}^2\right] \cdot 6\sigma_{\min}^{-1}(\Psi) \cdot \|\Sigma_{K'_{t+1}}\| \cdot \|K_{t+1'}\| \cdot \|R\|$$
$$\cdot \left(\|K'_{t+1}\| \cdot \|B\| \cdot \|A - BK'_{t+1}\| + \|K'_{t+1}\| \cdot \|B\| + 1\right) \cdot \|K_{t+1} - K'_{t+1}\|$$
$$\le \Upsilon_2\left[\|K_t\|, J(K_0)\right] \cdot \|\widehat{\Theta}_t - \Theta_{K_t}\|_{\text{fro}}. \qquad \text{(C.76)}$$

Meanwhile, in Theorem 4.2 we have shown that, there exists a polynomial $\Upsilon_3(\cdot, \cdot)$ such that, for $T$ sufficiently large, Algorithm 2 with $T$ iterations returns an estimator $\widehat{\Theta}_t$ for $\Theta_{K_t}$ such that

$$\|\widehat{\Theta}_t - \Theta_{K_t}\|_{\text{fro}} \le \frac{\Upsilon_3\left[\|K_t\|, J(K_0)\right]}{\kappa_{K_t}^* \cdot \sqrt{(1 - \rho)}} \cdot \frac{\log^3 T}{T^{1/4}} \qquad \text{(C.77)}$$

holds with probability at least $1 - T^{-4}$, where $\rho \in (\rho(A - BK_t), 1)$ and $\kappa_{K_t}^*$ is specified in Lemma 3.2, which depends only on $\rho$, $\sigma$, and $\sigma_{\min}(\Psi)$. Notice that $\log^3 T \cdot T^{-1/4} \le T^{-1/5}$ for $T$ sufficiently

large. Therefore, in the GTD algorithm for estimating $\Theta_{K_t}$, we set the number of iterations $T_t$ sufficiently large such that

$$\Upsilon_1\big[\|K_t\|, J(K_0)\big] \cdot \Upsilon_3\big[\|K_t\|, J(K_0)\big] \cdot {\kappa_{K_t}^*}^{-1} \cdot (1-\rho)^{-1/2} \cdot T_t^{-1/5} \leq \sigma_{\min}(\Psi),$$

$$\Upsilon_2\big[\|K_t\|, J(K_0)\big] \cdot \Upsilon_3\big[\|K_t\|, J(K_0)\big] \cdot {\kappa_{K_t}^*}^{-1} \cdot (1-\rho)^{-1/2} \cdot T_t^{-1/5}$$

$$\leq \epsilon/2 \cdot \sigma_{\min}(\Psi) \cdot \sigma_{\min}(R) \cdot \|\Sigma_{K^*}\|^{-1} \tag{C.78}$$

hold simultaneously. For such a $T_t$, combining (C.75) and (C.77), we conclude that (C.67) holds. Lemma C.7 implies that (C.66) is true. Combining (C.64), (C.66), (C.76), and (C.77), we conclude that

$$\big|J(K_{t+1}) - J(K'_{t+1})\big| \leq \epsilon/2 \cdot \sigma_{\min}(\Psi) \cdot \sigma_{\min}(R) \cdot \|\Sigma_{K^*}\|^{-1} \tag{C.79}$$

holds with probability at least $1 - T_t^{-4}$. Thus, when $J(K_t) - J(K^*) > \epsilon$, combining (C.63) and (C.79) we have

$$J(K_{t+1}) - J(K_t) \leq -\epsilon/2 \cdot \gamma\sigma_{\min}(\Psi) \cdot \sigma_{\min}(R) \cdot \|\Sigma_{K^*}\|^{-1} < 0.$$

Therefore, we have shown that, as long as $J(K_t) - J(K^*) \geq \epsilon$, $J(K_{t+1}) < J(K_t)$ holds with probability at least $1 - T_t^{-1/4}$.

Meanwhile, (C.63) implies that,

$$J(K'_{t+1}) - J(K^*) \leq \big[1 - \gamma \cdot \sigma_{\min}(\Psi) \cdot \sigma_{\min}(R) \cdot \|\Sigma_{K^*}\|^{-1}\big] \cdot \big[J(K_t) - J(K^*)\big]$$

By (C.79), when $J(K_t) - J(K^*) \geq \epsilon$, with probability $1 - T_t^{-4}$, we have

$$J(K_{t+1}) - J(K^*) \leq \big[1 - \gamma/2 \cdot \sigma_{\min}(\Psi) \cdot \sigma_{\min}(R) \cdot \|\Sigma_{K^*}\|^{-1}\big] \cdot \big[J(K_t) - J(K^*)\big],$$

which shows that, in terms of the policy parameter, natural actor-critic algorithm converges linearly. Specifically, with

$$N \geq 2\|\Sigma_{K^*}\|/\gamma \cdot \sigma_{\min}^{-1}(\Psi) \cdot \sigma_{\min}^{-1}(R) \cdot \log\big\{2[J(K_0) - J(K^*)]/\epsilon\big\} \tag{C.80}$$

policy updates, we have $J(K_N) - J(K^*) \leq \epsilon$ with high-probability, where $\gamma$ is specified in (C.74).

Finally, it remains to determine $T_t$ for all $t \in [N]$. Notice that $T_t$ satisfies the two inequalities in (C.78). Thus, we set

$$T_t \geq \Upsilon_4[\|K_t\|, J(K_0)] \cdot {\kappa_{K_t}^*}^{-5} \cdot (\Xi_{K_t}) \cdot \big[1 - \rho(A - BK_t)\big]^{-5/2} \cdot \epsilon^{-5}$$

for some polynomial function $\Upsilon_4(\cdot, \cdot)$. With such a $T_t$, the fail probability $T_t^{-4} \leq \epsilon^{-20}$. Notice that the total number of iterations depends on $\epsilon$ only through $\log(1/\epsilon)$. Thus, the total fail probability can be bounded by $\epsilon^{10}$. Therefore, we conclude the proof. $\qquad\square$

# D  Proofs of the Auxiliary Results

In this section, we provides the proofs for Proposition 3.1 and Lemma 3.2.

## D.1  Proof of Proposition 3.1

*Proof.* We first establish (3.8). Note that under $\pi_K$, we can write $u_t$ as $-Kx_t + \sigma \cdot \eta_t$, where $\eta_t \sim N(0, I_d)$. This implies that, for all $\geq 0$, we have

$$\mathbb{E}[c(x_t, u_t) \,|\, x_t] = x_t^\top Q x_t + \mathbb{E}_{\eta_t \sim N(0, I_d)}[(-Kx_t + \sigma \cdot \eta_t)^\top R(-Kx_t + \sigma \cdot \eta_t)]$$

$$= x_t^\top (Q + K^\top RK) x_t + \sigma^2 \cdot \operatorname{tr}(R). \tag{D.1}$$

Thus, combining (D.1) and the definition of $J(K)$ in (2.1), we have

$$J(K) = \lim_{T \to \infty} \mathbb{E}\bigg\{\frac{1}{T}\sum_{t \geq 0}^{T} \mathbb{E}[c(x_t, u_t) \,|\, x_t]\bigg\} = \lim_{T \to \infty} \mathbb{E}\bigg\{\frac{1}{T}\sum_{t \geq 0}^{T}[x_t^\top (Q + K^\top RK)x_t + \sigma^2 \cdot \operatorname{tr}(R)]\bigg\}$$

$$= \mathbb{E}_{x \sim \nu_K}[x^\top (Q + K^\top RK)x] + \sigma^2 \cdot \operatorname{tr}(R) = \operatorname{tr}\big[(Q + K^\top RK)\Sigma_K\big] + \sigma^2 \cdot \operatorname{tr}(R), \tag{D.2}$$

where the third inequality in (D.2) holds because the limiting distribution of $\{x_t\}_{t\geq 0}$ is $\nu_K$.

It remains to establish the second equality in (3.8). To this end, for $K \in \mathbb{R}^{k\times d}$ such that $\rho(A-BK) < 1$, we define operators we define $\mathcal{T}_K$ and $\mathcal{T}_K^\top$ by

$$\mathcal{T}_K(\Omega) = \sum_{t\geq 0}(A-BK)^t \Omega\big[(A-BK)^t\big]^\top, \qquad \mathcal{T}_K^\top(\Omega) = \sum_{t\geq 0}\big[(A-BK)^t\big]^\top \Omega(A-BK)^t,$$
(D.3)

where $\Omega \in \mathbb{R}^{d\times d}$ is positive definite. By definition, $\mathcal{T}_K(\Omega)$ and $\mathcal{T}_K^\top(\Omega)$ satisfy Lyapunov equations

$$\mathcal{T}_K(\Omega) = \Omega + (A-BK)\mathcal{T}_K(\Omega)(A-BK)^\top, \tag{D.4}$$
$$\mathcal{T}_K^\top(\Omega) = \Omega + (A-BK)^\top \mathcal{T}_K^\top(\Omega)(A-BK), \tag{D.5}$$

respectively. Moreover, for any positive definite matrices $\Omega_1, \Omega_2$, since $\rho(A-BK) < 1$, we have

$$\begin{aligned}\mathrm{tr}[\Omega_1 \cdot \mathcal{T}_K(\Omega_2)] &= \sum_{t\geq 0}\mathrm{tr}\big\{\Omega_1(A-BK)^t\Omega_2[(A-BK)^t]^\top\big\}\\ &= \sum_{t\geq 0}\mathrm{tr}\big\{[(A-BK)^t]^\top\Omega_1(A-BK)^t\Omega_2\big\} = \mathrm{tr}[\mathcal{T}_K^\top(\Omega_1)\cdot\Omega_2].\end{aligned} \tag{D.6}$$

Meanwhile, by combining (3.3), (3.4), (D.4), and (D.5), we have $\Sigma_K = \mathcal{T}_K(\Psi_\sigma)$ and $P_K = \mathcal{T}_K^\top(Q + K^\top RK)$. Thus, (D.6) implies that

$$\mathrm{tr}\big[(Q + K^\top RK)\cdot\Sigma_K\big] = \mathrm{tr}\big[(Q + K^\top RK)\cdot\mathcal{T}_K(\Psi_\sigma)\big] = \mathrm{tr}\big[\mathcal{T}_K^\top(Q + K^\top RK)\cdot\Psi_\sigma\big] = \mathrm{tr}(P_K\Psi_\sigma).$$

Combining this equation with (D.2), we establish the second equation of (3.8).

In the following, we establish the value functions. In the setting of LQR, the state-value function $V_K$ is given by

$$\begin{aligned}V_K(x) &= \sum_{t=0}^\infty\big\{\mathbb{E}[c(x_t, u_t)\,|\,x_0 = x, u_t = -Kx_t + \sigma\cdot\eta_t] - J(K)\big\}\\ &= \sum_{t=0}^\infty\big\{\mathbb{E}[x_t^\top(Q + K^\top RK)x_t] + \sigma^2\cdot\mathrm{tr}(R) - J(K)\big\}.\end{aligned} \tag{D.7}$$

Combining the linear dynamics in (3.2) and (D.7), we see that $V_K$ is a quadratic function, which is denoted by $V_k(x) = x^\top P_K x + \alpha_K$, where both $P_K$ and $\alpha_K$ depends on $K$. Note that $V_K$ satisfies the Bellman equation

$$V_K(x) = \mathbb{E}_{u\sim\pi_K}[c(x, u)] - J(K) + \mathbb{E}[V_K(x')\,|\,x],$$

where $x'$ is the next state given $(x, u)$. Thus, for any $x \in \mathbb{R}^d$, we have

$$x^\top P_K x = x(Q + K^\top RK)x + x^\top(A-BK)^\top P_K(A-BK)x.$$

Thus, $P_K$ is the unique positive definite solution to the Bellman equation in (3.4). Meanwhile, since $\mathbb{E}_{x\sim\nu_K}[V_K(x)] = 0$, we have $\alpha_K = -\mathrm{tr}(P_K\Sigma_K)$. Hence, we establish (3.5).

Furthermore, for any state-action pair $(x, u)$, we have

$$\begin{aligned}Q_K(x, u) &= c(x, u) - J(K) + \mathbb{E}[V_K(x')\,|\,x, u]\\ &= c(x, u) - J(K) + (Ax + Bu)^\top P_K(Ax + Bu) + \mathrm{tr}(P_K\Psi) - \mathrm{tr}(P_K\Sigma_K)\\ &= x^\top Qx + u^\top Ru + (Ax + Bu)^\top P_K(Ax + Bu) - \sigma^2\cdot\mathrm{tr}(R + P_K BB^\top) - \mathrm{tr}(P_K\Sigma_K),\end{aligned}$$

where $x'$ in the first equality is the next state following $(x, u)$, and the last equality follows from (3.8) and the fact that $\Psi_\sigma = \Psi + \sigma^2\cdot BB^\top$. Thus, we prove (3.6).

It remains to derive the policy gradient $\nabla_K J(K)$. By (3.8), we have

$$\nabla_K J(K) = 2RK\Sigma_K + \nabla_K \mathrm{tr}(Q_0 \cdot \Sigma_K)\big|_{Q_0 = Q + K^\top RK}, \tag{D.8}$$

where the second term denotes that we first take compute the gradient $\nabla_K \mathrm{tr}[Q_0\Sigma_K]$ with respect to $K$ and then set $Q_0 = Q + K^\top RK$. Recall that we can write $\Sigma_K = \mathcal{T}_K(\Psi_\sigma)$. The following lemma enables us to compute the gradient involving $\mathcal{T}_K$.

**Lemma D.1.** Let $W$ and $\Psi$ be two positive definite matrices. Then it holds that

$$\nabla_K \operatorname{tr}[W \cdot \mathcal{T}_K(\Psi)] = -2B^\top \mathcal{T}_K^\top(W)(A - BK)\mathcal{T}_K(\Psi).$$

*Proof.* To simplify the notation, we define operator $\mathcal{F}_K$ by

$$\mathcal{F}_K^\top(\Omega) = (A - BK)^\top \Omega (A - BK)$$

and let $\mathcal{F}_K^{\top,t}$ be the $t$-th composition of $\mathcal{F}_K$. Thus, by the definition of $\mathcal{T}_K^\top$ and $\mathcal{F}_K^\top$, we have

$$\mathcal{T}_K^\top(\Omega) = \sum_{t \geq 0} \mathcal{F}_K^{\top,t}(\Omega).$$

Moreover, by (D.4) we have

$$\operatorname{tr}[W \cdot \mathcal{T}_K(\Psi)] = \operatorname{tr}(W\Psi) + \operatorname{tr}[(A - BK)^\top W(A - BK) \cdot \mathcal{T}_K(\Psi)],$$

which implies that

$$\nabla_K \operatorname{tr}[W \cdot \mathcal{T}_K(\Psi)] = -2B^\top W(A - BK)\mathcal{T}_K(\Psi) + \nabla_K \operatorname{tr}[W_1 \mathcal{T}_K(\Psi)]\Big|_{W_1 = \mathcal{F}_K(\Omega)}. \tag{D.9}$$

For any $k \geq 1$, by recursively applying (D.9) for $k$ times, we have

$$\nabla_K \operatorname{tr}[W \cdot \mathcal{T}_K(\Psi)]$$

$$= -2B^\top \left[ \sum_{t=0}^{k} \mathcal{F}_K^{\top,t}(W) \right](A - BK)\mathcal{T}_K(\Psi) + \nabla_K \operatorname{tr}[W_1 \mathcal{T}_K(\Psi)]\Big|_{W_1 = \mathcal{F}_K^{(k+1)}(\Omega)}. \tag{D.10}$$

Meanwhile, since $\rho(A - BK) < 1$, we have

$$\lim_{k \to \infty} \operatorname{tr}[\mathcal{F}_K^{\top,k}(W)\mathcal{T}_K(\Psi)] \leq \lim_{k \to \infty} \|W\| \cdot \operatorname{tr}[\mathcal{T}_K(\Psi)] \cdot \rho(A - BK)^{2k} = 0.$$

Thus, by letting $k$ on the right-hand side of (D.10) go to infinity, we obtain

$$\nabla_K \operatorname{tr}[W \cdot \mathcal{T}_K(\Psi)] = -2B^\top \left[ \sum_{t=0}^{\infty} \mathcal{F}_K^{\top,t}(W) \right](A - BK)\mathcal{T}_K(\Psi) = -2B^\top \mathcal{T}_K^\top(W)(A - BK)\mathcal{T}_K(\Psi).$$

Therefore, we conclude the proof of the lemma. $\qquad\square$

By the above lemma, since $\Sigma_K = \mathcal{T}_K(\Psi_\sigma)$, we have

$$\nabla_K \operatorname{tr}(Q_0 \cdot \Sigma_K)\big|_{Q_0 = Q + K^\top RK} = \nabla_K \operatorname{tr}[Q_0 \cdot \mathcal{T}_K(\Psi_\sigma)]\big|_{Q_0 = Q + K^\top RK}$$

$$= -2B^\top \mathcal{T}_K^\top(Q + K^\top RK)(A - BK)\mathcal{T}_K(\Psi_\sigma) = -2B^\top P_K(A - BK)\Sigma_K, \tag{D.11}$$

where we use the fact that $P_K = \mathcal{T}_K^\top(Q + K^\top RK)$. Therefore, combining (D.8) and (D.11), we establish (3.9), which completes the proof of Proposition 3.1. $\qquad\square$

### D.2  Proof of Lemma 3.2

We present a stronger lemma than Lemma 3.2, whose proof automatically validates Lemma 3.2.

**Lemma D.2.** Suppose $\rho(A - BK) < 1$. Let $N(0, \widetilde{\Sigma}_K)$ be the stationary distribution of the state-action pair $(x, u)$ when following policy $\pi_K$. Then for $\Xi_K$ defined in (3.15), we have

$$\Xi_K = (\widetilde{\Sigma}_K \otimes_s \widetilde{\Sigma}_K) - (\widetilde{\Sigma}_K L^\top) \otimes_s (\widetilde{\Sigma}_K L^\top) = (\widetilde{\Sigma}_K \otimes_s \widetilde{\Sigma}_K)(I - L^\top \otimes_s L^\top). \tag{D.12}$$

Moreover, $\Xi_K$ is a invertible matrix whose operator norm is bounded by $2[\sigma^2 + (1 + \|K\|_{\mathrm{fro}}^2) \cdot \|\Sigma_K\|]$. There exists a positive number $\kappa_K^*$ such that the minimum singular value of the matrix in the left-hand side of (3.16) is lower bounded by a constant $\kappa_K^* > 0$, where $\kappa_K^*$ only depends on $\rho(A - BK)$, $\sigma$, and $\sigma_{\min}(\Psi)$. Furthermore, since $\Xi_K$ is invertible, the linear equation in (3.16) has unique solution $\vartheta_K^*$, whose first and second components are $J(K)$ and $\operatorname{svec}(\Theta_K)$, respectively.

*Proof.* Throughout the proof of Lemma D.2, for any state-action pair $(x, u) \in \mathbb{R}^{d+k}$, we denote the next state-action pair following policy $\pi_K$ by $(x', u')$. Then we can write

$$x' = Ax + Bu + \epsilon, \qquad u' = -Kx' + \sigma \cdot \eta = -KAx - KBu - K\epsilon + \sigma \cdot \eta, \qquad \text{(D.13)}$$

where $\epsilon \sim N(0, \Psi)$ and $\eta \in N(0, I_k)$. For notational simplicity, we denote $(x, u)$ and $(x', u')$ by $z$ and $z'$, respectively. Thus, we can write $z' = Lz + \varepsilon$, where we define

$$L = \begin{pmatrix} A & B \\ -KA & -KB \end{pmatrix} = \begin{pmatrix} I_d \\ -K \end{pmatrix} (A \quad B), \qquad \varepsilon = \begin{pmatrix} \epsilon \\ -K\epsilon + \sigma \cdot \eta \end{pmatrix}. \qquad \text{(D.14)}$$

Since it holds that $\rho(MN) = \rho(NM)$ for any two matrices $M$ and $N$ [32, Theorem 1.3.22], we have $\rho(L) = \rho(A - BK) < 1$. Meanwhile, by definition, $\varepsilon \in \mathbb{R}^{d+k}$ is a centered Gaussian random variable with covariance

$$\begin{pmatrix} \Psi & -\Psi K^\top \\ -K\Psi & K\Psi K^\top + \sigma^2 \cdot I_k \end{pmatrix}, \qquad \text{(D.15)}$$

which is denoted by $\widetilde{\Psi}_\sigma$ for notational simplicity. In addition, for $x \sim \nu_K$ and $u \sim \pi_K(\cdot \mid x)$, we denote the joint distribution of $z = (x, u)$ by $\widetilde{\nu}_K$, which is a centered Gaussian distribution in $\mathbb{R}^{d \times k}$. Since $x \sim N(0, \Sigma_K)$ and $u = -Kx + \sigma \cdot I_k$, we can write $\widetilde{\nu}_K$ as $N(0, \widetilde{\Sigma}_K)$, where $\widetilde{\Sigma}_K \in \mathbb{R}^{(d+k) \times (d+k)}$ can be written as

$$\widetilde{\Sigma}_K = \begin{pmatrix} \Sigma_K & -\Sigma_K K^\top \\ -K\Sigma_K & K\Sigma_K K^\top + \sigma^2 \cdot I_k \end{pmatrix} = \begin{pmatrix} 0 & 0 \\ 0 & \sigma^2 \cdot I_k \end{pmatrix} + \begin{pmatrix} I_d \\ -K \end{pmatrix} \Sigma_K \begin{pmatrix} I_d \\ -K \end{pmatrix}^\top. \qquad \text{(D.16)}$$

Thus, by triangle inequality we have

$$\left\| \widetilde{\Sigma}_K \right\|_{\text{fro}} \leq \sigma^2 \cdot k + \|\Sigma_K\| \cdot (d + \|K\|_{\text{fro}}^2), \qquad \left\| \widetilde{\Sigma}_K \right\| \leq \sigma^2 + (1 + \|K\|_{\text{fro}}^2) \cdot \|\Sigma_K\|, \qquad \text{(D.17)}$$

where in (D.17) we use the fact that $\|AB\|_{\text{fro}} \leq \|A\|_{\text{fro}} \cdot \|B\|$.

Furthermore, since $L$ defined in (D.14) satisfy $\rho(L) < 1$, $\widetilde{\Sigma}_K$ is the unique positive definite solution to the Lyapunov equation

$$\widetilde{\Sigma}_K = L\widetilde{\Sigma}_K L^\top + \widetilde{\Psi}_K, \qquad \text{(D.18)}$$

where $\widetilde{\Psi}_K$ is defined in (D.15). Moreover, the feature mapping can be written as $\phi(x, u) = \phi(z) = \text{svec}(zz^\top)$, which implies that

$$\begin{aligned} \phi(x, u) - \phi(x', u') &= \text{svec}\left[ zz^\top - (Lz + \varepsilon)(Lz + \varepsilon)^\top \right] \\ &= \text{svec}\left( zz^\top - Lzz^\top L^\top - Lz\varepsilon^\top - \varepsilon z^\top L^\top - \varepsilon\varepsilon^\top \right). \end{aligned}$$

Hence, since $\varepsilon$ is independent of $z$, by the definition of $\Xi_K$ in (3.15), we have

$$\Xi_K = \mathbb{E}_{z \sim \widetilde{\nu}_K}[\phi(z) \, \text{svec}(xx^\top - Lxx^\top L^\top - \widetilde{\Psi}_\sigma)^\top].$$

Now let $M$ and $N$ by any two matrices, by direct computation, we have

$$\begin{aligned} \text{svec}(M)^\top \Xi_K \, \text{svec}(N) &= \mathbb{E}_{z \sim \widetilde{\nu}_K}\left[ \langle zz^\top, M \rangle \cdot \langle zz^\top - Lzz^\top L^\top - \widetilde{\Psi}_\sigma, N \rangle \right] \\ &= \mathbb{E}_{z \sim \widetilde{\nu}_K}\left[ z^\top M z z^\top (N - L^\top NL)z \right] - \mathbb{E}_{z \sim \widetilde{\nu}_K}[z^\top M z] \cdot \langle \widetilde{\Psi}_\sigma, N \rangle \\ &= \mathbb{E}_{g \sim N(0, I_{d+k})}\left[ g^\top \widetilde{\Sigma}_K^{1/2} M \widetilde{\Sigma}_K^{1/2} g g^\top \widetilde{\Sigma}_K^{1/2} (N - L^\top NL)\widetilde{\Sigma}_K^{1/2} g \right] - \langle \widetilde{\Sigma}_K, M \rangle \cdot \langle \widetilde{\Psi}_\sigma, N \rangle, \end{aligned}$$
$$\text{(D.19)}$$

where $\widetilde{\Sigma}_K^{1/2}$ is the square root of $\widetilde{\Sigma}_K$ defined in (D.18). We utilize the following Lemma to compute the expectation of the product of quadratic forms of Gaussian random variables.

**Lemma D.3.** Let $g \sim N(0, I_d)$ be the standard Gaussian random variable in $\mathbb{R}^d$ and let $A_1, A_2$ be two symmetric matrices. Then we have

$$\mathbb{E}[g^\top A_1 g \cdot g^\top A_2 g] = 2\operatorname{tr}(A_1 A_2) + \operatorname{tr}(A_1) \cdot \operatorname{tr}(A_2).$$

*Proof.* See, e.g., [48, 43] for a detailed proof. $\qquad \square$

Applying this lemma to (D.19), we have

$$
\begin{aligned}
\mathrm{svec}(M)^\top &\Xi_K\, \mathrm{svec}(N) \\
&= 2\,\mathrm{tr}\big[\widetilde{\Sigma}_K^{1/2} M \widetilde{\Sigma}_K^{1/2} \cdot \widetilde{\Sigma}_K^{1/2}(N - L^\top N L)\widetilde{\Sigma}_K^{1/2}\big] \\
&\quad + \mathrm{tr}\big(\widetilde{\Sigma}_K^{1/2} M \widetilde{\Sigma}_K^{1/2}\big) \cdot \mathrm{tr}\big[\widetilde{\Sigma}_K^{1/2}(N - L^\top N L)\widetilde{\Sigma}_K^{1/2}\big] - \big\langle \widetilde{\Sigma}_K, M\big\rangle \cdot \big\langle \widetilde{\Psi}_\sigma, N\big\rangle \\
&= 2\big\langle M, \widetilde{\Sigma}_K(N - L^\top N L)\widetilde{\Sigma}_K\big\rangle + \big\langle M, \widetilde{\Sigma}_K\big\rangle \cdot \big[\big\langle N - L^\top N L, \widetilde{\Sigma}_K\big\rangle - \big\langle \widetilde{\Psi}_\sigma, N\big\rangle\big]. \quad \text{(D.20)}
\end{aligned}
$$

Note that $\widetilde{\Sigma}_K$ satisfy the Lyapunov equation in (D.18), which implies that

$$
\big\langle N - L^\top N L, \widetilde{\Sigma}_K\big\rangle = \big\langle N, \widetilde{\Sigma}_K\big\rangle - \big\langle N, L\widetilde{\Sigma}_K L^\top\big\rangle = \big\langle N, \widetilde{\Psi}_\sigma\big\rangle.
$$

Thus, by (D.20) we have

$$
\begin{aligned}
\mathrm{svec}(M)^\top \Xi_K\, \mathrm{svec}(N) &= 2\big\langle M, \widetilde{\Sigma}_K(N - L^\top N L)\widetilde{\Sigma}_K\big\rangle = 2\,\mathrm{svec}(M)^\top\,\mathrm{svec}\big[\widetilde{\Sigma}_K(N - L^\top N L)\widetilde{\Sigma}_K\big] \\
&= 2\,\mathrm{svec}(M)^\top\big(\widetilde{\Sigma}_K \otimes_s \widetilde{\Sigma}_K - \widetilde{\Sigma}_K L^\top \otimes_s \widetilde{\Sigma}_K L^\top\big)\,\mathrm{svec}(N)^\top \\
&= 2\,\mathrm{svec}(M)^\top\big[\big(\widetilde{\Sigma}_K \otimes_s \widetilde{\Sigma}_K\big)(I - L^\top \otimes L^\top)\big]\,\mathrm{svec}(N),
\end{aligned}
$$

where the last equality follows from the fact that

$$
(A \otimes_s B)(C \otimes_s D) = 1/2 \cdot (AC \otimes_s BD + AD \otimes_s BC)
$$

holds for any matrices $A, B, C, D$. Thus, we have established (D.12). Since $\rho(L) = \rho(A - BK) < 1$, $I - L^\top \otimes L^\top$ is positive definite, which implies that $\Xi_K$ is invertible.

Now we consider the linear equation in (3.16). Since $\Xi_K$ is invertible,

$$
\widetilde{\Xi}_K = \begin{pmatrix} 1 & 0 \\ \mathbb{E}_{(x,u)}[\phi(x,u)] & \Xi_K \end{pmatrix} = \begin{pmatrix} 1 & 0 \\ \mathrm{svec}(\widetilde{\Sigma}_K) & \Xi_K \end{pmatrix} \quad \text{(D.21)}
$$

is also invertible. Thus, (3.16) has unique solution $\vartheta_K^*$. Moreover, to bound the smallest singular value of $\widetilde{\Xi}_K$, we note that the inverse of $\widetilde{\Xi}_K$ can be written as

$$
\widetilde{\Xi}_K^{-1} = \begin{pmatrix} 1 & 0 \\ -\Xi_K^{-1}\,\mathrm{svec}(\widetilde{\Sigma}_K) & \Xi_K^{-1} \end{pmatrix},
$$

whose operator norm is bounded via

$$
\big\|\widetilde{\Xi}_K^{-1}\big\|^2 \le 1 + \big\|\Xi_K^{-1}\,\mathrm{svec}(\widetilde{\Sigma}_K)\big\|_2^2 + \|\Xi_K^{-1}\|^2. \quad \text{(D.22)}
$$

By (D.12), we have

$$
\begin{aligned}
\Xi_K^{-1}\,\mathrm{svec}(\widetilde{\Sigma}_K) &= (I - L^\top \otimes_s L^\top)^{-1}(\widetilde{\Sigma}_K \otimes_s \widetilde{\Sigma}_K)^{-1}\,\mathrm{svec}(\widetilde{\Sigma}_K) \\
&= (I - L^\top \otimes_s L^\top)^{-1}(\widetilde{\Sigma}_K^{-1} \otimes_s \widetilde{\Sigma}_K^{-1})\,\mathrm{svec}(\widetilde{\Sigma}_K) = (I - L^\top \otimes_s L^\top)^{-1}\,\mathrm{svec}(\widetilde{\Sigma}_K^{-1}). \quad \text{(D.23)}
\end{aligned}
$$

The following lemma characterizes the eigenvalues of symmetric Kronecker matrices.

**Lemma D.4** (Lemma 7.2 in [2])**.** Let $A$ and $B$ be two matrices in $\mathbb{R}^{m \times m}$ that can be diagonalized simultaneously. Moreover, let $\lambda_1, \ldots, \lambda_m$ and $\mu_1, \ldots, \mu_m$ be the eigenvalues of $A$ and $B$, respectively. Then, the eigenvalues of $A \otimes_s B$ are given by $\{1/2 \cdot (\lambda_i \mu_j + \lambda_j \mu_i), i, j \in [m]\}$.

By Lemma D.4, the spectral radius of $L^\top \otimes_s L^\top$ is bounded by $\rho^2(L) = \rho^2(A - BK) < 1$. By (D.23) we have

$$
\big\|\Xi_K^{-1}\,\mathrm{svec}(\widetilde{\Sigma}_K)\big\|_2 \le \big[1 - \rho^2(L)\big]^{-1} \cdot \|\widetilde{\Sigma}_K^{-1}\|_F \le \sqrt{d+k} \cdot \big[1 - \rho^2(L)\big]^{-1} \cdot \|\widetilde{\Sigma}_K^{-1}\|. \quad \text{(D.24)}
$$

Besides, by (D.12) we have

$$
\|\Xi_K^{-1}\| \le \big\|(I - L^\top \otimes_s L^\top)^{-1}\big\| \cdot \big\|\widetilde{\Sigma}_K^{-1} \otimes_s \widetilde{\Sigma}_K^{-1}\big\| \le \big[1 - \rho^2(L)\big]^{-1} \cdot \big\|\widetilde{\Sigma}_K^{-1}\big\|^2. \quad \text{(D.25)}
$$

Notice that $\|\widetilde{\Sigma}_K^{-1}\| = 1/\sigma_{\min}(\widetilde{\Sigma}_K)$. Hence, combining (D.22), (D.24), and (D.25) we conclude that

$$
\big\|\widetilde{\Xi}_K^{-1}\big\|^2 \le 1 + (d+k) \cdot \big[1 - \rho(L)^2\big]^{-2} \cdot [\sigma_{\min}(\widetilde{\Sigma}_K)]^{-2} + \big[1 - \rho(L)^2\big]^{-2} \cdot [\sigma_{\min}(\widetilde{\Sigma}_K)]^{-4},
$$

which implies that

$$\sigma_{\min}(\widetilde{\Xi}_K) \geq \frac{[1 - \rho^2(A - BK)] \cdot [\sigma_{\min}(\widetilde{\Sigma}_K)]^2}{\left(1 + [1 - \rho^2(A - BK)]^2 \cdot [\sigma_{\min}(\widetilde{\Sigma}_K)]^4 + (d + k) \cdot [\sigma_{\min}(\widetilde{\Sigma}_K)]^2\right)^{1/2}} > 0.$$

Moreover, to see that $\sigma_{\min}(\widetilde{\Sigma}_K)$ only depends on $\sigma$ and $\sigma_{\min}(\Psi)$, for any $a \in \mathbb{R}^d$ and $b \in \mathbb{R}^k$, we have

$$\begin{pmatrix} a \\ b \end{pmatrix}^\top \widetilde{\Sigma}_K \begin{pmatrix} a \\ b \end{pmatrix} = \mathbb{E}_{(x,u) \sim \widetilde{\nu}_K}[(a^\top x + b^\top u)^2] = \mathbb{E}_{x \sim \nu_K, \eta \sim N(0, I_k)}\{[(a - K^\top b)x + \sigma \cdot \eta]^2\}$$

$$\geq \sigma^2 \cdot \|b\|_2^2 + \sigma_{\min}(\Psi) \cdot \|a - K^\top b\|_2^2 \geq (\sigma^2 - \sigma_{\min}(\Psi) \cdot \|K\|^2) \cdot \|b\|_2^2 + \sigma_{\min}(\Psi) \cdot \|a\|_2^2.$$

Thus, suppose $\sigma^2$ is sufficiently large such that $\sigma^2 - \sigma_{\min}(\Psi) \cdot \|K\|^2 > 0$, $\sigma_{\min}(\widetilde{\Sigma}_K)$ is lower bounded by $\min\{\sigma^2 - \sigma_{\min}(\Psi) \cdot \|K\|^2, \sigma_{\min}(\Psi)\}$. Therefore, we can find a constant $\kappa_K^*$ depending only on $\rho(A - KB)$, $\sigma$, and $\sigma_{\min}(\Psi)$ such that $\sigma_{\min}(\widetilde{\Xi}_K) \geq \kappa_K^*$.

Finally, to obtain an upper bound on $\|\Xi_K\|$, by triangle inequality and Lemma D.4 we have

$$\|\Xi_K\| \leq \left\|\widetilde{\Sigma}_K \otimes_s \widetilde{\Sigma}_K\right\| \cdot \left(1 + \|L^\top \otimes_s L^\top\|\right) \leq \left\|\widetilde{\Sigma}_K\right\|^2 \cdot \left(1 + \|L\|^2\right) \leq 2\left\|\widetilde{\Sigma}_K\right\|^2,$$

where we use the fact that $\rho(L) < 1$. Applying (D.17) to the inequality above, we obtain that

$$\|\Xi_K\| \leq 2\big[\sigma^2 + (1 + \|K\|_{\mathrm{fro}}^2) \cdot \|\Sigma_K\|\big],$$

which concludes the proof. $\qquad\qquad\square$

[Supplementary Material 3]

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

 \big[1 - C_1 \cdot \gamma \cdot \|\Sigma_{K^*}\|^{-1}\big] \cdot \big[J(K_t) - J(K^*)\big] \tag{4.6}$$

for some constant $C_1 > 0$. In addition, for policy $\pi_{K_t}$, when the number of GTD iteration $T_t$ is sufficiently large, $K_{t+1}$ is close to $K'_{t+1}$, which further implies that $|J(K'_{t+1}) - J(K_{t+1})|$ is small. Thus, combining this and (4.6), we obtain the linear convergence of the actor-critic algorithm. See §C.2 for a detailed proof. $\qquad\square$

This theorem shows that natural actor-critic algorithm combined with GTD converges linearly to the optimal policy of LQR. Furthermore, the number of policy updates in this theorem matches those obtained by natural policy gradient algorithm [26, 44]. To the best of our knowledge, this result seems to be the first nonasymptotic convergence result for actor-critic algorithms with function approximation, whose existing theory are mostly asymptotic and based on ODE approximation. Furthermore, from the viewpoint of bilevel optimization, Theorem 4.3 offers theoretical guarantees for the actor-critic algorithm as a first-order online method for the bilevel program defined in (3.21), which serves a first attempt of understanding bilevel optimization with possibly nonconvex subproblems.

Furthermore, although we only consider the problem of LQR and analyze the natural actor-critic with GTD for policy evaluation, our theoretical framework can be applied to general reinforcement learning problems with other policy optimization methods for the actor (e.g. vanilla policy gradient [68], trust-region policy optimization [57] and proximal policy optimization [58]) and other policy evaluation methods for the critic such as TD(0) [65], least-square temporal-difference (LSTD) [13], and Retrace [47]. In particular, suppose the critic adopts compatible features [68] for policy evaluation using nonconvex optimization techniques, it can be shown that vanilla policy gradient converges to a local minimum of the expected total return with a sublinear rate [75]. Moreover, by leveraging the geometry of the expected total return as a functional of the policy, recently, [1, 39, 71, 59] prove that natural policy gradient [34], TRPO and PPO are all able to find the globally optimal policy. Using similar approaches, we can establish the convergence and global optimality of actor-critic methods.