[Reviews · NeurIPS 2019]

Reviewer 1



Originality =========== The paper extends previous analysis for the population version of policy gradient algorithm for LQR to the actor-critic algorithm with GTD critic. In addition, previous analysis only considered deterministic dynamics and deterministic policies, while the current one studies ergodic system with Gaussian noise and Gaussian policies. Quality =========== The paper seems theoretically sound. The proofs in the appendix are very well detailed, although I have not checked them carefully. The authors put their results in context with previous analysis, like the nonasymptotic analysis of GTD due to [17]. It is not clear how Assumption B.1 can be ensured in a model free setting, with no knowledge of A, B, Q or R. Numerical experiments illustrating the analysis even for simple MDPs would have strengthened the paper. Clarity =========== The paper is generally clear and well organised. Some notation is not properly introduced in a few occasions. Significance =========== The paper considers the LQR with Gaussian policy and dynamics, a model that has proved successful in several robotics applications. The contributed analysis can potentially shed some light on whether a model-free actor-critic algorithm is the right approach as opposed to learning the model dynamics. The techniques used for the analysis might be useful for other algorithms with similar linear updates. On the other hand, the authors claim that their analysis may serve as a preliminary step towards a complete theoretical understanding of bilevel optimization with nonconvex subproblems. However, the proposed techniques seem quite specific for the linear update setting, and I haven't found clear evidence that supports this claim in the text.

Reviewer 2



This is purely a theoretical paper. Although I'm not very familiar with this type of theory, the results seem sound from my review. The paper is quite well written, with just a few minor suggestions below. My main criticiism of this paper is that it is quite hard to read and follow if one is not very familiar with bilevel optimization for LQR (like myself). After reading the paper it is not at all clear to me why any of these results would carry over to anything outside of LQR; and with that in mind, it seems like there's not much we can learn from these results, outside of the LQR setting. I think there needs to be more high-level discussion in the paper to give readers not familiar with tihs type of theory a better idea for what is being proved, why it matters, and what we can learn from it. In particular, the paragraph starting at line 298 should be in its own "Discussion" section and expanded. Some questions: - In line 132, you say "Specifically, it is shown that policy iteration...", do you mean that it's shown in this paper or that it _has_ been shown previously? - In equation 3.1, shouldn't the subscript of I be k? - In line 152: what is \rho? Is it the stationary distribution? Relative to which policy? Minor suggestions: - line 114: "Viewing LQR from the lens of *an* MDP, the state..." - line 115: remove the "Besides, " - In equation 2.6, better to use a letter other than Q to avoid cofusion with the Q-value function. - line 120: "... it is known that the optimal action*s* are linear in..." - line 150: "the state dynamics *are* given by a linear..." - line 192: "denote svec(...) by $\theta...$" (remove the first "by") - line 238: "be updated *at* a faster pace." - line 240: "... ensures that the critic updates *at* a faster timescale." -

Reviewer 3



This is a nice paper that establishes convergence of actor-critic algorithm in the special case of LQR. The paper entirely addresses theoretical analysis, and continues the line of analysis of RL for LQR. While certain pieces of analysis are built on a recent paper on analysis of policy gradient methods for LQR (Fazel et al ICML 2018), analysis of actor-critic style algorithms seems more challenging due to the interaction during learning between actor and critic. The analysis techniques departed from the traditional ODE-based approach for actor-critic algorithms. The paper is well-written for the most part. Overall, I believe this would make a nice addition to NeurIPS. I do think that the bilevel optimization perspective, while interesting, does not really contribute much to the exposition of either the algorithm, or the analysis. It appears that the paper did not leverage much from bilevel optimization literature. Going the other direction, i.e. using what is presented as contribution to bilevel optimization seems unclear to me. Equation 3.18: it is stated (line 215) that the objective in equation 3.18 can be estimated unbiasedly using two consecutive state-action pairs. I’m not sure if I understand why that is the case. Would you come up against the double sampling problem for policy evaluation here? Some minor comments: - Equation (3.16), the symbols are used before being introduced in line 210 - Looks like there is a missing constant on line 294 (for some constant C is what you meant) - The analysis in the paper is for the on policy version of actor-critic. So algorithm 3 does not really come with the same guarantees. Perhaps the main body of the paper should make this clear.

[Author Response · NeurIPS 2019]

We thank the reviewers for their valuable comments and for pointing out the typos and clarity issues. We will revise
accordingly. We first explain the novelty and significance of our work and then answer the questions of each reviewer.

**(Novelty and Significance.)** Our novel bilevel optimization formulation of actor-critic works for general RL problems,
which motivates our algorithm which solves the policy evaluation subproblem at a faster timescale. We seem to first
establish the finite-time convergence of actor-critic for LQR with ergodic cost. Even for general RL problems, existing
convergence analyses are asymptotic and based on ODE approximations. Moreover, we develop a novel theoretical
analysis framework that decouples optimization problems faced by the actor and critic respectively. More importantly,
although we focus on LQR, our analysis can be readily extended to general RL problems with other policy optimization
methods for the actor (e.g. PPO and TRPO) and other policy evaluation evaluation methods (e.g. TD(0)). Applying
our analysis framework to general RL problems, we can show that actor-critic converges to a stationary point when
using compatible value function, which will be added in the revision. Moreover, our analysis of GTD seems the first
discrete-time convergence guarantee for policy evaluation under the ergodic setting, where the Bellman equation takes a
different form from the discounted setting. We will clarify our contributions in the revision.

**Reviewer #1**

**(Assumption B.1.)** Projection is only for the technical reason, which is used to obtain the sublinear convergence rate for
GTD updates. The convergence of GTD can be established without explicit projection thanks to to the convex-concave
structure. But more careful analysis are needed to obtain similar convergence rate without projection.

**(Gaussian policy.)** Due to the existence of the score function $\nabla_K \log \pi_K(u\,|\,x)$ in the policy gradient theorem, actor-
critic algorithm requires a stochastic policy. Here we adopt Gaussian policy for simplicity. In general, we could let the
policy be $u = -Kx + \epsilon$, where $\epsilon$ is a independent noise with zero mean and a known density. In this case, actor-critic
also finds the optimal policy due to 1) the family of value functions are compatible to the policy parametrization and 2)
any saddle point of $J(K)$ is the optimal policy (geometry of LQR). We will add a detailed discussion in the revision.
**(Minor comments.)** **(i) Numerical experiments.** We will add numerical experiments to illustrate the convergence
rate of our algorithm. **(ii) Line 23.** In the general case, we can only show policy gradient converges to a stationary point
of the objective. However, there is no guarantee on the performance of this learned policy, which can be arbitrarily
worse compared with the optimal policy. Thus, characterizing the optimality of policy gradient is an open problem. **(iii)**
**Line 63.** By "asynchrony" we mean the fact the online actor-critic involves coupled updates of the actor and the critic.
From the perspective of the critic, it aims to evaluate the value function of the critic, which is also changing. This fact
caused great trouble in the analysis of actor-critic. **(iv) Line 68.** $\mathrm{svec}(X)$ maps a symmetric matrix $X \in \mathbb{R}^{d \times d}$ to a
vector in $\mathbb{R}^{d(d+1)/2}$. The definition is given in §1 and we will add more details in the revision. **(v) Dual ascent.** In the
dual ascent view of actor-critic, the dual variable $\mu$ is a probability distribution over $\mathcal{X} \times \mathcal{U}$, which can be written as
$\mu = \rho_\pi \otimes \pi$ for some policy $\pi$, where $\rho_\pi$ is the stationary distribution over $\mathcal{X}$ induced by $\pi$. Directly optimizing the
dual variable requires a strong simulator that is able to sample arbitrary state-action pairs.

**Reviewer #4**
**(Discussion.)** The significance of this work should be considered in the reinforcement learning field. Indeed, both our
algorithm and theoretical framework can be directly applied to general reinforcement learning problems. Our GTD
analysis can be readily applied to general RL. For these problems, we can establish the convergence of actor-critic to
the stationary point of $J(\pi_\omega)$. Also see **(Novelty and Significance.)**.
**(i) Line 132.** The papers cited in Line 132 have established that policy iteration, dynamic programming, and policy
gradient are able to find the optimal policy of LQR. We will clarify this in the revision. **(ii) Line 152.** $\rho(A - BK)$ is
the spectral radius of matrix $A - BK$, which is different from $\rho_K$, the stationary distribution of $\pi_K$. We will change
the notation and clarify this. **(iii) Equation 3.1.** It should be $I_k$ instead of $I_d$ as $u \in \mathbb{R}^k$. Thanks for pointing out.

**Reviewer #5**
**(Bilevel Perspective.)** Here we use the bilevel optimization view to motivate our algorithm which updates the critic at
the faster pace as it solves the lower-level subproblem. We do not contribute to general bilevel optimization but our
focus is on RL: we provide a finite-time convergence analysis for actor-critic. Also see **(Novelty and Significance.)**.
**(Eqn. (3.18).)** In policy evaluation, at $(x, u) \in \mathcal{X} \times \mathcal{U}$, "Double sampling" refers to sampling two next states-action
pairs $(x_1, u_1)$ and $(x_2, u_2)$ where $x_1$ and $x_2$ are independent and sampled from $P(\cdot\,|\,x, u)$. This is not practical.
However, eqn.(3.18) only utilizes $(x, u)$ and $(x_1, u_1)$, which are two consecutive state-action pairs in a Markov chain.
**(Off-Policy AC.)** When the importance sampling ratio $\tau_K(x, u) = \pi_K(u\,|\,x)/\pi_b(u\,|\,x)$ satisfy certain regularity
conditions, using the optimization formulation in (A.2), we can similarly establish the sublinear convergence rate of
off-policy GTD, which yields an estimator of the natural policy gradient direction. Thus, we can similarly establish the
convergence guarantees for off-policy AC. We will add these theoretical results in the revision.
**(Comparison with Fazel et al.)** Fazel et al. show that on-policy policy gradient converges to the optimal policy
of undercounted LQR. We study actor-critic for LQR with ergodic costs and noisy state dynamics. They focus on
deterministic policies and estimate the policy gradient via zeroth-order optimization. We adopt Gaussian policy with on-
and off-policy actor-critic. Although our analysis of policy updates shares some similarity to Fazel et al., our analysis of
critic GTD updates is novel and our theoretical framework can be extended to actor-critic for general RL problems.

[Meta-Review · NeurIPS 2019]

The paper develops finite-time convergence to global optimum for actor-critic in LQR problems. Actor-critic algorithms are hard to analyze, but by focusing on LQR the paper manages to obtain fairly strong results. While LQR in some ways is easier than typical (general) MDPs considered in many RL papers, it is an important problem in robotics and control, so novel and solid results like this work would be nice to publish. The main problems raised by reviewers are (1) lack of (even simple) numerical demonstration; (2) some issues with writing that are mentioned in the detailed reviews.